# Learning Mixtures of Experts with EM: A Mirror Descent Perspective

**Quentin Fruytier** [1]   **Aryan Mokhtari** [1]   **Sujay Sanghavi** [1]

## Abstract

Classical Mixtures of Experts (MoE) are Machine Learning models that involve partitioning the input space, with a separate "expert" model trained on each partition. Recently, MoE-based model architectures have become popular as a means to reduce training and inference costs. There, the partitioning function and the experts are both learnt jointly via gradient descent-type methods on the log-likelihood. In this paper we study theoretical guarantees of the Expectation Maximization (EM) algorithm for the training of MoE models. We first rigorously analyze EM for MoE where the conditional distribution of the target and latent variable conditioned on the feature variable belongs to an exponential family of distributions and show its equivalence to projected Mirror Descent with unit step size and a Kullback-Leibler Divergence regularizer. This perspective allows us to derive new convergence results and identify conditions for local linear convergence; In the special case of mixture of 2 linear or logistic experts, we additionally provide guarantees for linear convergence based on the signal-to-noise ratio. Experiments on synthetic and (small-scale) real-world data supports that EM outperforms the gradient descent algorithm both in terms of convergence rate and the achieved accuracy.

## 1. Introduction

Classical Mixtures of Experts (MoE) (Jacobs et al., 1991; Jordan & Jacobs, 1994) are a crucial class of parametric latent variable models that have gained significant popularity in deep learning for their ability to reduce both training and inference costs (Chen et al., 2022). MoE are particularly effective when the feature space can be divided and processed by specialized models, known as experts. Instead of relying on a single, large model to handle all input-output mappings, MoE utilize an ensemble of specialized experts. Each expert is responsible for a specific subset of the input space, allowing the system to efficiently route inputs to the most appropriate expert. This partitioning enables each expert to focus on mapping its designated inputs to outputs using a separate, optimized model. By leveraging multiple specialized experts rather than a monolithic model, MoE can achieve greater scalability and flexibility. This modular approach not only enhances computational efficiency but also allows for improved performance, as each expert can be finely tuned to handle its particular segment of the input space effectively. Real-world applications of MoE such as Sparse MoE span across various domains, including language translation, speech recognition, recommendation systems, and more (Fedus et al., 2022; Ma et al., 2018; Hinton et al., 2012; Liu et al., 2024).

In its most generic form, training an MoE model involves training both *(a)* the parameters in the individual experts, and *(b)* the gating function that routes inputs to the appropriate expert. Typically, these are both learnt jointly by first formulating the final loss function as applied to the ensembled output, and then minimizing this joint loss function (via SGD or its variants) over the parameters of the gate and the experts.

In this paper we investigate, primarily from a theoretical perspective, the training of classical MoE as defined by Jacobs et al. (1991) using a classic algorithm: Expectation Maximization (EM). As opposed to SGD-based methods which are agnostic to whether a parameter is in the gate or in an expert, EM first formulates two separate problems – one for the router, and another for the experts – in a specific way. It then solves each problem in isolation and in parallel, and then collates the outputs to arrive at the updated set of gate and expert parameters.

For our theoretical results, we consider the setting of general MoE where the joint distribution of the target and latent variable conditioned on the feature variable belongs to an exponential family of distributions. We then narrow in on two simpler instances of MoE models: Mixture of Linear Experts (where each expert is a linear regressor) and Mix-

---

[1]Electrical and Computer Engineering Department, University of Texas at Austin, Austin, Texas, United States of America. Correspondence to: Quentin Fruytier <qdf76@my.utexas.edu>, Aryan Mokhtari <mokhtari@austin.utexas.edu>, Sujay Sanghavi <sanghavi@mail.utexas.edu>.

*Proceedings of the $42^{nd}$ International Conference on Machine Learning*, Vancouver, Canada. PMLR 267, 2025. Copyright 2025 by the author(s).

ture of Logistic Experts (where each expert is a logistic regressor). The router in each case is a linear softmax.

**Main Contributions:** The primary finding of this paper is to unveil the correspondence between EM for general MoE to projected Mirror Descent. A similar correspondence was first discovered by Kunstner et al. (2021), but was limited to generative models for which the joint complete data distribution belongs to an exponential family of distributions; this did not include MoE. As such, our contributions can be seen as a generalization of (Kunstner et al., 2021) to all generative models for which the conditional distribution of the target and latent variables conditioned on the feature variable belongs to an exponential family of distributions. We next state the details of our contributions.

1) In Theorem 4.1, we show that when EM is applied to general MoE, the iterates are equivalent to the ones generated by projected Mirror Descent on the conditional likelihood function with unit step-size and KL divergence. Here, the projection is an expectation moment projection over the parameter space. By leveraging this correspondence, in Theorem 4.2, we obtain sufficient conditions for which EM applied to general MoE converges to a stationary point or the true parameters. We further characterize the explicit convergence rate for each of the considered settings.

2) Next, in Theorem 5.1, we narrow in on the special cases of mixtures of 2 linear or logistic experts and show EM is equivalent to Mirror Descent with unit step-size and KL divergence, without requiring any extra projection. This perspective allows us to recover classic MD convergence results which we contextualize for our setting in Corollary B.1. Finally, we characterize the sufficient conditions for convergence in terms of the Missing Information Matrix (MIM) in Theorem B.2 and, subsequently, the signal-to-noise ratio (SNR) of the generative model in Theorem B.4.

3) Finally, on synthetic and small scale proof of concept real world datasets, we observe that EM outperforms gradient descent both in terms of convergence rate and the achieved performance. As well as supporting our theoretical results, this re-iterates the power of the EM algorithm for fitting MoE previously suggested by (Jordan & Jacobs, 1994; Jordan & Xu, 1995).

## 1.1. Related Work

The EM algorithm (Dempster et al., 1977) is a powerful tool for fitting latent variable models. Previous research on EM has demonstrated that, under mild smoothness assumptions, its parameter iterates converge to a stationary point of the log-likelihood objective (Dempster et al., 1977; Wu, 1983; Tseng, 2004). Subsequent research on EM introduced new analytical frameworks to provide specialized guarantees,

particularly regarding the convergence of EM iterates to the true model parameters and the rate of this convergence. An adopted framework in the past decade, introduced by Balakrishnan et al. (2017), interprets EM as a variant of the gradient descent algorithm. For latent variable models with a strongly convex EM objective that satisfies a condition known as "first-order stability," it was shown that EM iterates converge linearly to the true parameters. Subsequent works utilized this framework to show a local linear rate for Mixtures of Gaussians and Mixtures of Linear Regressions (Balakrishnan et al., 2017; Daskalakis et al., 2017; Dwivedi et al., 2018; Kwon et al., 2019; Kwon & Caramanis, 2020b; Kwon et al., 2021; Kwon & Caramanis, 2020a).

In a recent work, Kunstner et al. (2021) proved that EM – where the complete data distribution belongs in an exponential family – is equivalent to the mirror descent algorithm with unit step-size and Kullback–Leibler (KL) divergence regularizer:

$$\mathrm{KL}\left[p(\boldsymbol{x};\boldsymbol{\theta})||p(\boldsymbol{x},\boldsymbol{\phi})\right] = \mathbb{E}_{X|\boldsymbol{\theta}}\left[\log\left(\frac{p(\boldsymbol{x};\boldsymbol{\theta})}{p(\boldsymbol{x};\boldsymbol{\phi})}\right)\right]. \quad (1)$$

This led to the first non-asymptotic convergence rates for EM, independent of parameterization. While this characterization of EM included Mixtures of Gaussians, it failed to extend to Mixtures of Regression or MoE. Still, the authors analyzed the setting where the distribution of the latent variable conditioned on all other variables is an exponential family distribution for which they showed EM converged sub-linearly to a stationary point. Our work extends these findings to MoE, obtaining sufficient conditions under which EM converges sub-linearly and linearly to the true parameters.

There have also been works attempting to explore and understand how to fit MoE. The foundational paper (Jordan & Jacobs, 1994) was the first to empirically use EM and EM-variants to fit MoE, yielding encouraging results. Then, the follow-up paper by Jordan & Xu (1995) showed that for MoE and Hierarchical MoE with strongly convex negative log-likelihood objective, EM and EM-like iterations converge linearly to the true parameters where the rate constant depends on the eigenvalues of the Hessian matrix. However, the objective is generally non convex, raising doubts about whether the necessary assumptions for their result hold, even locally. Other works have remarked that the nature of the gating function creates a form of competition between the experts during training that can lead to local minima. Bayesian methods for MoE include variational learning and maximum aposteriori (MAP) estimation. But, Yuksel et al. (2012) noted that these solutions are not trivial due to the softmax gate not admitting a conjugate prior and are prone to getting stuck at local minima. Makkuva et al. (2019) analyzed a variant of the EM algorithm for MoE which consists in 1) first recovering the expert param-

eters using a tensor decomposition method, then 2) whilst freezing the experts, utilizing EM to fit the gating function's parameters only. They proved that their approach recovers the true parameters at a *nearly* linear rate. Then, Becker et al. (2020) experimented with an EM variant algorithm for Gaussian Mixture of Experts called Expectation Information Maximization (EIM) which featured an extra information projection step. They obtained promising empirical results on both synthetic and real world datasets. Our work extends upon previous works by making the direct connection between EM for MoE and projected mirror descent. We also unveil the sufficient conditions for EM to converge sub-linearly to a stationary point or true parameter, and for special cases, characterize these sufficient conditions with respect to the SNR of the generative model.

## 2. Mixture of Experts

Next, we formally describe the setting under consideration for the Mixture of $k$ Experts. The notation used throughout is summarized in the Notation Section of the appendix.

**Data Generation Model:** First, the input or feature vector variable $\boldsymbol{x} \in \mathbb{R}^d$ is sampled based on a probability density function $p(\boldsymbol{x})$. Second, given the feature vector $\boldsymbol{x}$, a latent variable $z \in [k]$, responsible for routing $\boldsymbol{x}$ to the appropriate expert is sampled with probability mass function $P(z|\boldsymbol{x})$. Finally, given the pair $(\boldsymbol{x}, z)$, the target $y$ is generated according to the probability distribution $p(y|\boldsymbol{x}, z)$. Hence, the complete data distribution is $p(\boldsymbol{x}, y, z) = p(y|\boldsymbol{x}, z)p(z|\boldsymbol{x})p(\boldsymbol{x})$ and the joint input-output probability distribution can be written as

$$p(\boldsymbol{x}, y) = p(\boldsymbol{x}) \sum_{z \in [k]} p(y|\boldsymbol{x}, z)P(z|\boldsymbol{x}). \qquad (2)$$

This is the most general form of data generation under mixture of $k$-experts, but here we focus on the case where $\boldsymbol{x}$ is sampled from a unit spherical Gaussian distribution, i.e. $\boldsymbol{x} \sim \mathcal{N}(\boldsymbol{0}, \boldsymbol{I}_d)$, and $P(z|\boldsymbol{x}) = P(z|\boldsymbol{x}; \boldsymbol{w}^*)$ is parameterized by a set of vectors $(\boldsymbol{w}_1^*, \ldots, \boldsymbol{w}_k^*)$ and can be cast as

$$P(z = i|\boldsymbol{x}; \boldsymbol{w}^*) = \frac{e^{\boldsymbol{x}^\top \boldsymbol{w}_i^*}}{\sum_{j \in [k]} e^{\boldsymbol{x}^\top \boldsymbol{w}_j^*}}, \qquad i \in [k]. \quad (3)$$

In other words, the probability mass function of the latent variable is the softmax function between the inner product of $\boldsymbol{x}$ with the parameter vectors concatenated as $\boldsymbol{w}^* = (\boldsymbol{w}_1^*, ..., \boldsymbol{w}_k^*) \in \mathbb{R}^{d \times k}$.

Regarding $p(y|\boldsymbol{x}, z)$, the probability distribution of the target variable $y$ conditioned on the input variable $x$ and latent routing variable $z = i$ (i.e., expert $i$), we also assume that it is parameterized by a vector $\boldsymbol{\beta}_i^*$ and we have $p(y|\boldsymbol{x}, z = i) = p(y|\boldsymbol{x}, z = i; \boldsymbol{\beta}_i^*)$. Thus, given the concatenated vector $\boldsymbol{\beta}^* = (\boldsymbol{\beta}_1^*, ..., \boldsymbol{\beta}_k^*) \in \mathbb{R}^{s \times k}$, we can write $p(y|\boldsymbol{x}, z) = p(y|\boldsymbol{x}, z; \boldsymbol{\beta}^*)$.

In this paper, we will consider three different settings. The first, *General MoE*, is the most general setting we consider. It comprises all MoE where the distribution of $y, z$ conditioned on $\boldsymbol{x}$ belongs to an exponential family of distribution and enjoys a natural re-parameterization $p(y, z|\boldsymbol{x}, \boldsymbol{\theta}^*) = p(y, z; \boldsymbol{\theta}_{\boldsymbol{x}}^*)$ with $\boldsymbol{\theta}_{\boldsymbol{x}}^* = \eta(\boldsymbol{x}, \boldsymbol{\theta}^*) \in \tilde{\Omega}$. That is to say that the conditional distribution can be written as

$$p(y, z|\boldsymbol{x}, \boldsymbol{\theta}^*) \propto \exp\left\{\langle s(y, z), \boldsymbol{\theta}_{\boldsymbol{x}}^*\rangle + A(\boldsymbol{\theta}_{\boldsymbol{x}}^*)\right\}, \quad (4)$$

where $s(y, z), \boldsymbol{\theta}_{\boldsymbol{x}}$, and $A(\cdot)$ are, respectively, referred to as the sufficient statistic, natural parameterization, and log partition of the exponential family of distributions. This setting includes the popular cases where $p(y|\boldsymbol{x}, z)$ is Gaussian or multivariate Bernoulli. In the former case, the exponential family of distribution corresponds to a Gaussian Mixture.

The second setting, *Mixture of Linear Experts*, is when

$$p(y|\boldsymbol{x}, z = i; \boldsymbol{\beta}_i^*) \propto \exp\left\{\frac{(y - \boldsymbol{x}^\top \boldsymbol{\beta}_i^*)^2}{2}\right\} \qquad (5)$$

as the density function of the normal distribution. This is equivalent to assume that the target $y = \boldsymbol{x}^\top \boldsymbol{\beta}_i^* + \epsilon$ where $\epsilon$ is additive zero mean Gaussian noise with unit variance.

Finally, *Mixture of Logistic Experts*, is when the density function $p(y|\boldsymbol{x}, z = i; \boldsymbol{\beta}_i^*)$ can be written as

$$P(y = 1|\boldsymbol{x}, z = i; \boldsymbol{\beta}_i^*) = \frac{\exp(\boldsymbol{x}^\top \boldsymbol{\beta}_i^*)}{1 + \exp(\boldsymbol{x}^\top \boldsymbol{\beta}_i^*)} \qquad (6)$$

and $P(y = -1|\boldsymbol{x}, z = i; \boldsymbol{\beta}_i^*) = 1 - P(y = 1|\boldsymbol{x}, z = i; \boldsymbol{\beta}_i^*)$.

**Maximum Likelihood Loss:** Given the assumed data distribution of $(\boldsymbol{x}, y)$, our goal is to find the set of feasible parameters $\boldsymbol{\beta} \in \mathbb{R}^{d \times k}$ and $\boldsymbol{w} \in \mathbb{R}^{d \times k}$ that maximize the log likelihood function. For ease of notation we define $\boldsymbol{\theta} := (\boldsymbol{\beta}, \boldsymbol{w}) \in \mathbb{R}^{2d \times k}$ as the concatenation of all parameters. Specifically, from (2), the expected log likelihood is

$$\mathbb{E}_{\boldsymbol{X}}\left[\log p(\boldsymbol{x})\right] + \mathbb{E}_{\boldsymbol{X}, Y}\left[\log\left(\sum_{z \in [k]} p(y|\boldsymbol{x}, z)P(z|\boldsymbol{x})\right)\right].$$

Given that only the last term of the sum depends on parameters $\boldsymbol{\theta} = (\boldsymbol{w}, \boldsymbol{\beta})^\top$ the negative log-likelihood objective function, $\mathcal{L}(\boldsymbol{\theta})$, that we aim to minimize can be written as

$$-\mathbb{E}_{\boldsymbol{X}, Y}\left[\log\left(\sum_{z \in [k]} p(y|\boldsymbol{x}, z; \boldsymbol{\beta})P(z|\boldsymbol{x}; \boldsymbol{w})\right)\right] \quad (7)$$

Note that as we will discuss in detail, for both mixtures of linear or logistic experts, the above objective function is known to be non convex with respect to $\boldsymbol{\theta}$. In fact, this is generally true for Mixtures of Gaussian, Mixtures of Regressions, and Mixtures of Experts. In the next section we will discuss the use of the EM algorithm for solving this optimization problem.

## 3. EM for Mixtures of Experts

Next, we present the EM algorithm for MoE. EM takes a structured approach to minimizing the objective $\mathcal{L}(\boldsymbol{\theta})$ in (7). Each iteration of EM is decomposed into two steps as follows. The first step is called "expectation": For current parameter estimate $\boldsymbol{\theta}^t$, we compute the expectation of the complete-data log-likelihood with respect to the latent variables, using the current parameter estimates $\boldsymbol{\theta}^t$ and denote it by $Q(\boldsymbol{\theta}|\boldsymbol{\theta}^t)$, i.e.,

$$Q(\boldsymbol{\theta}|\boldsymbol{\theta}^t) = -\mathbb{E}_{X,Y}\left[\mathbb{E}_{Z|\boldsymbol{x},y;\boldsymbol{\theta}^t}[\log p(\boldsymbol{x},y,z;\boldsymbol{\theta})]\right]. \quad (8)$$

Then, in the second step called "maximization", we simply minimize the objective $Q(\boldsymbol{\theta}|\boldsymbol{\theta}^t)$ (or maximize $-Q(\boldsymbol{\theta}|\boldsymbol{\theta}^t)$) with respect to $\boldsymbol{\theta} \in \Omega$ and obtain our new parameter as

$$\boldsymbol{\theta}^{t+1} := \underset{\boldsymbol{\theta}\in\Omega}{\operatorname{argmin}}\, Q(\boldsymbol{\theta}|\boldsymbol{\theta}^t). \quad (9)$$

Since $\log p(y,z|\boldsymbol{x};\boldsymbol{\theta}) = \log p(y|z,\boldsymbol{x};\boldsymbol{\beta}) + \log p(z|\boldsymbol{x};\boldsymbol{w})$, it follows that the EM objective (8) is linearly separable in the parameters $\boldsymbol{\beta}$ and $\boldsymbol{w}$. Thus, we can rewrite $Q(\boldsymbol{\theta}|\boldsymbol{\phi})$ as the sum of two functions that depend only on $\boldsymbol{\beta}$ and $\boldsymbol{w}$, respectively. Subsequently, the EM update (9) is obtained as the concatenation $\boldsymbol{\theta}^{t+1} = (\boldsymbol{w}^{t+1},\boldsymbol{\beta}^{t+1})^{\top}$, where

$$\boldsymbol{w}^{t+1} = \underset{\boldsymbol{w}\in\mathbb{R}^d}{\operatorname{argmin}} -\mathbb{E}_{\boldsymbol{X},Y}\left[\mathbb{E}_{Z|\boldsymbol{x},y;\boldsymbol{\theta}^t}[\log p(z|\boldsymbol{x};\boldsymbol{w})]\right],$$

$$\boldsymbol{\beta}^{t+1} = \underset{\boldsymbol{\beta}\in\mathbb{R}^d}{\operatorname{argmin}} -\mathbb{E}_{\boldsymbol{X},Y}\left[\mathbb{E}_{Z|\boldsymbol{x},y;\boldsymbol{\theta}^t}[\log p(y|z,\boldsymbol{x};\boldsymbol{\beta})]\right].$$

EM has two well understood characteristics: *(a)* its update always minimize an objective that is an upper bound on the likelihood, and *(b)* fixed points of the EM update are also stationary points of the likelihood. We now show this below. The original objective, $\mathcal{L}(\boldsymbol{\theta})$, can be decomposed into the difference between the EM objective and another function that is bounded below by 0. Specifically, for any $\boldsymbol{\theta}, \boldsymbol{\phi} \in \Omega$,

$$\mathcal{L}(\boldsymbol{\theta}) = -\mathbb{E}_{\boldsymbol{X},Y}\left[\log(p(y|\boldsymbol{x};\boldsymbol{\theta}))\right]$$
$$= -\mathbb{E}_{\boldsymbol{X},Y}\mathbb{E}_{Z|\boldsymbol{x},y,\boldsymbol{\phi}}\left[\log(p(y|\boldsymbol{x},z;\boldsymbol{\theta}))\right]$$
$$= -\mathbb{E}_{\boldsymbol{X},Y}\mathbb{E}_{Z|\boldsymbol{x},y,\boldsymbol{\phi}}\left[\log\left(\frac{p(y,z|\boldsymbol{x};\boldsymbol{\theta})}{p(z|y,\boldsymbol{x};\boldsymbol{\theta})}\right)\right].$$

Denoting $H(\boldsymbol{\theta}|\boldsymbol{\phi}) := -\mathbb{E}_{\boldsymbol{X},Y}\mathbb{E}_{Z|\boldsymbol{x},y,\boldsymbol{\phi}}[\log p(z|\boldsymbol{x},y;\boldsymbol{\theta})]$, it follows that

$$\mathcal{L}(\boldsymbol{\theta}) = Q(\boldsymbol{\theta}|\boldsymbol{\phi}) - H(\boldsymbol{\theta}|\boldsymbol{\phi}). \quad (10)$$

where $H(\boldsymbol{\theta}|\boldsymbol{\phi})$ is bounded below by 0. Thus, $Q(\boldsymbol{\theta}|\boldsymbol{\phi})$ acts as an upper bound on the negative log-likelihood.

Next, applying Jensen's inequality shows that $H(\boldsymbol{\theta}|\boldsymbol{\phi})$ is minimized at $\boldsymbol{\theta} = \boldsymbol{\phi}$ where $H(\boldsymbol{\theta}|\boldsymbol{\theta}) = 0$. Consequently, it follows that the negative log-likelihood gradient matches that of the surrogate EM objective at $\boldsymbol{\phi} = \boldsymbol{\theta}$, i.e., $\nabla\mathcal{L}(\boldsymbol{\theta}) = \nabla Q(\boldsymbol{\theta}|\boldsymbol{\theta})$. This suggests that any stationary point of $\mathcal{L}(\boldsymbol{\theta})$ is also a stationary point of the EM algorithm and vice versa.

## 3.1. EM for Symmetric Mixture of 2-Experts

So far, we discussed EM for the most general form of MoE. Next, we derive EM for the special case of *Symmetric Mixture of Experts* (SymMoE), the focus of our analysis in Section 5. SymMoE is a simplified version of MoE where (i) the number of experts is restricted to 2, represented as $z \in \{-1, 1\}$, and (ii) the experts are symmetric around the linear separator, i.e., $\tilde{\boldsymbol{\beta}}^* := \boldsymbol{\beta}_1^* = -\boldsymbol{\beta}_{-1}^*$. This symmetric structure simplifies the probability density functions introduced earlier, making the subsequent analysis easier to follow. We explore these simplifications in detail below.

As we are restricted to two experts, the expression for $P(z = 1|\boldsymbol{x}; \boldsymbol{w}^*) = 1 - P(z = -1|\boldsymbol{x}; \boldsymbol{w}^*)$ is

$$P(z = 1|\boldsymbol{x}; \boldsymbol{w}^*) = \frac{e^{\boldsymbol{x}^\top \boldsymbol{w}_1^*}}{e^{\boldsymbol{x}^\top \boldsymbol{w}_1^*} + e^{\boldsymbol{x}^\top \boldsymbol{w}_2^*}}.$$

For ease of notation, we define $\tilde{\boldsymbol{w}}^* := \boldsymbol{w}_1^* - \boldsymbol{w}_2^*$ and reparameterize the probability mass function of $z$ given $\boldsymbol{x}$ as

$$P(z|\boldsymbol{x}; \tilde{\boldsymbol{w}}^*) = \frac{\exp\{\frac{z+1}{2}\boldsymbol{x}^\top \tilde{\boldsymbol{w}}^*\}}{1 + e^{\boldsymbol{x}^\top \tilde{\boldsymbol{w}}^*}}. \quad (11)$$

Thus, under this simplification, the EM update of the gating parameter $\boldsymbol{w}$ is now given as

$$\boldsymbol{w}^{t+1} = \underset{\boldsymbol{w}\in\mathbb{R}^d}{\operatorname{argmin}}\, \mathbb{E}_{\boldsymbol{X},Y}\mathbb{E}_{Z|\boldsymbol{x},y;\boldsymbol{\theta}^t}\left[\log\left(\frac{1 + e^{\boldsymbol{x}^\top \boldsymbol{w}}}{e^{\frac{z+1}{2}\boldsymbol{x}^\top \boldsymbol{w}}}\right)\right].$$

While the above minimization problem is strongly convex, it does not have a closed form solution. In our experiments, we use gradient descent to obtain $\boldsymbol{w}^{t+1}$.

For the special case of a *symmetric mixture of linear experts* (SymMoLinE), the expression for $p(y|\boldsymbol{x}, z; \boldsymbol{\beta}^*)$ given in (5) can be simplified as

$$p(y|\boldsymbol{x}, z; \boldsymbol{\beta}^*) \propto \exp\left\{\frac{(y - z\boldsymbol{x}^\top \tilde{\boldsymbol{\beta}}^*)^2}{2}\right\}. \quad (12)$$

Under this simplification, the EM update of the expert parameter $\boldsymbol{\beta}$ is now obtained more compactly as

$$\boldsymbol{\beta}^{t+1} = \mathbb{E}_{X,Y}\left[\left(2p(z = 1|\boldsymbol{x}, y; \boldsymbol{\theta}^t) - 1\right)\boldsymbol{x}y\right].$$

Similarly, for *symmetric mixtures of logistic experts* (SymMoLogE) with $y \in \{-1, 1\}$, the expression of $p(y|\boldsymbol{x}, z; \boldsymbol{\beta}^*)$ given in (6) can also be simplified as

$$P(y|\boldsymbol{x}, z; \boldsymbol{\beta}^*) = \frac{\exp\{\left(\frac{yz+1}{2}\right)\boldsymbol{x}^\top \boldsymbol{\beta}^*\}}{1 + e^{\boldsymbol{x}^\top \boldsymbol{\beta}^*}}. \quad (13)$$

Under this simplification, the EM update of the expert parameter $\boldsymbol{\beta}$ is now given as

$$\boldsymbol{\beta}^{t+1} = \underset{\boldsymbol{\beta}\in\mathbb{R}^d}{\operatorname{argmin}}\, \mathbb{E}_{\boldsymbol{X},Y}\left[\mathbb{E}_{Z|\boldsymbol{x},y,\boldsymbol{\theta}^t}\left[\log\left(\frac{1 + e^{\boldsymbol{x}^\top \boldsymbol{\beta}}}{e^{\frac{yz+1}{2}\boldsymbol{x}^\top \boldsymbol{\beta}}}\right)\right]\right].$$

## 3.2. EM for Deep and Sparse MoE

So far, we have derived EM for the foundational formulations of MoE, as initially proposed in (Jacobs et al., 1991). While EM is straightforward to derive in these cases, the same does not hold for deep MoE—and especially not for Sparse MoE. A deep MoE consists of $l \geq 2$ MoE blocks, as defined in Section 2, stacked sequentially. In this setup, the input $x$ passes through the first MoE block and then sequentially through all subsequent blocks to produce the output $y$.

For completeness, and to encourage future work on large-scale applications of EM for MoE, we propose a formulation of EM for deep and sparse MoE in Appendix D.4. Instead of solving a separate latent-variable problem at each layer—as is done in classical MoE—we posit that the latent variable $z \in [k]^l$ should represent the entire sequence of experts selected across the network. This allows us to construct the EM objective in Equation (8).

An EM-like solution can then be derived for Sparse MoE, where the loss is computed solely from the sequences of experts observed through greedy expert selection. This formulation provides a principled approach to training deep and sparse MoE models using EM.

## 4. Main Result

In the previous section, we derived the EM update for both MoE and SymMoE as the solution to minimizing the EM objective in (8). We further demonstrated that this solution can be decomposed into the concatenation of the respective solutions to two minimization sub-problems. In this section, we will show that this update is exactly equivalent to performing a single step of the projected Mirror Descent (MD) update on $\mathcal{L}(\boldsymbol{\theta})$ with a unit step size and the KL divergence as a regularizer. To illustrate more clearly that minimizing $Q(\boldsymbol{\theta}|\boldsymbol{\theta}^t)$ in (8) corresponds to a projected MD step on the loss $\mathcal{L}(\boldsymbol{\theta})$ at the point $\boldsymbol{\theta}^t$, we first provide a brief overview of the core concept behind the MD update.

In most gradient-based methods, the next iteration is obtained by minimizing an upper bound of the objective function. For example, in Gradient Descent (GD), the next iterate is found by minimizing the first-order Taylor expansion of the objective at $\boldsymbol{\theta}^t$, with a squared norm regularizer. Specifically, for minimizing $\mathcal{L}(.)$, the GD update with step size $\eta$ at $\boldsymbol{\theta}^t$ is equivalent to minimizing the following function:

$$\mathcal{L}(\boldsymbol{\theta}^t) + \langle \nabla \mathcal{L}(\boldsymbol{\theta}^t), \boldsymbol{\theta} - \boldsymbol{\theta}^t \rangle + \frac{1}{2\eta} \|\boldsymbol{\theta} - \boldsymbol{\theta}^t\|_2^2.$$

This function indeed serves as an upper bound for $\mathcal{L}(.)$ if $\eta \leq 1/L$, where $L$ is the Lipschitz constant of $\nabla \mathcal{L}(\boldsymbol{\theta})$.

*Mirror Descent* solves a similar sub-problem where instead of a squared norm regularizer, we employ the *Bregman Di-*

*vergence* regularizer: The Bregman divergence induced by a differentiable, convex function $h : \mathbb{R}^d \to \mathbb{R}$, measures the difference between $h(\boldsymbol{\theta})$ and its first-order approximation at $\boldsymbol{\theta}^t$, i.e.,

$$D_h(\boldsymbol{\theta}, \boldsymbol{\theta}^t) = h(\boldsymbol{\theta}) - h(\boldsymbol{\theta}^t) - \langle \nabla h(\boldsymbol{\theta}^t), \boldsymbol{\theta} - \boldsymbol{\theta}^t \rangle. \quad (14)$$

Thus, the iterations of MD are derived by minimizing the following expression:

$$\mathcal{L}(\boldsymbol{\theta}^t) + \langle \nabla \mathcal{L}(\boldsymbol{\theta}^t), \boldsymbol{\theta} - \boldsymbol{\theta}^t \rangle + \frac{1}{\eta} D_h(\boldsymbol{\theta}^t, \boldsymbol{\theta}). \quad (15)$$

Finally, in projected mirror descent, the update is completed by projecting the solution obtained from minimizing (15) onto a subspace. As mentioned, this scheme is reasonable if the function approximation using the Bregman divergence serves as an upper bound for the function $\mathcal{L}(\boldsymbol{\theta})$. This can be ensured when the step size $\eta$ is sufficiently small, and the condition of relative smoothness is satisfied (Lu et al., 2018).

In the upcoming theorem, we formally establish that for general MoE, minimizing the EM objective function defined in equation (9) is exactly equivalent to minimizing the subproblem associated with a single step of MD, as defined in equation (15), followed by a projection step. This result demonstrates that the EM update for General MoE is essentially performing an MD step in a specific natural re-parameterization space, then projecting the resulting solution onto $\Omega$. The proof of this result is involved and is deferred to Appendix A.1.

**Theorem 4.1.** *For General MoE, there exists a natural reparameterization $\boldsymbol{\theta_x} \in \{\eta(\cdot, \boldsymbol{\theta}) : \theta \in \Omega\}$ with*

$$\mathcal{L}(\boldsymbol{\theta}) = \mathbb{E}_X [L(\boldsymbol{\theta_x})] \quad (16)$$

*and a mirror map $A(\boldsymbol{\theta_x})$ such that the EM update in (9) simplifies and is equivalent to the expectation moment projection,*

$$\underset{\boldsymbol{\theta} \in \Omega}{\operatorname{argmin}} \, \mathbb{E}_X \left[ KL \left[ p\left(y, z \middle| \tilde{\theta}_{\boldsymbol{x}}^{t+1}\right) \middle\| p\left(y, z | \eta(\boldsymbol{x}, \boldsymbol{\theta})\right) \right] \right], \quad (17)$$

*where for each $\boldsymbol{x}$, $\tilde{\boldsymbol{\theta}}_{\boldsymbol{x}}^{t+1}$ is obtained from the following MD step,*

$$\underset{\boldsymbol{\psi} \in \tilde{\Omega}}{\operatorname{argmin}} \langle \nabla L(\boldsymbol{\theta}_{\boldsymbol{x}}^t), \boldsymbol{\psi} - \boldsymbol{\theta}_{\boldsymbol{x}}^t \rangle + D_A(\boldsymbol{\psi}, \boldsymbol{\theta}_{\boldsymbol{x}}^t), \quad (18)$$

*with $L(\boldsymbol{\theta_x})$ being 1-smooth relative to $A(\boldsymbol{\theta_x})$. Further, $\forall \boldsymbol{\psi}_1, \boldsymbol{\psi}_2 \in \tilde{\Omega}$, the divergence function $D_A(\boldsymbol{\psi}_1, \boldsymbol{\psi}_2)$ is equal to the $KL$ divergence on $p(y, z, |\boldsymbol{\psi})$:*

$$D_A(\boldsymbol{\psi}_1, \boldsymbol{\psi}_2) = KL[p(y, z | \boldsymbol{\psi}_2) \| p(y, z | \boldsymbol{\psi}_1)]. \quad (19)$$

*Proof sketch.* We utilize the assumed property that the conditional distribution $p(y, z | \boldsymbol{x}, \boldsymbol{\theta})$ belongs to an exponential

family of distributions to decompose the EM surrogate as

$$Q(\boldsymbol{\theta}|\boldsymbol{\theta}^t) - Q(\boldsymbol{\theta}^t|\boldsymbol{\theta}^t) =$$
$$\mathbb{E}_X\left[\langle\nabla L(\boldsymbol{\theta}_{\boldsymbol{x}}^t), \boldsymbol{\theta}_{\boldsymbol{x}} - \boldsymbol{\theta}_{\boldsymbol{x}}^t\rangle + D_A(\boldsymbol{\theta}_{\boldsymbol{x}}, \boldsymbol{\theta}_{\boldsymbol{x}}^t)\right]. \quad (20)$$

The above derivation follows similar steps to Kunstner et al. Theorem 1 and is provided in Appendix A for completeness. We note that because $\theta_x = \eta(x, \theta)$ is not necessarily linear in $x$, we cannot conclude that minimizing (1) with respect to $\theta$ results in a direct MD step. Instead, recall the point-wise (in $x$) MD iterate, $\tilde{\theta}_{\boldsymbol{x}}^{t+1} := \operatorname{argmin}_{\boldsymbol{\theta}_{\boldsymbol{x}}}\langle\nabla L(\boldsymbol{\theta}_{\boldsymbol{x}}^t), \boldsymbol{\theta}_{\boldsymbol{x}} - \boldsymbol{\theta}_{\boldsymbol{x}}^t\rangle + D_A(\boldsymbol{\theta}_{\boldsymbol{x}}, \boldsymbol{\theta}_{\boldsymbol{x}}^t)$. Differentiating and setting equal to 0, it holds that

$$\nabla A(\tilde{\theta}_{\boldsymbol{x}}^{t+1}) = s(\boldsymbol{x}; \boldsymbol{\theta}^t). \quad (21)$$

Using (2) and the decomposing $\nabla L(\boldsymbol{\theta}_{\boldsymbol{x}}^t) = \nabla A(\boldsymbol{\theta}_{\boldsymbol{x}}^t) - s(\boldsymbol{x}; \boldsymbol{\theta}^t)$, we see that (1) can be further simplified to

$$Q(\boldsymbol{\theta}|\boldsymbol{\theta}^t) - Q(\boldsymbol{\theta}^t|\boldsymbol{\theta}^t) =$$
$$\mathbb{E}_X\left[-\langle\nabla A(\tilde{\theta}_{\boldsymbol{x}}^{t+1}), \boldsymbol{\theta}_{\boldsymbol{x}} - \boldsymbol{\theta}_{\boldsymbol{x}}^t\rangle + A(\boldsymbol{\theta}_{\boldsymbol{x}}) - A(\boldsymbol{\theta}_{\boldsymbol{x}}^t)\right].$$

Because $A(\tilde{\theta}_{\boldsymbol{x}}^{t+1})$ and $A(\boldsymbol{\theta}_{\boldsymbol{x}}^t)$ only depend on $\theta^t$, minimizing the above with respect to $\boldsymbol{\theta}$ is equivalent to minimizing the following with respect to $\theta$

$$\mathbb{E}_X\left[D_A(\boldsymbol{\theta}_{\boldsymbol{x}}, \tilde{\boldsymbol{\theta}}_{\boldsymbol{x}}^{t+1})\right]. \quad (22)$$

Finally, substituting the Bregman Divergence induced by $A$ by the KL divergence yields the claim (this derivation is also included in Appendix A.1 for completeness). □

We remark that our proof for this result is not merely limited to the setting of MoE, but is satisfied for any mixture for which the distribution $p(y, z|\boldsymbol{x}, \boldsymbol{\theta})$ of the target and latent variable conditioned on the feature variable belongs to an exponential family of distribution.

### 4.1. Convergence Guarantees for General MoE

Before moving into the convergence results, we specify the necessary assumptions previously outlined in (Kunstner et al., 2021) for the iterations of the EM Algorithm to be well-defined for our class of MoE models. See Appendix A.2 for a more in-depth discussion of the implications of these assumptions.

$A_1$. The conditional distribution $p(y, z|\boldsymbol{\theta}_{\boldsymbol{x}})$ is a steep, minimal exponential family of distribution and $\eta(\boldsymbol{x}, \cdot)$ is a continuously differentiable function.

$A_2$. The optimal objective function value is bounded below, i.e., $\mathcal{L}(\boldsymbol{\theta}^*) > -\infty$, on the constraint set $\Omega$.

$A_3$. The following sub-level sets $\Omega_\theta := \{\phi \in \Omega : Q(\phi|\boldsymbol{\theta}) \le Q(\boldsymbol{\theta}|\boldsymbol{\theta})\}$ are compact.

Next, we briefly introduce key definitions that will be used later. We say $\boldsymbol{\theta}^1$ is initialized in a locally average-convex region of $\mathcal{L}(\boldsymbol{\theta})$ with respect to the random variable $X$, if there exists a convex set $\Theta \subseteq \Omega$ containing $\boldsymbol{\theta}^1, \boldsymbol{\theta}^*$ such that for all $\phi, \boldsymbol{\theta} \in \Theta$,

$$\mathbb{E}_X[L(\boldsymbol{\phi}_{\boldsymbol{x}})] \ge \mathbb{E}_X[L(\boldsymbol{\theta}_{\boldsymbol{x}}) + \langle\nabla L(\boldsymbol{\theta}_{\boldsymbol{x}}), \boldsymbol{\phi}_{\boldsymbol{x}} - \boldsymbol{\theta}_{\boldsymbol{x}}\rangle] \quad (23)$$

where $\boldsymbol{\theta}_{\boldsymbol{x}} := \eta(\boldsymbol{x}, \boldsymbol{\theta})$. Furthermore, $\Theta$ is called $\alpha$-average-convex relative to $h$ if

$$\mathbb{E}_X[L(\boldsymbol{\phi}_{\boldsymbol{x}})] \ge \mathbb{E}_X[L(\boldsymbol{\theta}_{\boldsymbol{x}})]$$
$$+ \mathbb{E}_X[\langle\nabla L(\boldsymbol{\theta}_{\boldsymbol{x}}), \boldsymbol{\phi}_{\boldsymbol{x}} - \boldsymbol{\theta}_{\boldsymbol{x}}\rangle + \alpha D_h(\boldsymbol{\phi}_{\boldsymbol{x}}, \boldsymbol{\theta}_{\boldsymbol{x}})] \quad (24)$$

Now, thanks to the previously shown correspondence between EM for general MoE and projected MD, we are able to present novel convergence properties of the EM Algorithm when applied to General MoE. The theorem that follows provides sufficient conditions and explicit rates for the convergence of EM iterations to a stationary point or true parameters. The proof, adapted from (Lu et al., 2018), is a bit more involved due to the nature of the extra projection step. It is included in Appendix A.2.

**Theorem 4.2** (Convergence of EM). *Assuming $A_1$ - $A_3$. For general MoE with re-parameterization given by $\boldsymbol{\theta}_{\boldsymbol{x}} := \eta(\boldsymbol{x}, \boldsymbol{\theta})$, strictly convex mirror map $A(\boldsymbol{\theta}_{\boldsymbol{x}})$, and if for all $\tilde{\boldsymbol{\theta}}_{\boldsymbol{x}}^{t+1}, \boldsymbol{\theta}_{\boldsymbol{x}}^{t+1}, \boldsymbol{\phi}_{\boldsymbol{x}} \in \{\boldsymbol{\theta}_{\boldsymbol{x}}^t, \boldsymbol{\theta}_{\boldsymbol{x}}^*\}$,*

$$\mathbb{E}_X\left[D_A\left(\boldsymbol{\phi}_{\boldsymbol{x}}, \tilde{\boldsymbol{\theta}}_{\boldsymbol{x}}^{t+1}\right)\right] \ge$$
$$\mathbb{E}_X\left[D_A\left(\boldsymbol{\theta}_{\boldsymbol{x}}^{t+1}, \tilde{\boldsymbol{\theta}}_{\boldsymbol{x}}^{t+1}\right) + D_A\left(\boldsymbol{\phi}_{\boldsymbol{x}}, \boldsymbol{\theta}_{\boldsymbol{x}}^{t+1}\right)\right], \quad (25)$$

*then, the EM iterates $\{\boldsymbol{\theta}^t\}_{t\in[T]}$ satisfy:*

1) *Stationnarity. For no additional conditions,*

$$\min_{t\in[T]} \mathbb{E}_X\left[D_A(\boldsymbol{\theta}_{\boldsymbol{x}}^t, \boldsymbol{\theta}_{\boldsymbol{x}}^{t+1})\right] \le \frac{\mathcal{L}(\boldsymbol{\theta}^1) - \mathcal{L}(\boldsymbol{\theta}^*)}{T}; \quad (26)$$

2) *Sub-linear Rate to $\boldsymbol{\theta}^*$. If $\boldsymbol{\theta}^1$ is initialized in $\Theta$, a locally average-convex region of $\mathcal{L}(\boldsymbol{\theta})$ containing $\boldsymbol{\theta}^*$, then*

$$\mathcal{L}(\boldsymbol{\theta}^T) - \mathcal{L}(\boldsymbol{\theta}^*) \le \frac{\mathbb{E}_X\left[D_A(\boldsymbol{\theta}_{\boldsymbol{x}}^*, \boldsymbol{\theta}_{\boldsymbol{x}}^1)\right]}{T} \quad (27)$$

3) *Linear Rate to $\boldsymbol{\theta}^*$. If $\boldsymbol{\theta}^1$ is initialized in $\Theta \subseteq \Omega$, a locally average-convex region of $\mathcal{L}(\boldsymbol{\theta})$ relative to $A(\boldsymbol{\theta})$ that contains $\boldsymbol{\theta}^*$, then*

$$\mathcal{L}(\boldsymbol{\theta}^T) - \mathcal{L}(\boldsymbol{\theta}^*) \le (1-\alpha)^T \mathbb{E}_X\left[D_A(\boldsymbol{\theta}_{\boldsymbol{x}}^*, \boldsymbol{\theta}_{\boldsymbol{x}}^1)\right] \quad (28)$$

The above condition on the initialization to belong in a locally average-convex region is satisfied trivially if $L(\boldsymbol{\theta}_{\boldsymbol{x}})$ is almost surely convex relative to $A(\boldsymbol{\theta}_{\boldsymbol{x}})$. Such an assumption is stronger, but more in line with standard sufficient conditions for optimality of MD.

As noted in the literature, EM's convergence is sensitive to initialization. If $\boldsymbol{\theta}^1$ is initialized within a locally average-convex region of $\mathcal{L}(\boldsymbol{\theta})$, the EM iterates for the MoE problem will converge sub-linearly to the true parameter. However, if $\boldsymbol{\theta}^1$ is in a region where $\mathcal{L}(\boldsymbol{\theta})$ is strongly average-convex relative to $A$, the iterates will converge linearly. This last assumption is different from that of prior work, which typically require $\boldsymbol{\theta}^1$ to be initialized in a locally strongly convex region.

### 4.2. Discussion of Main Result

The results show that the EM update (9) for general MoE is equivalent to projected mirror descent with a unit step-size and KL divergence regularizer on the complete data distribution. We offer the following additional remarks.

First, if we have oracle access to the EM updates for $\boldsymbol{w}$ and $\boldsymbol{\beta}$, EM requires no hyper-parameters, unlike GD, which is sensitive to the step size. This can be especially advantageous for cases where the $\boldsymbol{\beta}$-update has a closed-form solution (as is the case for linear experts), making EM's benefits over GD more evident. Additionally, while GD regularizes progress based on the Euclidean distance between iterates, EM adjusts progress based on the divergence between probability distributions across iterations. This is often more suitable for latent variable models, where small Euclidean changes may cause large shifts in the mixture distribution, and vice versa.

Second, whereas previous analysis of EM for various settings hinged on various types of analyses ranging from verifying obscure conditions to – less reproducible at scale – direct proofs, the connection to MD that we unveil greatly unifies the process of analysis and provides more intuition as to the inner workings of EM. In particular, as we will discuss in Section 5, our framework for analysis allows to easily provide intuitive conditions for linear convergence to the true parameters that are based on the MIM and subsequently, the SNR of the generative model.

Third, while Jordan & Xu (1995) also demonstrated that the EM algorithm for MoE converges linearly to the true parameters, the sufficient conditions they provided are more restrictive. Specifically, their analysis requires the Hessian to be negative definite, and the convergence rate depends explicitly on its eigenvalues. These conditions are similar in nature to those typically required for GD-type methods. In contrast, our sufficient conditions for optimality align with those of MD, which more accurately captures the convergence behavior of EM, as established by the equivalence shown in Theorem 4.1.

Finally, large-scale applications often favor a mini-batch training paradigm, as it tends to yield better performance for a given computational cost. Large-scale implementation

of EM can directly benefit from this paradigm for solving each iteration's convex optimization subproblem (i.e., Equation (9)) where a GD-style method is typically used. Scaling laws for the mini-batch paradigm suggest that reducing the batch size should be accompanied by a proportional reduction in the learning rate (Shuai et al., 2024; Malladi et al., 2022; Goyal et al., 2017). However, since EM is equivalent to MD with a fixed learning rate of 1, this kind of modular tuning is not directly applicable to EM. This does not imply that there is no optimal batch size for EM. Rather, extending theoretical guarantees to stochastic and mini-batch settings can be approached through the framework of stochastic mirror descent (SMD) and mini-batch MD (MBMD), both of which have been studied in the context of composite optimization (Duchi et al., 2010). We highlight this as an open direction for future research on scalable implementations of EM for MoE.

## 5. Special case of SymMoLinE and SymMoLogE

In this section, we narrow in on two special cases: the Symmetric Mixture of Linear Experts (SymMoLinE) and the Symmetric Mixture of Logistic Experts (SymMoLogE) models. We first show that minimizing the EM objective function defined in (9) is exactly equivalent to minimizing the subproblem associated with a single step of MD as defined in equation (15), with a step size of $\eta = 1$. This time, we fully specify the mirror map and show it is strictly convex. Unlike the previous result, this correspondence between EM and MD does not feature the extra projection step that was present for general MoE. This allows us to more easily characterize the sufficient condition of EM for linear convergence to the true parameters, relating it to the MIM and SNR of the generative model (see Appendix B.5 and B.6). The proof is provided in Appendix B.2.

**Theorem 5.1.** *For SymMoLinE and SymMoLogE, there is a mirror map $A(\boldsymbol{\theta})$ such that the EM update in (9) is equivalent to*

$$\operatorname*{argmin}_{\boldsymbol{\theta} \in \Omega} \langle \nabla \mathcal{L}(\boldsymbol{\theta}^t), \boldsymbol{\theta} - \boldsymbol{\theta}^t \rangle + D_A(\boldsymbol{\theta}, \boldsymbol{\theta}^t), \quad (29)$$

*where, $\forall \boldsymbol{\phi}, \boldsymbol{\theta} \in \Omega$, the divergence function $D_A(\boldsymbol{\theta}, \boldsymbol{\theta}^t)$ is equal to the $KL$ divergence on the complete data:*

$$D_A(\boldsymbol{\phi}, \boldsymbol{\theta}) = KL[p(\boldsymbol{x}, y, z; \boldsymbol{\theta}) \| p(\boldsymbol{x}, y, z; \boldsymbol{\phi})]. \quad (30)$$

*In particular, in the case of SymMoLinE,*

$$A(\boldsymbol{\theta}) = \mathbb{E}_{\boldsymbol{X}} \left[ \frac{(\boldsymbol{x}^\top \boldsymbol{\beta})^2}{2} + \log \left( 1 + e^{\boldsymbol{x}^\top \boldsymbol{w}} \right) \right], \quad (31)$$

*while in the case of SymMoLogE,*

$$A(\boldsymbol{\theta}) = \mathbb{E}_{\boldsymbol{X}} \left[ \log \left( \left( 1 + e^{\boldsymbol{x}^\top \boldsymbol{\beta}} \right) \left( 1 + e^{\boldsymbol{x}^\top \boldsymbol{w}} \right) \right) \right]. \quad (32)$$

*Finally, in both cases, the map $A(\boldsymbol{\theta})$ is strictly convex in $\boldsymbol{\theta}$ and $\mathcal{L}(\boldsymbol{\theta})$ is 1-smooth relative to $A(\boldsymbol{\theta})$.*

As shown in the proof of the result, it is evident from (41) and (45) that $p(\boldsymbol{x}, y, z; \boldsymbol{\theta})$ does not belong to an exponential family of distributions for either SymMoLinE or SymMoLogE. Therefore, this result does not simply follow as a corollary of (Kunstner et al., 2021, Proposition 1), but stands as an independent finding, introducing another class of latent variable models where EM is equivalent to MD. A couple of follow-up remarks are in order.

First, as the loss is 1-smooth relative to $A$, this validates the choice of $\eta = 1$ for the Mirror Descent update, and subsequent convergence results from the MD literature. Specifically, in Corollary B.1 of Appendix B.4, we contextualize convergence results from (Lu et al., 2018; Kunstner et al., 2021) for SymMoLinE and SymMoLogE that feature 1) a guaranteed sub-linear rate of convergence to a stationary point at no additional assumption, 2) a sub-linear rate of convergence to the true parameter if initialized within a convex region of the loss-function that includes the true parameters, and 3) a linear rate of convergence to the true parameter if initialized within a region of the loss function that contains the true parameters and is strongly-convex relative to the mirror map. Then, in Theorem B.2 of Appendix B.5, we further characterize the assumptions required for linear convergence by relating the relative strong convexity of the objective to the eigenvalues of the MIM. Lastly, in Theorem B.4 of Appendix B.6, we characterize the existence of the local region of convergence as a function of the SNR of the generative model for the cases of SymMoLinE and SymMoLogE. We then conclude in Appendix B.7 with a discussion of the implications of the results.

## 6. Experiments

In this section, we empirically validate our theoretical results by comparing the performance of EM with Gradient EM (Algorithm 2), and Gradient Descent (GD, Algorithm 3). Recall that EM for MoE obtains its next parameter iterate as the concatenation to the solutions of two minimization problems. Instead, Gradient EM obtains its next parameter iterate as the concatenation of a single gradient update on the respective sub-minimization problems of EM. This differs from GD that obtains its next parameter iterate as the gradient update on the negative log-likelihood objective $\boldsymbol{\theta}$. We evaluate these methods on both a synthetic dataset and the real-world Fashion MNIST dataset, consistently reporting significant improvements for EM and Gradient EM over GD. We also provide mini-batch CIFAR-10 experiments with more than 2-experts in Appendix C.1. Note that our aim is not to achieve state-of-the-art accuracy, but to reiterate that EM can be more suitable than GD for fitting specific models.

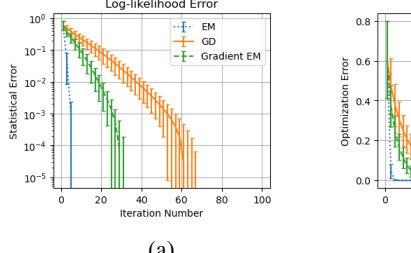

(a)                    (b)

*Figure 1.* Convergence of objective errors $\mathcal{L}(\boldsymbol{\theta}^t) - \mathcal{L}(\boldsymbol{\theta}^*)$ and $\mathcal{L}(\boldsymbol{\theta}^t) - \mathcal{L}(\boldsymbol{\theta}^T)$ in Fig 1a and Fig 1b, respectively, averaged over 50 instances when fitting a SymMoLinE.

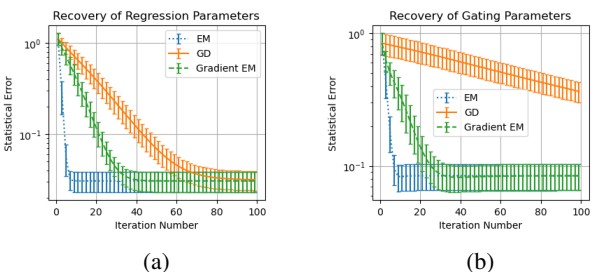

(a)                    (b)

*Figure 2.* This figure shows the progress made towards the true parameters, $\frac{\|\boldsymbol{\beta}^t - \boldsymbol{\beta}^*\|_2}{\|\boldsymbol{\beta}^*\|_2}$ and $\frac{\|\boldsymbol{w}^t - \boldsymbol{w}^*\|_2}{\|\boldsymbol{w}^*\|_2}$ in figures 2a and 2b respectively, averaged over 50 instances when fitting a SymMoLinE

**Synthetic Dataset.** We created the synthetic dataset so as to simulate a population setting of SymMoLinE. We sampled $10^3$ data points from an SymMoLinE with known additive unit Gaussian noise (i.e. $\mathcal{N}(0, 1)$) and true parameters $\boldsymbol{\beta}^*, \boldsymbol{w}^* \in \mathbb{R}^{10}$ that sastisfy $\|\boldsymbol{\beta}^*\|_2 = \|\boldsymbol{w}^*\|_2 = 4$. Subsequently, we run full-batch EM, Gradient EM, and GD for 50 iterations and report the results on the training set averaged over 50 instances. Each time, re-sampling the true parameters, initial parameters, and whole dataset. The initial parameters, are randomly initialize within a neighborhood of the true parameters, and are consistent across all benchmarks.

Figure 1 shows the objective function progress. EM requires fewer iterations to fit the mixture compared to both Gradient EM and GD, with Gradient EM also outperforming GD in fitting time. Figure 2 illustrates the progress toward recovering the true SymMoLinE parameters. Once again, EM requires significantly fewer iterations to fit the mixture compared to both Gradient EM and GD, with Gradient EM also taking considerably less time than GD.

Overall, we observe that all three algorithms exhibit a linear convergence rate, both in optimizing the objective function and fitting the true parameters. This aligns with our theoretical results for MoE and is consistent with findings for Mixtures of Gaussians and Mixtures of Linear Regression

in high SNR scenarios. To validate our results further, we perform a paired t-test (Ross & Willson, 2018). For EM and Gradient EM compared to GD, we obtain a T-statistic $\geq 22$ indicating that the difference in final accuracy is statistically significant (p-value $\sim 0.000$).

**Validation Experiment on Fashion MNIST.** For the small scale proof of concept experiment on Fashion MNIST (Xiao et al., 2017), we alter the dataset so as to simulate a mixture of 2 Logistic Experts. To do so, we perform an inversion transformation on the images at random with probability $\frac{1}{2}$. Effectively, the transformation inverts the images from a white article of clothing on a black background to a black article of clothing on a white background. As shown in Table 1, the single expert on the original Fashion MNIST dataset reaches an accuracy of $83.2\%$ on the test set. Meanwhile, the single expert cannot achieve better than an accuracy of $10.2\%$ on the altered dataset. This suggests a 2-component MoLogE is appropriate for fitting the altered dataset, so long as the ground truth partitioning is linear in image space.

The 2-component MoLogE to be trained consists of one Linear gating layer of dimension $2 \times 28 \times 28$, and 2 logistic experts of dimension $10 \times 28 \times 28$ each. We randomly initialize each linear layer to be unit norm and execute the algorithms on the same datasets and with the same initializations. For Gradient EM, the only additional code needed over GD is to define the EM Loss function appropriately, and then perform a Gradient Step on the Gating parameters and the Expert parameters separately as describe in Algorithm 2. For EM, for each iteration, we perform several gradient steps in an inner loop to approximately recover the solutions to the sub-problems described in (9). We report our findings for the full-batch iteration of the respective algorithms in Table 2 and Figure 3.

In Table 2, we report the respective final test accuracy and cross-entropy loss values after 100 iterations of EM, Gradient EM and GD for fitting a 2-component MoLogE on the altered Fashion MNIST dataset, averaged over 25 instances. We see that EM boasts a much improved final test accuracy that nearly recovers the single expert accuracy on the original unaltered Fashion MNIST dataset of $79.2\%$. Meanwhile, Gradient EM also registers an improvement over GD. In Figure 3, we report the progress made on the accuracy and objective function for the test set over the 100 iterations, averaged over 25 instances. As was observed in our synthetic experiment, EM takes considerably less iterations to fit the mixture than both Gradient EM and GD, where the former also takes considerably less time to fit the mixture than GD. To validate our results further, we perform a paired t-test. For EM and Gradient EM compared to GD, we obtain a T-statistic $\geq 17$ indicating that the difference in final accuracy is statistically significant (p-value $\sim 0.000$).

*Table 1.* Performance for single Logistic Expert

|  | Accuracy | Random Invert |
|---|---|---|
| **Single Expert** | *83.2%* | *No* |
| **Single Expert** | 10.2% | Yes |

*Table 2.* Performance for 2-Component MoLogE

|  | Accuracy | Cross Entropy |
|---|---|---|
| **EM** | *78.5%* | *0.827* |
| **Gradient EM** | 66.0% | 1.29 |
| **Gradient Descent** | 62.4% | 1.30 |

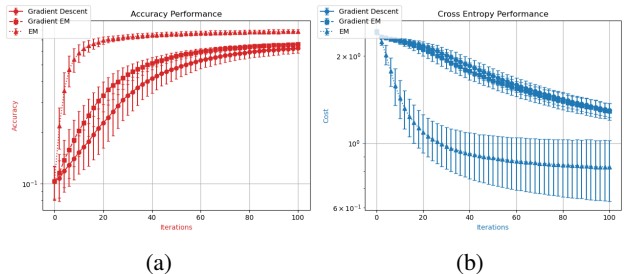

(a)                    (b)

*Figure 3.* Test accuracy and objective function, $\frac{1}{n}\sum_{i=1}^{n}\mathbb{1}_{\hat{y}_i=y_i}$ and $\mathcal{L}(\theta^t)$ in 3a and 3b, respectively, averaged over 25 instances for a 2-component MoLogE train on Random Invert FMNIST.

## 7. Conclusion

In this paper, we theoretically addressed the problem of Mixtures of Experts (MoE) with the use of the EM algorithm. We first showed that the EM update for MoE could be interpreted as a projected Mirror Descent step on the log-likelihood with a unit step size and a KL divergence regularizer, extending the result of (Kunstner et al., 2021) beyond complete data distribution belonging to an exponential family. Building on this, we characterized different convergence rates for EM in this setting under various assumptions about the log-likelihood function and specified when these assumptions held. Lastly, we empirically observed that EM can outperform gradient descent in both convergence rate and final performance.

## Impact Statement

This paper presents work whose goal is to advance the field of Machine Learning. There are many potential societal consequences of our work, none which we feel must be specifically highlighted here.

## Acknowledgements

This research was supported by NSF Encore Tripods (2217069), NSF's AI Institute IFML (2019844) and NSF Grant 2007668.

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

# Appendix

## Appendix Contents

**Notations**

We summarize here the notations used throughout the paper. For clarity, we distinguish between different types of mathematical objects (e.g., vectors, random variables, distributions) and follow standard conventions where possible.

The Kullback-Leibler (KL) divergence of a distribution $p$ from a distribution $q$ is denoted by $\mathrm{KL}[q \,\|\, p] := \int q(x) \log\left(q(x)/p(x)\right) dx$. We use lowercase letters such as $p$ to denote continuous probability density functions and uppercase letters such as $P$ to denote discrete probability mass functions. The Euclidean (or $\ell_2$) norm of a vector is denoted by $\|\cdot\|_2$. We use the compact notation $[k] := \{1, 2, ..., k\}$.

We denote vectors using bold lowercase letters (e.g., $\boldsymbol{x}$), and random variables using uppercase letters (e.g., $X$). Bold uppercase letters (e.g., $\boldsymbol{X}$) are used to represent either vector-valued random variables or matrices; the distinction between the two is clear from context. For a matrix $\boldsymbol{M}$, we denote its $i^{\text{th}}$ eigenvalue by $\lambda_i$, and the corresponding eigenvector by $\boldsymbol{v}_i$. The minimum and maximum eigenvalues of $\boldsymbol{M}$ are denoted by $\lambda_{\min}$ and $\lambda_{\max}$, respectively. We use $\boldsymbol{I}_d$ to denote the $d \times d$ identity matrix, and $\boldsymbol{e}_i$ to denote the $i^{\text{th}}$ standard basis (unit) vector in $\mathbb{R}^d$.

Expectations are written as $\mathbb{E}_{\boldsymbol{X}}[f(\boldsymbol{X})] = \int p(\boldsymbol{x}) f(\boldsymbol{x}) d\boldsymbol{x}$, where the distribution of $\boldsymbol{X}$ is implicitly defined by the context. When needed, we make the dependence on parameters explicit by writing $\boldsymbol{X}; \boldsymbol{\theta}$, where $\boldsymbol{\theta}$ denotes the parameters of the distribution of $\boldsymbol{X}$. The notation $\mathcal{N}(\boldsymbol{\mu}, \boldsymbol{\Sigma})$ refers to the multivariate normal distribution with mean vector $\boldsymbol{\mu}$ and covariance matrix $\boldsymbol{\Sigma}$.

# A. EM, Projected Mirror Descent, and General MoE.

In this Section, we provide all the complete proofs and discussions relating to results from Section 4.

## A.1. EM is Projected Mirror Descent for General MoE.

In this section, we provide the full and detailed proof of the main result, Theorem 4.1. For ease of comprehension and in the hope that this will provide useful insights into other types of non-exponential family mixtures for which EM is also connected to MD, we prove our result following the same general ideas as that of Kunstner et al. (2021, Proposition 1).

For ease of reading, we re-state the theorem below:

**Theorem 4.1:** For General MoE, there exists a natural re-parameterization $\boldsymbol{\theta_x} \in \{\eta(\cdot, \boldsymbol{\theta}) : \theta \in \Omega\}$ with

$$\mathcal{L}(\boldsymbol{\theta}) = \mathbb{E}_X\left[L(\boldsymbol{\theta_x})\right]$$

and a mirror map $A(\boldsymbol{\theta_x})$ such that the EM update in (9) simplifies and is equivalent to the expectation moment projection,

$$\underset{\boldsymbol{\theta} \in \Omega}{\operatorname{argmin}} \, \mathbb{E}_X\left[\mathrm{KL}\left[p\left(y, z \Big| \tilde{\theta}_{\boldsymbol{x}}^{t+1}\right) \Big\| p\left(y, z | \eta(\boldsymbol{x}, \boldsymbol{\theta})\right)\right]\right],$$

where for each $\boldsymbol{x}$, $\tilde{\theta}_{\boldsymbol{x}}^{t+1}$ is obtained from the following MD step,

$$\underset{\boldsymbol{\psi} \in \tilde{\Omega}}{\operatorname{argmin}} \langle \nabla L(\boldsymbol{\theta}_{\boldsymbol{x}}^t), \boldsymbol{\psi} - \boldsymbol{\theta}_{\boldsymbol{x}}^t \rangle + D_A(\boldsymbol{\psi}, \boldsymbol{\theta}_{\boldsymbol{x}}^t),$$

with $L(\boldsymbol{\theta_x})$ being 1-smooth relative to $A(\boldsymbol{\theta_x})$. Further, $\forall \boldsymbol{\psi}_1, \boldsymbol{\psi}_2 \in \tilde{\Omega}$, the divergence function $D_A(\boldsymbol{\psi}_1, \boldsymbol{\psi}_2)$ is equal to the $KL$ divergence on $p(y, z, | \psi)$:

$$D_A(\boldsymbol{\psi}_1, \boldsymbol{\psi}_2) = \mathrm{KL}[p(y, z | \boldsymbol{\psi}_2) \| p(y, z | \boldsymbol{\psi}_1)].$$

*Proof.* The EM is centered around iterative minimization of the surrogate upper-bound $Q(\phi|\theta)$. For conditionally exponential family of distribution, We can decompose it in terms of the sufficient statistic and log-partition:

$$Q(\boldsymbol{\theta}|\boldsymbol{\theta}^t) = -\mathbb{E}_{Y,X}\left[\sum_z \ln(p(y, \boldsymbol{x}, z; \boldsymbol{\theta})) P(z|y, \boldsymbol{x}; \boldsymbol{\theta}^t)\right]$$

$$= -\mathbb{E}_{Y,X}\left[\sum_z \left(\ln(p(y, z|\boldsymbol{x}; \boldsymbol{\theta})) + \ln(p(\boldsymbol{x}))\right) P(z|y, \boldsymbol{x}; \boldsymbol{\theta}^t)\right]$$

$$= -\mathbb{E}_{Y,X}\left[\sum_z \left(\langle S(y, z), \theta_{\boldsymbol{x}}\rangle - A(\boldsymbol{\theta}_{\boldsymbol{x}}^t)\right) + \ln(p(\boldsymbol{x}))\right) P(z|y, \boldsymbol{x}; \boldsymbol{\theta}^t)\right]$$

$$= \mathbb{E}_X\left[-\langle \underbrace{\mathbb{E}_{Y|\boldsymbol{x}, \boldsymbol{\theta}^*} \mathbb{E}_{Z|y, \boldsymbol{x}, \boldsymbol{\theta}^t}\left[S(y, z)\right]}_{s(\boldsymbol{x}; \boldsymbol{\theta}^t)}, \boldsymbol{\theta}_{\boldsymbol{x}}\rangle + A(\boldsymbol{\theta}_{\boldsymbol{x}}) - \ln(p(\boldsymbol{x}))\right]$$

$$= \mathbb{E}_X\left[-\langle s(\boldsymbol{x}; \boldsymbol{\theta}^t), \boldsymbol{\theta}_{\boldsymbol{x}}\rangle + A(\boldsymbol{\theta}_{\boldsymbol{x}}) - \ln(p(\boldsymbol{x}))\right].$$

Therefore, the above also implies

$$\mathbb{E}_X\left[\nabla L(\boldsymbol{\theta}_{\boldsymbol{x}}^t)\right] = \mathbb{E}_X\left[\nabla Q(\boldsymbol{\theta}_{\boldsymbol{x}}^t | \boldsymbol{\theta}_{\boldsymbol{x}}^t)\right] \tag{33}$$

$$= \mathbb{E}_X\left[\nabla A(\boldsymbol{\theta}_{\boldsymbol{x}}) - s(\boldsymbol{x}; \boldsymbol{\theta}^t)\right] \tag{34}$$

Simple algebra then shows that

$$Q(\boldsymbol{\theta}|\boldsymbol{\theta}^t) - Q(\boldsymbol{\theta}^t|\boldsymbol{\theta}^t) = \mathbb{E}_X\left[-\langle s(\boldsymbol{x}; \boldsymbol{\theta}^t), \boldsymbol{\theta}_{\boldsymbol{x}}\rangle + A(\boldsymbol{\theta}_{\boldsymbol{x}}) + \langle s(\boldsymbol{x}; \boldsymbol{\theta}^t), \boldsymbol{\theta}_{\boldsymbol{x}}^t\rangle - A(\boldsymbol{\theta}_{\boldsymbol{x}}^t)\right]$$

$$= \mathbb{E}_X\left[-\langle s(\boldsymbol{x}; \boldsymbol{\theta}^t), \boldsymbol{\theta}_{\boldsymbol{x}} - \boldsymbol{\theta}_{\boldsymbol{x}}^t\rangle + A(\boldsymbol{\theta}_{\boldsymbol{x}}) - A(\boldsymbol{\theta}_{\boldsymbol{x}}^t)\right]$$

$$\overset{i)}{=} \mathbb{E}_X\left[-\langle s(\boldsymbol{x}; \boldsymbol{\theta}^t) - \nabla A(\boldsymbol{\theta}_{\boldsymbol{x}}^t), \boldsymbol{\theta}_{\boldsymbol{x}} - \boldsymbol{\theta}_{\boldsymbol{x}}^t\rangle + D_A(\boldsymbol{\theta}_{\boldsymbol{x}}, \boldsymbol{\theta}_{\boldsymbol{x}}^t)\right]$$

$$= \mathbb{E}_X\left[\langle \nabla L(\boldsymbol{\theta}_{\boldsymbol{x}}^t), \boldsymbol{\theta}_{\boldsymbol{x}} - \boldsymbol{\theta}_{\boldsymbol{x}}^t\rangle + D_A(\boldsymbol{\theta}_{\boldsymbol{x}}, \boldsymbol{\theta}_{\boldsymbol{x}}^t)\right]$$

where $i$) adds and subtracts $\langle \nabla A(\boldsymbol{\theta}_{\boldsymbol{x}}^t), \boldsymbol{\theta}_{\boldsymbol{x}} - \boldsymbol{\theta}_{\boldsymbol{x}}^t \rangle$. This is especially important as EM minimizes $Q(\boldsymbol{\theta}|\boldsymbol{\theta}^t) - Q(\boldsymbol{\theta}^t|\boldsymbol{\theta}^t)$ in each iteration. Now recall that $\mathcal{L}(\boldsymbol{\theta}) = Q(\boldsymbol{\theta}|\boldsymbol{\theta}^t) - H(\boldsymbol{\theta}|\boldsymbol{\theta}^t)$ and $H(\boldsymbol{\theta}^t|\boldsymbol{\theta}^t) - H(\boldsymbol{\theta}|\boldsymbol{\theta}^t) \leq 0$, it follows that

$$
\begin{aligned}
\mathcal{L}(\boldsymbol{\theta}) - \mathcal{L}(\boldsymbol{\theta}^t) &= Q(\boldsymbol{\theta}|\boldsymbol{\theta}^t) - Q(\boldsymbol{\theta}^t|\boldsymbol{\theta}^t) - H(\boldsymbol{\theta}|\boldsymbol{\theta}^t) - H(\boldsymbol{\theta}^t|\boldsymbol{\theta}^t) \\
&\leq Q(\boldsymbol{\theta}|\boldsymbol{\theta}^t) - Q(\boldsymbol{\theta}^t|\boldsymbol{\theta}^t) \\
&= \mathbb{E}_X \left[ \langle \nabla L(\boldsymbol{\theta}_{\boldsymbol{x}}^t), \boldsymbol{\theta}_{\boldsymbol{x}} - \boldsymbol{\theta}_{\boldsymbol{x}}^t \rangle + D_A(\boldsymbol{\theta}_{\boldsymbol{x}}, \boldsymbol{\theta}_{\boldsymbol{x}}^t) \right]
\end{aligned}
$$

Thus far, we have shown that the EM iteration under the considered setting is equivalent to minimizing the following upper bound on $\mathcal{L}(\boldsymbol{\theta})$, w.r.t. $\boldsymbol{\theta}$:

$$
\mathcal{L}(\boldsymbol{\theta}^t) + \mathbb{E}_X \left[ \langle \nabla L(\boldsymbol{\theta}_{\boldsymbol{x}}^t), \boldsymbol{\theta}_{\boldsymbol{x}} - \boldsymbol{\theta}_{\boldsymbol{x}}^t \rangle + D_A(\boldsymbol{\theta}_{\boldsymbol{x}}, \boldsymbol{\theta}_{\boldsymbol{x}}^t) \right] \tag{35}
$$

Now, recall $\tilde{\theta}_{\boldsymbol{x}}^{t+1} := \arg\min_{\boldsymbol{\theta}_{\boldsymbol{x}}} \langle \nabla L(\boldsymbol{\theta}_{\boldsymbol{x}}^t), \boldsymbol{\theta}_{\boldsymbol{x}} - \boldsymbol{\theta}_{\boldsymbol{x}}^t \rangle + D_A(\boldsymbol{\theta}_{\boldsymbol{x}}, \boldsymbol{\theta}_{\boldsymbol{x}}^t)$ which is the outcome of a mirror descent step. Differentiating and setting equal to 0, it holds that

$$
\nabla A(\tilde{\theta}_{\boldsymbol{x}}^{t+1}) = s(\boldsymbol{x}; \boldsymbol{\theta}^t). \tag{36}
$$

Using this and decomposing $\nabla L(\boldsymbol{\theta}_{\boldsymbol{x}})$ above, we see that (35) is equal to

$$
\begin{aligned}
&= \mathcal{L}(\boldsymbol{\theta}^t) + \mathbb{E}_X \left[ \langle \nabla A(\boldsymbol{\theta}_{\boldsymbol{x}}^t) - s(\boldsymbol{x}; \boldsymbol{\theta}_{\boldsymbol{x}}^t), \boldsymbol{\theta}_{\boldsymbol{x}} - \boldsymbol{\theta}_{\boldsymbol{x}}^t \rangle + D_A(\boldsymbol{\theta}_{\boldsymbol{x}}, \boldsymbol{\theta}_{\boldsymbol{x}}^t) \right] \\
&= \mathcal{L}(\boldsymbol{\theta}^t) + \mathbb{E}_X \left[ \langle \nabla A(\boldsymbol{\theta}_{\boldsymbol{x}}^t) - \nabla A(\tilde{\theta}_{\boldsymbol{x}}^{t+1}), \boldsymbol{\theta}_{\boldsymbol{x}} - \boldsymbol{\theta}_{\boldsymbol{x}}^t \rangle + D_A(\boldsymbol{\theta}_{\boldsymbol{x}}, \boldsymbol{\theta}_{\boldsymbol{x}}^t) \right] \\
&= \mathcal{L}(\boldsymbol{\theta}^t) + \mathbb{E}_X \left[ -\langle \nabla A(\tilde{\theta}_{\boldsymbol{x}}^{t+1}), \boldsymbol{\theta}_{\boldsymbol{x}} - \boldsymbol{\theta}_{\boldsymbol{x}}^t \rangle + A(\boldsymbol{\theta}_{\boldsymbol{x}}) - A(\boldsymbol{\theta}_{\boldsymbol{x}}^t) \right].
\end{aligned}
$$

Thus, minimizing (35) with respect to $\boldsymbol{\theta}_{\boldsymbol{x}}$ is equivalent to minimizing

$$
\mathbb{E}_X \left[ D_A(\boldsymbol{\theta}_{\boldsymbol{x}}, \tilde{\theta}_{\boldsymbol{x}}^{t+1}) \right]. \tag{37}
$$

Substituting the Bregman Divergence induced by $A$ by the KL divergence yields the claim.

It remains to verify that $D_A(\boldsymbol{\theta}_{\boldsymbol{x}}, \tilde{\boldsymbol{\theta}}_{\boldsymbol{x}}^{t+1}) = \mathrm{KL}\left[ p(y,z|\tilde{\boldsymbol{\theta}}_{\boldsymbol{x}}^{t+1}) || p(y,z|\boldsymbol{\theta}_{\boldsymbol{x}}) \right]$ and that the function $L(\boldsymbol{\theta}_{\boldsymbol{x}})$ is 1-smooth relative to $A(\boldsymbol{\theta}_{\boldsymbol{x}})$. This follows directly from previous work by Kunstner et al. (2021) since $L(\boldsymbol{\theta}_{\boldsymbol{x}})$ is the expected negative log-likelihood of $y|\boldsymbol{x}$ where $A(\boldsymbol{\theta}_{\boldsymbol{x}})$ is the log-partition of the exponential distribution $p(y,z|\boldsymbol{\theta}_{\boldsymbol{x}})$. For completeness, the derivation is as follows. For $Y, Z|\boldsymbol{x}$ belonging to an exponential family of distribution, the KL divergence can be decomposed directly using (36) as follows to obtain the Bregman Divergence:

$$
\begin{aligned}
\mathrm{KL}\left[ p(y,z|\tilde{\boldsymbol{\theta}}_{\boldsymbol{x}}^{t+1}) || p(y,z|\boldsymbol{\theta}_{\boldsymbol{x}}) \right] &:= \mathbb{E}_{Y,Z|\boldsymbol{x}; \tilde{\boldsymbol{\theta}}_{\boldsymbol{x}}^{t+1}} \left[ \log \left( \frac{p(y,z; \tilde{\boldsymbol{\theta}}_{\boldsymbol{x}}^{t+1})}{p(y,z; \boldsymbol{\theta}_{\boldsymbol{x}})} \right) \right] \\
&= \mathbb{E}_{Y,Z|\boldsymbol{x}; \tilde{\boldsymbol{\theta}}_{\boldsymbol{x}}^{t+1}} \left[ \langle S(y,z), \tilde{\boldsymbol{\theta}}_{\boldsymbol{x}}^{t+1} - \boldsymbol{\theta}_{\boldsymbol{x}} \rangle + A(\boldsymbol{\theta}_{\boldsymbol{x}}) - A(\tilde{\boldsymbol{\theta}}_{\boldsymbol{x}}^{t+1}) \right] \\
&= \langle \nabla A(\tilde{\boldsymbol{\theta}}_{\boldsymbol{x}}^{t+1}), \tilde{\boldsymbol{\theta}}_{\boldsymbol{x}}^{t+1} - \boldsymbol{\theta}_{\boldsymbol{x}} \rangle + A(\boldsymbol{\theta}_{\boldsymbol{x}}) - A(\tilde{\boldsymbol{\theta}}_{\boldsymbol{x}}^{t+1}) \\
&= D_A(\boldsymbol{\theta}_{\boldsymbol{x}}, \tilde{\boldsymbol{\theta}}_{\boldsymbol{x}}^{t+1}).
\end{aligned}
$$

$\square$

## A.2. Convergence Results for EM applied to General MoE.

In this section, we provide the full proof of Theorem 4.2. Before we begin, we recall and discuss the regularity assumptions previously made. Recall assumptions $A_1$, $A_2$, and $A_3$:

$A_1$   The conditional distribution $p(y,z|\boldsymbol{\theta}_{\boldsymbol{x}})$ is a steep, minimal exponential family of distribution and $\eta(\boldsymbol{x}, \cdot)$ is a continuously differentiable function,

$A_2$   The optimal objective function values is bounded below, i.e., $\mathcal{L}(\boldsymbol{\theta}^*) > -\infty$, on the constraint set $\Omega$,

$A_3$   The following sub-level sets $\Omega_{\theta} := \{ \boldsymbol{\phi} \in \Omega : Q(\boldsymbol{\phi}|\boldsymbol{\theta}) \leq Q(\boldsymbol{\theta}|\boldsymbol{\theta}) \}$ are compact. .

Assumption $A_1$ serves to ensure that $\mathcal{L}(\theta) = \mathbb{E}_X [L(\theta_x)]$ is differentiable, that the EM surrogate is also differentiable and has a solution. It further serves to guarantee the mirror map $A$ is smooth, ensuring that projecting into the dual space is well-defined. We note that if the re-parametrization function is continuously differentiable in $\theta$, it will hold that $A_1$ is satisfied for the popular case that $p(y, z|x)$ is a Gaussian mixture (this includes MoE with Gaussian experts). Next, $A_2$ and $A_3$ are classical optimization assumptions that serve to guarantee the solution of the M-step is unique and exists within the constraint set $\Omega$, thereby ensuring the EM iterations are well defined.

Further, we make the additional remark that the projection, (17), in Theorem 4.1 can be seen to be equivalent to the following projection over the space of functions on $x$.

$$\eta(x, \theta^{t+1}) = \theta_x^{t+1} = \underset{\phi_x \in \{\eta(\cdot, \theta): \theta \in \Omega\}}{\operatorname{argmin}} \mathbb{E}_X \left[ D_A(\phi_x, \tilde{\theta}_x^{t+1}) \right] \tag{38}$$

We can see that the set $\{\eta(\cdot, \theta) : \theta \in \Omega\}$ is not necessarily guaranteed to be convex. Such a result would require the re-parametrization function $\eta(x, \theta)$ to be affine. In situations where this set is not convex, we cannot take our weak generalized Pythagorean identity (25) for granted, and thus we include it as an extra assumption to be satisfied for these convergence results. Still, convexity is not necessary for our generalized Pythagorean inequality to hold. For instance, ensuring $\tilde{\theta}_x^{t+1}$ is in the relative interior of $\{\eta(\cdot, \theta) : \theta \in \Omega\}$, or simply satisfying the inequality in expectation will suffice, but may be difficult to show.

For ease of reading, we re-state the result below:

**Theorem 4.2:** For general MoE with re-parameterization given by $\theta_x := \eta(x, \theta)$, strictly convex mirror map $A(\theta_x)$, and if for all $\tilde{\theta}_x^{t+1}, \theta_x^{t+1}, \phi_x \in \{\theta_x^t, \theta_x^*\}$,

$$\mathbb{E}_X \left[ D_A \left( \phi_x, \tilde{\theta}_x^{t+1} \right) \right] \geq \mathbb{E}_X \left[ D_A \left( \theta_x^{t+1}, \tilde{\theta}_x^{t+1} \right) + D_A \left( \phi_x, \theta_x^{t+1} \right) \right],$$

then, the EM iterates $\{\theta^t\}_{t \in [T]}$ satisfy:

1) **Stationnarity.** For no additional conditions,

$$\min_{t \in [T]} \mathbb{E}_X \left[ D_A(\theta_x^t, \theta_x^{t+1}) \right] \leq \frac{\mathcal{L}(\theta^1) - \mathcal{L}(\theta^*)}{T};$$

2) **Sub-linear Rate to $\theta^*$.** If $\theta^1$ is initialized in $\Theta$, a locally average-convex region of $\mathcal{L}(\theta)$ containing $\theta^*$, then

$$\mathcal{L}(\theta^T) - \mathcal{L}(\theta^*) \leq \frac{\mathbb{E}_X \left[ D_A(\theta_x^*, \theta_x^1) \right]}{T}$$

3) **Linear Rate to $\theta^*$.** If $\theta^1$ is initialized in $\Theta \subseteq \Omega$, a locally average-strongly convex region of $\mathcal{L}(\theta)$ relative to $A(\theta)$ that contains $\theta^*$, then

$$\mathcal{L}(\theta^T) - \mathcal{L}(\theta^*) \leq (1 - \alpha)^T \mathbb{E}_X \left[ D_A(\theta_x^*, \theta_x^1) \right]$$

*Proof.* The proof is divided into three parts that correspond to each of the three sub-results of the corollary.

To aid the reader's comprehension, we re-state the cosine law for Bregman divergence (also known as 3-point lemma).

**Lemma A.1** (cosine law for Bregman divergence). *Assume the mapping $A$ is proper and convex. Then, for all $a, b, c \in \tilde{\Omega}$, it holds that*

$$D_A(a, b) = D_A(a, c) + D_A(c, b) - \langle \nabla A(b) - \nabla A(c), a - c \rangle. \tag{39}$$

**Part 1):** Stationarity.
We begin by utilizing the result from Theorem 4.1 that the conditional log-likelihood $L(\theta_x)$ is 1-smooth relative to the mirror map $A(\theta_x)$. For all $\theta_x \in \tilde{\Omega}$,

$$L(\theta_x^{t+1}) \leq L(\theta_x^t) + \langle \nabla L(\theta_x^t), \theta_x^{t+1} - \theta_x^t \rangle + D_A(\theta_x^{t+1}, \theta_x^t).$$

Taking the expectation on both sides with respect to the random feature variable $x$ yields:

$$\mathcal{L}(\theta^{t+1}) \leq \mathcal{L}(\theta^t) + \mathbb{E}_X \left[ \langle \nabla L(\theta_x^t), \theta_x^{t+1} - \theta_x^t \rangle + D_A(\theta_x^{t+1}, \theta_x^t) \right].$$

Combining (33) and (36) then plugging into the above, we obtain

$$\mathcal{L}(\boldsymbol{\theta}^{t+1}) \leq \mathcal{L}(\boldsymbol{\theta}^t) + \mathbb{E}_X\left[\langle \nabla A(\boldsymbol{\theta}_{\boldsymbol{x}}^t) - \nabla A(\tilde{\boldsymbol{\theta}}_{\boldsymbol{x}}^{t+1}), \boldsymbol{\theta}_{\boldsymbol{x}}^{t+1} - \boldsymbol{\theta}_{\boldsymbol{x}}^t\rangle + D_A(\boldsymbol{\theta}_{\boldsymbol{x}}^{t+1}, \boldsymbol{\theta}_{\boldsymbol{x}}^t)\right].$$

Decomposing $D_A(\boldsymbol{\theta}_{\boldsymbol{x}}^{t+1}, \boldsymbol{\theta}_{\boldsymbol{x}}^t)$ above and canceling appropriate terms yields

$$\mathcal{L}(\boldsymbol{\theta}^{t+1}) \leq \mathcal{L}(\boldsymbol{\theta}^t) + \underbrace{\mathbb{E}_X\left[-\langle \nabla A(\tilde{\boldsymbol{\theta}}_{\boldsymbol{x}}^{t+1}), \boldsymbol{\theta}_{\boldsymbol{x}}^{t+1} + A(\boldsymbol{\theta}_{\boldsymbol{x}}^{t+1}) - A(\boldsymbol{\theta}_{\boldsymbol{x}}^t)\right]}_{i)}.$$

It can then be checked that i) is equal to $\mathbb{E}_X\left[D_A(\boldsymbol{\theta}_{\boldsymbol{x}}^{t+1}, \tilde{\boldsymbol{\theta}}_{\boldsymbol{x}}^{t+1}) - D_A(\boldsymbol{\theta}_{\boldsymbol{x}}^t, \tilde{\boldsymbol{\theta}}_{\boldsymbol{x}}^{t+1})\right]$. Therefore, substituting the above and utilizing the inequality (25), it follows that

$$\mathcal{L}(\boldsymbol{\theta}^{t+1}) \leq \mathcal{L}(\boldsymbol{\theta}^t) - \mathbb{E}_X\left[D_A(\boldsymbol{\theta}_{\boldsymbol{x}}^t, \boldsymbol{\theta}_{\boldsymbol{x}}^{t+1})\right].$$

Re-arranging the terms and averaging over $T$-iterations yields the claim:

$$\mathbb{E}_X\left[D_A(\boldsymbol{\theta}_{\boldsymbol{x}}^t, \boldsymbol{\theta}_{\boldsymbol{x}}^{t+1})\right] \leq \mathcal{L}(\boldsymbol{\theta}^{t+1}) - \mathcal{L}(\boldsymbol{\theta}^t)$$

$$\implies \min_{t \leq T} \mathbb{E}_X\left[D_A(\boldsymbol{\theta}_{\boldsymbol{x}}^t, \boldsymbol{\theta}_{\boldsymbol{x}}^{t+1})\right] \leq \frac{1}{T}\sum_{t=1}^T \mathcal{L}(\boldsymbol{\theta}^{t+1}) - \mathcal{L}(\boldsymbol{\theta}^t) = \frac{\mathcal{L}(\boldsymbol{\theta}^1) - \mathcal{L}(\boldsymbol{\theta}^T)}{T}$$

**Part 2):** Sub-linear rate to $\theta^*$.
We begin by utilizing the result from Theorem 4.1 that the conditional log-likelihood $L(\boldsymbol{\theta}_{\boldsymbol{x}})$ is 1-smooth relative to the mirror map $A(\boldsymbol{\theta}_{\boldsymbol{x}})$. For all $\boldsymbol{\theta}_{\boldsymbol{x}} \in \tilde{\Omega}$, then apply expectation with respect to $\boldsymbol{x}$ on both sides yielding:

$$\mathcal{L}(\boldsymbol{\theta}^{t+1}) \leq \mathcal{L}(\boldsymbol{\theta}^t) + \mathbb{E}_X\left[\langle \nabla L(\boldsymbol{\theta}_{\boldsymbol{x}}^t), \boldsymbol{\theta}_{\boldsymbol{x}}^{t+1} - \boldsymbol{\theta}_{\boldsymbol{x}}^t\rangle + D_A(\boldsymbol{\theta}_{\boldsymbol{x}}^{t+1}, \boldsymbol{\theta}_{\boldsymbol{x}}^t)\right].$$

We then add and subtract $\mathbb{E}_X\left[\langle \nabla L(\boldsymbol{\theta}_{\boldsymbol{x}}^t), \boldsymbol{\theta}_{\boldsymbol{x}}^*\rangle\right]$ to obtain

$$\mathcal{L}(\boldsymbol{\theta}^{t+1}) \leq \mathcal{L}(\boldsymbol{\theta}^t) + \mathbb{E}_X\left[\langle \nabla L(\boldsymbol{\theta}_{\boldsymbol{x}}^t), \boldsymbol{\theta}_{\boldsymbol{x}}^{t+1} - \boldsymbol{\theta}_{\boldsymbol{x}}^* + \boldsymbol{\theta}_{\boldsymbol{x}}^* - \boldsymbol{\theta}_{\boldsymbol{x}}^t\rangle + D_A(\boldsymbol{\theta}_{\boldsymbol{x}}^{t+1}, \boldsymbol{\theta}_{\boldsymbol{x}}^t)\right].$$

We then use the average local convexity assumption, (23), and obtain

$$\mathcal{L}(\boldsymbol{\theta}^{t+1}) \leq \mathcal{L}(\boldsymbol{\theta}^*) + \mathbb{E}_X\left[\langle \nabla L(\boldsymbol{\theta}_{\boldsymbol{x}}^t), \boldsymbol{\theta}_{\boldsymbol{x}}^{t+1} - \boldsymbol{\theta}_{\boldsymbol{x}}^*\rangle + D_A(\boldsymbol{\theta}_{\boldsymbol{x}}^{t+1}, \boldsymbol{\theta}_{\boldsymbol{x}}^t)\right].$$

Combining (33) and (36) then plugging into the above, we obtain

$$\mathcal{L}(\boldsymbol{\theta}^{t+1}) \leq \mathcal{L}(\boldsymbol{\theta}^*) + \mathbb{E}_X\left[\langle \nabla A(\boldsymbol{\theta}_{\boldsymbol{x}}^t) - \nabla A(\tilde{\boldsymbol{\theta}}_{\boldsymbol{x}}^{t+1}), \boldsymbol{\theta}_{\boldsymbol{x}}^{t+1} - \boldsymbol{\theta}_{\boldsymbol{x}}^*\rangle + D_A(\boldsymbol{\theta}_{\boldsymbol{x}}^{t+1}, \boldsymbol{\theta}_{\boldsymbol{x}}^t)\right]. \tag{40}$$

We now decompose $D_A(\boldsymbol{\theta}_{\boldsymbol{x}}^{t+1}, \boldsymbol{\theta}_{\boldsymbol{x}}^t)$ using Lemma A.1 with $a = \tilde{\boldsymbol{\theta}}_{\boldsymbol{x}}^{t+1}$, $b = \boldsymbol{\theta}_{\boldsymbol{x}}^t$, and $c = \boldsymbol{\theta}_{\boldsymbol{x}}^{t+1}$ and obtain

$$\langle \nabla A(\boldsymbol{\theta}_{\boldsymbol{x}}^t) - \nabla A(\tilde{\boldsymbol{\theta}}_{\boldsymbol{x}}^{t+1}), \boldsymbol{\theta}_{\boldsymbol{x}}^{t+1} - \boldsymbol{\theta}_{\boldsymbol{x}}^*\rangle + D_A(\boldsymbol{\theta}_{\boldsymbol{x}}^{t+1}, \boldsymbol{\theta}_{\boldsymbol{x}}^t)$$
$$= \langle \nabla A(\boldsymbol{\theta}_{\boldsymbol{x}}^t) - \nabla A(\tilde{\boldsymbol{\theta}}_{\boldsymbol{x}}^{t+1}), \boldsymbol{\theta}_{\boldsymbol{x}}^{t+1} - \boldsymbol{\theta}_{\boldsymbol{x}}^*\rangle + D_A(\tilde{\boldsymbol{\theta}}_{\boldsymbol{x}}^{t+1}, \boldsymbol{\theta}_{\boldsymbol{x}}^t) - D_A(\tilde{\boldsymbol{\theta}}_{\boldsymbol{x}}^{t+1}, \boldsymbol{\theta}_{\boldsymbol{x}}^{t+1}) + \langle \nabla A(\boldsymbol{\theta}_{\boldsymbol{x}}^t) - \nabla A(\boldsymbol{\theta}_{\boldsymbol{x}}^{t+1}), \tilde{\boldsymbol{\theta}}_{\boldsymbol{x}}^{t+1} - \boldsymbol{\theta}_{\boldsymbol{x}}^{t+1}\rangle$$
$$= A(\tilde{\boldsymbol{\theta}}_{\boldsymbol{x}}^{t+1}) - A(\boldsymbol{\theta}_{\boldsymbol{x}}^t) + \langle \nabla A(\boldsymbol{\theta}_{\boldsymbol{x}}^t), \boldsymbol{\theta}_{\boldsymbol{x}}^t - \boldsymbol{\theta}_{\boldsymbol{x}}^*\rangle + \langle -\nabla A(\tilde{\boldsymbol{\theta}}_{\boldsymbol{x}}^{t+1}), \boldsymbol{\theta}_{\boldsymbol{x}}^{t+1} - \boldsymbol{\theta}_{\boldsymbol{x}}^*\rangle + A(\boldsymbol{\theta}_{\boldsymbol{x}}^{t+1}) - A(\tilde{\boldsymbol{\theta}}_{\boldsymbol{x}}^{t+1})$$

We now add and subtract $\langle \nabla A(\boldsymbol{\theta}_{\boldsymbol{x}}^t), \tilde{\boldsymbol{\theta}}_{\boldsymbol{x}}^{t+1}\rangle$ and $\langle A(\tilde{\boldsymbol{\theta}}_{\boldsymbol{x}}^{t+1}), \tilde{\boldsymbol{\theta}}_{\boldsymbol{x}}^{t+1}\rangle$ and group terms to obtain

$$D_A(\boldsymbol{\theta}_{\boldsymbol{x}}^{t+1}, \tilde{\boldsymbol{\theta}}_{\boldsymbol{x}}^{t+1}) + \underbrace{D_A(\tilde{\boldsymbol{\theta}}_{\boldsymbol{x}}^{t+1}, \boldsymbol{\theta}_{\boldsymbol{x}}^t) + \langle \nabla A(\boldsymbol{\theta}_{\boldsymbol{x}}^t) - \nabla A(\tilde{\boldsymbol{\theta}}_{\boldsymbol{x}}^{t+1}), \tilde{\boldsymbol{\theta}}_{\boldsymbol{x}}^{t+1} - \boldsymbol{\theta}_{\boldsymbol{x}}^*\rangle}_{ii)}$$

We now apply Lemma A.1 again to ii) with $a = \boldsymbol{\theta}_{\boldsymbol{x}}^*$, $b = \boldsymbol{\theta}_{\boldsymbol{x}}^t$, $c = \tilde{\boldsymbol{\theta}}_{\boldsymbol{x}}^{t+1}$ and obtain the sub-result:

$$\langle \nabla A(\boldsymbol{\theta}_{\boldsymbol{x}}^t) - \nabla A(\tilde{\boldsymbol{\theta}}_{\boldsymbol{x}}^{t+1}), \boldsymbol{\theta}_{\boldsymbol{x}}^{t+1} - \boldsymbol{\theta}_{\boldsymbol{x}}^*\rangle + D_A(\boldsymbol{\theta}_{\boldsymbol{x}}^{t+1}, \boldsymbol{\theta}_{\boldsymbol{x}}^t) = D_A(\boldsymbol{\theta}_{\boldsymbol{x}}^*, \boldsymbol{\theta}_{\boldsymbol{x}}^t) - D_A(\boldsymbol{\theta}_{\boldsymbol{x}}^*, \tilde{\boldsymbol{\theta}}_{\boldsymbol{x}}^{t+1}) + D_A(\boldsymbol{\theta}_{\boldsymbol{x}}^{t+1}, \tilde{\boldsymbol{\theta}}_{\boldsymbol{x}}^{t+1})$$

Plugging the above equality into (40), we obtain

$$\mathcal{L}(\boldsymbol{\theta}^{t+1}) \le \mathcal{L}(\boldsymbol{\theta}^*) + \mathbb{E}_X \left[ D_A(\boldsymbol{\theta}_{\boldsymbol{x}}^*, \boldsymbol{\theta}_{\boldsymbol{x}}^t) - D_A(\boldsymbol{\theta}_{\boldsymbol{x}}^*, \tilde{\boldsymbol{\theta}}_{\boldsymbol{x}}^{t+1}) + D_A(\boldsymbol{\theta}_{\boldsymbol{x}}^{t+1}, \tilde{\boldsymbol{\theta}}_{\boldsymbol{x}}^{t+1}) \right].$$

Then, using 25, we obtain

$$\mathcal{L}(\boldsymbol{\theta}^{t+1}) \le \mathcal{L}(\boldsymbol{\theta}^*) + \mathbb{E}_X \left[ D_A(\boldsymbol{\theta}_{\boldsymbol{x}}^*, \boldsymbol{\theta}_{\boldsymbol{x}}^t) - D_A(\boldsymbol{\theta}_{\boldsymbol{x}}^*, \boldsymbol{\theta}_{\boldsymbol{x}}^{t+1}) \right].$$

Re-arranging, then averaging over $T$ iterations yields the claim:

$$T(\mathcal{L}(\boldsymbol{\theta}^T) - \mathcal{L}(\boldsymbol{\theta}^*)) \le \sum_{t=1}^T \left( \mathcal{L}(\boldsymbol{\theta}^t) - \mathcal{L}(\boldsymbol{\theta}^*) \right) \le \sum_{t=1}^T \mathbb{E}_X \left[ D_A(\boldsymbol{\theta}_{\boldsymbol{x}}^*, \boldsymbol{\theta}_{\boldsymbol{x}}^t) - D_A(\boldsymbol{\theta}_{\boldsymbol{x}}^*, \boldsymbol{\theta}_{\boldsymbol{x}}^{t+1}) \right]$$

$$\implies \mathcal{L}(\boldsymbol{\theta}^T) - \mathcal{L}(\boldsymbol{\theta}^*) \le \frac{\mathbb{E}_X \left[ D_A(\boldsymbol{\theta}_{\boldsymbol{x}}^*, \boldsymbol{\theta}_{\boldsymbol{x}}^1) - D_A(\boldsymbol{\theta}_{\boldsymbol{x}}^*, \boldsymbol{\theta}_{\boldsymbol{x}}^T) \right]}{T} \le \frac{\mathbb{E}_X \left[ D_A(\boldsymbol{\theta}_{\boldsymbol{x}}^*, \boldsymbol{\theta}_{\boldsymbol{x}}^1) \right]}{T}$$

**Part 3):** Linear rate to $\theta^*$.

We begin by utilizing the result from Theorem 4.1 that the conditional log-likelihood $L(\boldsymbol{\theta}_{\boldsymbol{x}})$ is 1-smooth relative to the mirror map $A(\boldsymbol{\theta}_{\boldsymbol{x}})$. For all $\boldsymbol{\theta}_{\boldsymbol{x}} \in \tilde{\Omega}$, then apply expectation with respect to $\boldsymbol{x}$ on both sides yielding:

$$\mathcal{L}(\boldsymbol{\theta}^{t+1}) \le \mathcal{L}(\boldsymbol{\theta}^t) + \mathbb{E}_X \left[ \langle \nabla L(\boldsymbol{\theta}_{\boldsymbol{x}}^t), \boldsymbol{\theta}_{\boldsymbol{x}}^{t+1} - \boldsymbol{\theta}_{\boldsymbol{x}}^t \rangle + D_A(\boldsymbol{\theta}_{\boldsymbol{x}}^{t+1}, \boldsymbol{\theta}_{\boldsymbol{x}}^t) \right].$$

We then add and subtract $\mathbb{E}_X \left[ \langle \nabla L(\boldsymbol{\theta}_{\boldsymbol{x}}^t), \boldsymbol{\theta}_{\boldsymbol{x}}^* \rangle \right]$ to obtain

$$\mathcal{L}(\boldsymbol{\theta}^{t+1}) \le \mathcal{L}(\boldsymbol{\theta}^t) + \mathbb{E}_X \left[ \langle \nabla L(\boldsymbol{\theta}_{\boldsymbol{x}}^t), \boldsymbol{\theta}_{\boldsymbol{x}}^{t+1} - \boldsymbol{\theta}_{\boldsymbol{x}}^* + \boldsymbol{\theta}_{\boldsymbol{x}}^* - \boldsymbol{\theta}_{\boldsymbol{x}}^t \rangle + D_A(\boldsymbol{\theta}_{\boldsymbol{x}}^{t+1}, \boldsymbol{\theta}_{\boldsymbol{x}}^t) \right].$$

We then use the local $\alpha$-strongly average-convexity assumption, (23), and obtain

$$\mathcal{L}(\boldsymbol{\theta}^{t+1}) \le \mathcal{L}(\boldsymbol{\theta}^*) + \mathbb{E}_X \left[ \langle \nabla L(\boldsymbol{\theta}_{\boldsymbol{x}}^t), \boldsymbol{\theta}_{\boldsymbol{x}}^{t+1} - \boldsymbol{\theta}_{\boldsymbol{x}}^* \rangle + D_A(\boldsymbol{\theta}_{\boldsymbol{x}}^{t+1}, \boldsymbol{\theta}_{\boldsymbol{x}}^t) - \alpha D_A(\boldsymbol{\theta}_{\boldsymbol{x}}^*, \boldsymbol{\theta}_{\boldsymbol{x}}^t) \right].$$

Then, following the same steps as for the sub-linear case, we Combining (33) and (36), utilize the cosine law for Bregman Divergence $D_A(\cdot, \cdot)$ twice, then apply (25) to obtain:

$$\mathcal{L}(\boldsymbol{\theta}^{t+1}) \le \mathcal{L}(\boldsymbol{\theta}^*) + \mathbb{E}_X \left[ D_A(\boldsymbol{\theta}_{\boldsymbol{x}}^*, \boldsymbol{\theta}_{\boldsymbol{x}}^t) - D_A(\boldsymbol{\theta}_{\boldsymbol{x}}^*, \boldsymbol{\theta}_{\boldsymbol{x}}^{t+1}) - \alpha D_A(\boldsymbol{\theta}_{\boldsymbol{x}}^*, \boldsymbol{\theta}_{\boldsymbol{x}}^t) \right]$$
$$\le \mathcal{L}(\boldsymbol{\theta}^*) + \mathbb{E}_X \left[ (1 - \alpha) D_A(\boldsymbol{\theta}_{\boldsymbol{x}}^*, \boldsymbol{\theta}_{\boldsymbol{x}}^t) \right].$$

Unraveling the recurrence over $T$ iterations, yields the result:

$$\mathcal{L}(\boldsymbol{\theta}^T) - \mathcal{L}(\boldsymbol{\theta}^*) \le (1 - \alpha)^T \mathbb{E}_X \left[ D_A(\boldsymbol{\theta}_{\boldsymbol{x}}^*, \boldsymbol{\theta}_{\boldsymbol{x}}^1) \right]$$

This completes the proof. □

# B. EM, Mirror Descent, and SymMoLogE and SymMoLinE.

In this section, we provide all results, proofs, and discussion pertaining to EM for symmetric mixtures of logistic or linear experts.

## B.1. EM is Mirror Descent for SymMoLogE and SymMoLinE

In this section, we provide the full and detailed proof of Theorem 5.1. For ease of comprehension and in the hope that this will provide useful insights into other types of non-exponential family mixtures for which EM is also connected to MD, we prove our result following the same general ideas as that of (Kunstner et al., 2021, Proposition 1). We split the proof into two parts (SymMoLinE and SymMoLogE) which can be found in Appendix B.2 and B.3.

For ease of reading, we re-state the theorem below:

**Theorem 5.1:** For SymMoLinE and SymMoLogE, there is a mirror map $A(\boldsymbol{\theta})$ such that the EM update in (9) simplifies and is equivalent to

$$\underset{\boldsymbol{\theta} \in \Omega}{\operatorname{argmin}} \langle \nabla \mathcal{L}(\boldsymbol{\theta}), \boldsymbol{\theta} - \boldsymbol{\theta}^t \rangle + D_A(\boldsymbol{\theta}, \boldsymbol{\theta}^t),$$

where $\forall \boldsymbol{\phi}, \boldsymbol{\theta} \in \Omega$ the divergence function $D_A(\boldsymbol{\theta}, \boldsymbol{\theta}^t)$ is equal to the $KL$ divergence on the complete data:

$$D_A(\boldsymbol{\phi}, \boldsymbol{\theta}) = \mathrm{KL}[p(\boldsymbol{x}, y, z; \boldsymbol{\theta}) \| p(\boldsymbol{x}, y, z; \boldsymbol{\phi})].$$

In particular, in the case of SymMoLinE,

$$A(\boldsymbol{\theta}) = \mathbb{E}_{\boldsymbol{X}} \left[ \frac{(\boldsymbol{x}^\top \boldsymbol{\beta})^2}{2} + \log \left( 1 + e^{\boldsymbol{x}^\top \boldsymbol{w}} \right) \right],$$

while in the case of SymMoLogE,

$$A(\boldsymbol{\theta}) = \mathbb{E}_{\boldsymbol{X}} \left[ \log \left( \left( 1 + e^{\boldsymbol{x}^\top \boldsymbol{\beta}} \right) \left( 1 + e^{\boldsymbol{x}^\top \boldsymbol{w}} \right) \right) \right].$$

Finally, in both cases, the map $A(\boldsymbol{\theta})$ is strictly convex in $\boldsymbol{\theta}$ and $\mathcal{L}(\boldsymbol{\theta})$ is 1-smooth relative to $A(\boldsymbol{\theta})$.

## B.2. Proof of Theorem 5.1 for SymMoLinE

*Proof.* Recall that we consider a 2 component SymMoLinE (see Section 3.1) where $z \in \{-1, 1\}$ is the latent unobserved variable, and

1) $\boldsymbol{x} \sim \mathcal{N}(\boldsymbol{0}, \boldsymbol{I}_d)$,

2) $P(z|\boldsymbol{x}; \boldsymbol{w}) = \frac{\exp\{\frac{z+1}{2}\boldsymbol{x}^\top \boldsymbol{w}\}}{1 + e^{\boldsymbol{x}^\top \boldsymbol{w}}}$,

3) $p(y|\boldsymbol{x}, z; \boldsymbol{\beta}) = \frac{\exp\left\{ -\frac{(y - z\boldsymbol{x}^\top \boldsymbol{\beta})^2}{2} \right\}}{\sqrt{2\pi}}$.

We begin by deriving a near exponential form of the complete data probability density function $p(\boldsymbol{x}, z, y; \boldsymbol{\theta})$:

$$
\begin{aligned}
& p(\boldsymbol{x}, y, z; \boldsymbol{\theta}) \\
&= p(y|\boldsymbol{x}, z; \boldsymbol{\theta}) P(z|\boldsymbol{x}; \boldsymbol{\theta}) p(\boldsymbol{x}) \\
&= \exp\{\log p(y|\boldsymbol{x}, z; \boldsymbol{\beta}) + \log P(z|\boldsymbol{x}; \boldsymbol{w}) + \log p(\boldsymbol{x})\} \\
&= \exp\left\{ \frac{-(y - z\boldsymbol{x}^\top \boldsymbol{\beta})^2}{2} - \frac{1}{2}\log(2\pi) + \left(\frac{z+1}{2}\right)\boldsymbol{x}^\top \boldsymbol{w} - \log(1 + e^{\boldsymbol{x}^\top \boldsymbol{w}}) + \log p(\boldsymbol{x}) \right\} \\
&= \exp\left\{ \frac{-y^2}{2} + yz\boldsymbol{x}^\top \boldsymbol{\beta} - \frac{z^2(\boldsymbol{x}^\top \boldsymbol{\beta})^2}{2} + \left(\frac{z+1}{2}\right)\boldsymbol{x}^\top \boldsymbol{w} - \log(1 + e^{\boldsymbol{x}^\top \boldsymbol{w}}) + \log p(\boldsymbol{x}) - \frac{1}{2}\log(2\pi) \right\} \\
&= \exp\left\{ \left\langle \begin{bmatrix} yz\boldsymbol{x} \\ \frac{z\boldsymbol{x}}{2} \end{bmatrix}, \begin{bmatrix} \boldsymbol{\beta} \\ \boldsymbol{w} \end{bmatrix} \right\rangle + \frac{\boldsymbol{x}^\top \boldsymbol{w}}{2} - \frac{(\boldsymbol{x}^\top \boldsymbol{\beta})^2}{2} - \log(1 + e^{\boldsymbol{x}^\top \boldsymbol{w}}) + \log p(\boldsymbol{x}) - \frac{y^2}{2} - \frac{1}{2}\log(2\pi) \right\}.
\end{aligned}
$$

Thus we have recovered the decomposition,

$$p(\boldsymbol{x}, y, z; \boldsymbol{\theta}) = \exp \left\{ \left\langle \underbrace{\begin{bmatrix} \frac{z\boldsymbol{x}}{2} \\ yz\boldsymbol{x} \end{bmatrix}}_{S(\boldsymbol{x}, y, z)}, \begin{bmatrix} \boldsymbol{w} \\ \boldsymbol{\beta} \end{bmatrix} \right\rangle + a(\boldsymbol{x}, y, \boldsymbol{\theta}) \right\}, \tag{41}$$

where in $a(\boldsymbol{x}, y, \boldsymbol{\theta})$, the feature variable $\boldsymbol{x}$ cannot be linearly separated from the parameter $\boldsymbol{\theta}$:

$$a(\boldsymbol{x}, y, \boldsymbol{\theta}) = \frac{\boldsymbol{x}^\top \boldsymbol{w}}{2} - \frac{(\boldsymbol{x}^\top \boldsymbol{\beta})^2}{2} - \log\left(1 + e^{\boldsymbol{x}^\top \boldsymbol{w}}\right) + \log p(\boldsymbol{x}) - \frac{y^2}{2} - \frac{1}{2} \log(2\pi).$$

At this point, we pause and discuss the implications of the obtained form. First we recall that for a random variable $\boldsymbol{U}$ to belong to an exponential family, it must satisfy

$$p(\boldsymbol{u}; \boldsymbol{\theta}) = h(\boldsymbol{u}) \exp\left\{\langle s(\boldsymbol{u}), \boldsymbol{\theta} \rangle - A(\boldsymbol{\theta})\right\}$$

for some $h(\cdot), s(\cdot), \boldsymbol{\theta}, A(\cdot)$ that are called the normalization function, sufficient statistics, natural parameters, and log-partition function respectively. To clarify, we note that 1) $A(\boldsymbol{\theta})$ must be a function of the parameters only and cannot depend on $\boldsymbol{u}$ and 2) $h(\boldsymbol{u})$ must be a function of the variable $\boldsymbol{u}$ only and cannot depend on the parameters. In other words, it must be that $h(\boldsymbol{u})$ and $A(\boldsymbol{\theta})$ are linearly separated inside the $\exp$. With the above in mind, we see that SymMoLinE is not an exponential family. Further, we remark that the above formulation showing $S(\boldsymbol{x}, y, z)$ is linear with $(\boldsymbol{w}, \boldsymbol{\beta})^\top$ does not extend beyond the **symmetric** setting of the Mixture of Linear Experts; for $k \geq 3$, this relationships becomes non-linear. This turns out to be problematic for showing EM is equivalent to MD for $k \geq 3$. Lastly, note that taking the expectation of $a(\boldsymbol{x}, y, \boldsymbol{\theta})$ over $(\boldsymbol{x}, y) \sim p(\boldsymbol{x}, y; \boldsymbol{\theta}^*)$ yields,

$$\mathbb{E}_{\boldsymbol{X}, Y}[a(\boldsymbol{x}, y, \boldsymbol{\theta})] = -\mathbb{E}_{\boldsymbol{X}} \left[ \frac{(\boldsymbol{x}^\top \boldsymbol{\beta})^2}{2} + \log\left(1 + e^{\boldsymbol{x}^\top \boldsymbol{w}}\right) \right] - C$$

$$= -A(\boldsymbol{\theta}) - C,$$

where the above follows from $\mathbb{E}_{\boldsymbol{X}}[\frac{\boldsymbol{x}^\top \boldsymbol{w}}{2}] = 0$ and $C := -\mathbb{E}_{\boldsymbol{X}, Y}[\log p(\boldsymbol{x}) - \frac{y^2}{2} - \frac{1}{2} \log(2\pi)]$ is not a function of the parameter $\boldsymbol{\theta}$. With the obtained form (41), we now continue with the proof.

**Part a):** Show EM is MD, i.e., $\operatorname{argmin}_{\boldsymbol{\theta} \in \Omega} Q(\boldsymbol{\theta}|\boldsymbol{\theta}^t) = \operatorname{argmin}_{\boldsymbol{\theta} \in \Omega} \langle \nabla \mathcal{L}(\boldsymbol{\theta}), \boldsymbol{\theta} - \boldsymbol{\theta}^t \rangle + D_A(\boldsymbol{\theta}, \boldsymbol{\theta}^t)$.

Taking the appropriate expectation, the EM objective $Q$ can be written as

$$\begin{aligned}
Q(\boldsymbol{\theta}|\boldsymbol{\theta}^t) &= -\mathbb{E}_{\boldsymbol{X}, Y}\left[ \mathbb{E}_{Z|\boldsymbol{x}, y; \boldsymbol{\theta}^t} \left[ \log p(\boldsymbol{x}, y, z; \boldsymbol{\theta}) \right] \right] \\
&= -\mathbb{E}_{\boldsymbol{X}, Y}\left[ \mathbb{E}_{Z|\boldsymbol{x}, y; \boldsymbol{\theta}^t} \left[ \langle S(\boldsymbol{x}, y, z), \boldsymbol{\theta} \rangle + a(\boldsymbol{x}, y, \boldsymbol{\theta}) \right] \right] \\
&= -\mathbb{E}_{\boldsymbol{X}, Y}[a(\boldsymbol{x}, y, \boldsymbol{\theta})] - \mathbb{E}_{\boldsymbol{X}, Y}\left[ \mathbb{E}_{Z|\boldsymbol{x}, y; \boldsymbol{\theta}^t} \left[ \langle S(\boldsymbol{x}, y, z), \boldsymbol{\theta} \rangle \right] \right] \\
&= A(\boldsymbol{\theta}) - \langle s(\boldsymbol{\theta}^t), \boldsymbol{\theta} \rangle + C
\end{aligned}$$

where $s(\boldsymbol{\theta}^t) := \mathbb{E}_{\boldsymbol{X}, Y} \mathbb{E}_{Z|\boldsymbol{x}, y; \boldsymbol{\theta}^t}[S(\boldsymbol{x}, y, z)]$. As a consequence, it is also true that

$$\nabla Q(\boldsymbol{\theta}^t|\boldsymbol{\theta}^t) = \nabla A(\boldsymbol{\theta}^t) - s(\boldsymbol{\theta}^t). \tag{42}$$

Continuing, we use the above to simplify the expression for $Q(\boldsymbol{\theta}|\boldsymbol{\theta}^t) - Q(\boldsymbol{\theta}^t|\boldsymbol{\theta}^t)$ that will subsequently give us the MD loss:

$$\begin{aligned}
Q(\boldsymbol{\theta}|\boldsymbol{\theta}^t) - Q(\boldsymbol{\theta}^t|\boldsymbol{\theta}^t) &= A(\boldsymbol{\theta}) - \langle s(\boldsymbol{\theta}^t), \boldsymbol{\theta} \rangle - A(\boldsymbol{\theta}^t) + \langle s(\boldsymbol{\theta}^t), \boldsymbol{\theta}^t \rangle \\
&= -\langle s(\boldsymbol{\theta}^t), \boldsymbol{\theta} - \boldsymbol{\theta}^t \rangle + \langle \nabla A(\boldsymbol{\theta}^t), \boldsymbol{\theta} - \boldsymbol{\theta}^t \rangle - \langle \nabla A(\boldsymbol{\theta}^t), \boldsymbol{\theta} - \boldsymbol{\theta}^t \rangle + A(\boldsymbol{\theta}) - A(\boldsymbol{\theta}^t) \\
&\overset{i)}{=} \langle \nabla Q(\boldsymbol{\theta}^t|\boldsymbol{\theta}^t), \boldsymbol{\theta} - \boldsymbol{\theta}^t \rangle + D_A(\boldsymbol{\theta}, \boldsymbol{\theta}^t) \\
&\overset{ii)}{=} \langle \nabla \mathcal{L}(\boldsymbol{\theta}^t), \boldsymbol{\theta} - \boldsymbol{\theta}^t \rangle + D_A(\boldsymbol{\theta}, \boldsymbol{\theta}^t)
\end{aligned}$$

where we first adding and subtracting $\langle \nabla A(\boldsymbol{\theta}^t), \boldsymbol{\theta} - \boldsymbol{\theta}^t \rangle$ then $i)$ follows from (42) and $ii)$ follows from $\nabla \mathcal{L}(\boldsymbol{\theta}) = \nabla Q(\boldsymbol{\theta}|\boldsymbol{\theta})$ (see Section 3 for the derivation).

Finally, the first part of our result follows trivially as

$$\underset{\boldsymbol{\theta} \in \Omega}{\operatorname{argmin}} \, Q(\boldsymbol{\theta}|\boldsymbol{\theta}^t) = \underset{\boldsymbol{\theta} \in \Omega}{\operatorname{argmin}} \, Q(\boldsymbol{\theta}|\boldsymbol{\theta}^t) - Q(\boldsymbol{\theta}^t|\boldsymbol{\theta}^t).$$

**Part b):** Show $D_A(\boldsymbol{\phi}, \boldsymbol{\theta}) = \mathrm{KL}[p(\boldsymbol{x}, y, z; \boldsymbol{\theta}) \| p(\boldsymbol{x}, y, z; \boldsymbol{\phi})]$

This result follows simply from decomposing $\mathrm{KL}[p(\boldsymbol{x}, y, z; \boldsymbol{\theta}) \| p(\boldsymbol{x}, y, z; \boldsymbol{\phi})]$ as follows:

$$\mathrm{KL}[p(\boldsymbol{x}, y, z; \boldsymbol{\theta}) \| p(\boldsymbol{x}, y, z; \boldsymbol{\phi})] = \mathbb{E}_{\boldsymbol{X}, Y, Z | \boldsymbol{\theta}} \left[ \log \frac{p(\boldsymbol{x}, y, z; \boldsymbol{\theta})}{p(\boldsymbol{x}, y, z; \boldsymbol{\phi})} \right]$$

$$\overset{(41)}{=} \langle s(\boldsymbol{\theta}), \boldsymbol{\theta} - \boldsymbol{\phi} \rangle - A(\boldsymbol{\theta}) + A(\boldsymbol{\phi}) \pm \mathbb{E}_{\boldsymbol{X}, Y | \boldsymbol{\theta}} \left[ \log p(\boldsymbol{x}) - \frac{y^2}{2} - \frac{1}{2} \log(2\pi) \right]$$

$$= A(\boldsymbol{\phi}) - A(\boldsymbol{\theta}) - \langle s(\boldsymbol{\theta}), \boldsymbol{\phi} - \boldsymbol{\theta} \rangle$$

$$\overset{i)}{=} A(\boldsymbol{\phi}) - A(\boldsymbol{\theta}) - \langle \nabla A(\boldsymbol{\theta}), \boldsymbol{\phi} - \boldsymbol{\theta} \rangle.$$

where $i)$ follows from the fact that $\boldsymbol{\phi} = \boldsymbol{\theta}$ minimizes $-\mathbb{E}_{\boldsymbol{X}, Z, Y | \boldsymbol{\theta}} [\log p(\boldsymbol{x}, y, z; \boldsymbol{\phi})]$. To see this, we use Jensen's inequality:

$$0 \leq -\mathbb{E}_{\boldsymbol{X}, Z, Y | \boldsymbol{\theta}} [\log p(\boldsymbol{x}, y, z; \boldsymbol{\theta})]$$

$$\overset{\text{Jensen's}}{\leq} -\log \mathbb{E}_{\boldsymbol{X}, Z, Y | \boldsymbol{\theta}} [p(\boldsymbol{x}, y, z; \boldsymbol{\theta})] = -\log \int_{\boldsymbol{x}, y, z} p(\boldsymbol{x}, y, z : \boldsymbol{\theta})^2 d\boldsymbol{x} dz dy$$

$$\overset{\text{Jensen's}}{\leq} -\log \left( \int_{\boldsymbol{x}, y, z} p(\boldsymbol{x}, y, z; \boldsymbol{\theta}) d\boldsymbol{x} dz dy \right)^2 = -\log(1) = 0.$$

Finally, taking the derivative with respect to $\boldsymbol{\phi}$ and setting equal to 0 completes the proof:

$$\mathbf{0} = \frac{\partial}{\partial \boldsymbol{\theta}} \mathbb{E}_{\boldsymbol{X}, Z, Y | \boldsymbol{\theta}} [\log p(\boldsymbol{x}, y, z; \boldsymbol{\phi})] |_{\boldsymbol{\phi} = \boldsymbol{\theta}}$$

$$= \mathbb{E}_{\boldsymbol{X}, Z, Y | \boldsymbol{\theta}} \left[ S(\boldsymbol{x}, y, z) + \frac{\partial}{\partial \boldsymbol{\phi}} a(\boldsymbol{x}, y, \boldsymbol{\phi}) |_{\boldsymbol{\phi} = \boldsymbol{\theta}} \right]$$

$$= s(\boldsymbol{\theta}) - \nabla A(\boldsymbol{\theta}).$$

**Part c):** Show $\mathcal{L}(\boldsymbol{\theta})$ is 1-smooth relative to $A(\boldsymbol{\theta})$.

The function $\mathcal{L}(\boldsymbol{\theta})$ is said to be 1-smooth relative to $A(\boldsymbol{\theta})$ if for all $\boldsymbol{\phi}, \boldsymbol{\theta}$, it holds that

$$\mathcal{L}(\boldsymbol{\theta}) \leq \mathcal{L}(\boldsymbol{\phi}) + \langle \nabla \mathcal{L}(\boldsymbol{\phi}, \boldsymbol{\theta} - \boldsymbol{\phi} \rangle + D_A(\boldsymbol{\theta}, \boldsymbol{\phi}).$$

Recall the following from Section 3. The objective function $\mathcal{L}(\boldsymbol{\theta})$ is related to the EM objective $Q(\boldsymbol{\phi}|\boldsymbol{\theta})$ by (10),

$$\mathcal{L}(\boldsymbol{\theta}) = Q(\boldsymbol{\theta}|\boldsymbol{\phi}) - H(\boldsymbol{\theta}|\boldsymbol{\phi}),$$

where $H(\boldsymbol{\phi}|\boldsymbol{\theta}) \geq 0$ and $H(\boldsymbol{\theta}|\boldsymbol{\theta}) = 0$ for all $\boldsymbol{\phi}, \boldsymbol{\theta} \in \Omega$. Consequently, it then holds that for all $\boldsymbol{\phi}, \boldsymbol{\theta} \in \Omega$,

$$\mathcal{L}(\boldsymbol{\theta}) = Q(\boldsymbol{\theta}|\boldsymbol{\theta}) \tag{43}$$

$$\mathcal{L}(\boldsymbol{\theta}) \leq Q(\boldsymbol{\theta}|\boldsymbol{\phi}). \tag{44}$$

Recall also from part a) that $Q(\boldsymbol{\theta}|\boldsymbol{\phi}) - Q(\boldsymbol{\phi}|\boldsymbol{\phi}) = \langle \nabla \mathcal{L}(\boldsymbol{\phi}), \boldsymbol{\theta} - \boldsymbol{\phi} \rangle + D_A(\boldsymbol{\theta}, \boldsymbol{\phi})$. Then, the claim follows naturally from the above as follows:

$$\mathcal{L}(\boldsymbol{\theta}) \overset{(44)}{\leq} Q(\boldsymbol{\theta}|\boldsymbol{\phi})$$

$$\overset{a)}{=} Q(\boldsymbol{\phi}|\boldsymbol{\phi}) + \langle \nabla \mathcal{L}(\boldsymbol{\phi}), \boldsymbol{\theta} - \boldsymbol{\phi} \rangle + D_A(\boldsymbol{\theta}, \boldsymbol{\phi})$$

$$\overset{(43)}{=} \mathcal{L}(\boldsymbol{\phi}) + \langle \nabla \mathcal{L}(\boldsymbol{\phi}), \boldsymbol{\theta} - \boldsymbol{\phi} \rangle + D_A(\boldsymbol{\theta}, \boldsymbol{\phi})$$

It follows that $\mathcal{L}(\boldsymbol{\theta})$ is 1-smooth relative to $A(\boldsymbol{\theta})$.

**Part d):** $A(\boldsymbol{\theta})$ and the MD objective is convex with respect to $\boldsymbol{\theta}$.

Here, we will show that the mirror descent objective is strongly convex in $\boldsymbol{\theta}$. It is important for this to hold so that the iterations of MD are well-defined; the minimizer of a strongly convex objective exists and is unique.

Note that the mirror descent objective, $\langle \nabla \mathcal{L}(\boldsymbol{\theta}), \boldsymbol{\theta} - \boldsymbol{\theta}^t \rangle + D_A(\boldsymbol{\theta}, \boldsymbol{\theta}^t)$, is strongly convex in $\boldsymbol{\theta}$ if $A(\boldsymbol{\theta})$ is strongly convex in $\boldsymbol{\theta}$. Therefore, since $A(\boldsymbol{\theta})$ given in (31) is twice continuously differentiable, it is strongly convex with respect to $\boldsymbol{\theta}$ if and only if $\nabla^2 A(\boldsymbol{\theta}) \succeq r \boldsymbol{I}_{2d}$, for some $r > 0$. We begin:

$$
\begin{aligned}
\nabla^2 A(\boldsymbol{\theta}) &= \frac{\partial^2}{\partial \boldsymbol{\theta}^2} \mathbb{E}_{\boldsymbol{X}} \left[ \frac{(\boldsymbol{x}^\top \boldsymbol{\beta})^2}{2} + \log\left(1 + e^{\boldsymbol{x}^\top \boldsymbol{w}}\right) \right] \\
&= \mathbb{E}_{\boldsymbol{X}} \left[ \frac{\partial^2}{\partial \boldsymbol{\theta}^2} \left( \frac{(\boldsymbol{x}^\top \boldsymbol{\beta})^2}{2} + \log\left(1 + e^{\boldsymbol{x}^\top \boldsymbol{w}}\right) \right) \right] \\
&= \begin{pmatrix} \mathbb{E}_{\boldsymbol{X}} \left[ \boldsymbol{x}\boldsymbol{x}^\top \frac{e^{\boldsymbol{x}^\top \boldsymbol{w}}}{\left(1 + e^{\boldsymbol{x}^\top \boldsymbol{w}}\right)^2} \right] & \boldsymbol{0} \\ \boldsymbol{0} & \mathbb{E}_{\boldsymbol{X}} \left[ \boldsymbol{x}\boldsymbol{x}^\top \right] \end{pmatrix} \\
&= \begin{pmatrix} \mathbb{E}_{\boldsymbol{X}} \left[ \boldsymbol{x}\boldsymbol{x}^\top \frac{e^{\boldsymbol{x}^\top \boldsymbol{w}}}{\left(1 + e^{\boldsymbol{x}^\top \boldsymbol{w}}\right)^2} \right] & \boldsymbol{0} \\ \boldsymbol{0} & \boldsymbol{I}_d \end{pmatrix}
\end{aligned}
$$

where the last line follows from the assumption that $\boldsymbol{x}$ is sampled from a unit spherical Gaussian distribution: $\mathbb{E}_{\boldsymbol{X}} \left[ \boldsymbol{x}\boldsymbol{x}^\top \right] = \boldsymbol{I}_d$ for $\boldsymbol{x} \sim \mathcal{N}(\boldsymbol{0}, \boldsymbol{I}_d)$.

From the above, we see that $A(\boldsymbol{\theta})$ is strictly convex, and it is strongly convex if $\mathbb{E}_{\boldsymbol{X}} \left[ \boldsymbol{x}\boldsymbol{x}^\top \frac{e^{\boldsymbol{x}^\top \boldsymbol{w}}}{\left(1 + e^{\boldsymbol{x}^\top \boldsymbol{w}}\right)^2} \right] \succeq r \boldsymbol{I}_d$ for some $r > 0$. This follows from Lemma B.5 where we show its eigenvalues are bounded below by $\min \left\{ \Omega\left(\frac{1}{\|\boldsymbol{w}\|_2}\right), \Omega\left(\frac{1}{\|\boldsymbol{w}\|_2^3}\right) \right\}$. Thus, it holds that

$$
\nabla^2 A(\boldsymbol{\theta}) \succeq \min \left\{ \Omega\left(\frac{1}{\|\boldsymbol{w}\|_2}\right), \Omega\left(\frac{1}{\|\boldsymbol{w}\|_2^3}\right), 1 \right\} \boldsymbol{I}_{2d}.
$$

Restricting the feasible set $\Omega$ to be all $\boldsymbol{\theta} \in \mathbb{R}^{2d}$ with $\|\boldsymbol{\theta}\|_2 \le N$ for some $N \in [0, \infty)$, it holds that $A(\boldsymbol{\theta})$ is strongly convex with respect to $\boldsymbol{\theta}$ on $\Omega$.

With part d) proven, this concludes the proof of Theorem 5.1 for SymMoLinE. We now prove the same for SymMoLogE, referring to this section where necessary.

$\square$

### B.3. Proof of Theorem 5.1 for SymMoLogE

*Proof.* Recall that we consider a 2 component SymMoLogE (see Section 3.1) where $z \in \{-1, 1\}$ is the latent unobserved variable, and

1) $\boldsymbol{x} \sim \mathcal{N}(\boldsymbol{0}, \boldsymbol{I}_d)$,

2) $P(z | \boldsymbol{x}; \boldsymbol{w}) = \frac{\exp\{\frac{z+1}{2} \boldsymbol{x}^\top \boldsymbol{w}\}}{1 + e^{\boldsymbol{x}^\top \boldsymbol{w}}}$.

3) $P(y | \boldsymbol{x}, z; \boldsymbol{\beta}) = \frac{\exp\{\left(\frac{yz+1}{2}\right) \boldsymbol{x}^\top \boldsymbol{\beta}\}}{1 + e^{\boldsymbol{x}^\top \boldsymbol{\beta}}}$.

We begin by deriving a near exponential form of the complete data probability density function $p(\boldsymbol{x}, z, y; \boldsymbol{\theta})$:

$$p(\boldsymbol{x}, z, y; \boldsymbol{\theta})$$
$$= P(y|\boldsymbol{x}, z; \boldsymbol{\theta}) P(z|\boldsymbol{x}; \boldsymbol{\theta}) p(\boldsymbol{x})$$
$$= \exp\{\log P(y|\boldsymbol{x}, z; \boldsymbol{\beta}) + \log P(z|\boldsymbol{x}; \boldsymbol{w}) + \log p(\boldsymbol{x})\}$$
$$= \exp\left\{ \log\left( \left(\frac{\exp\{\boldsymbol{x}^\top\boldsymbol{\beta}\}}{1+e^{\boldsymbol{x}^\top\boldsymbol{\beta}}}\right)^{\frac{yz+1}{2}} \left(\frac{1}{1+e^{\boldsymbol{x}^\top\boldsymbol{\beta}}}\right)^{1-\frac{yz+1}{2}} \right) + \left(\frac{z+1}{2}\right)\boldsymbol{x}^\top\boldsymbol{w} - \log(1+e^{\boldsymbol{x}^\top\boldsymbol{w}}) + \log p(\boldsymbol{x}) \right\}$$
$$= \exp\left\{ \frac{yz+1}{2}\log\left(\frac{\exp\{\boldsymbol{x}^\top\boldsymbol{\beta}\}}{1+e^{\boldsymbol{x}^\top\boldsymbol{\beta}}}\right) + \left(1-\frac{yz+1}{2}\right)\log\left(\frac{1}{1+e^{\boldsymbol{x}^\top\boldsymbol{\beta}}}\right) + \left(\frac{z+1}{2}\right)\boldsymbol{x}^\top\boldsymbol{w} - \log(1+e^{\boldsymbol{x}^\top\boldsymbol{w}}) + \log p(\boldsymbol{x}) \right\}$$
$$= \exp\left\{ \left(\frac{yz+1}{2}\right)\boldsymbol{x}^\top\boldsymbol{\beta} - \log\left(1+e^{\boldsymbol{x}^\top\boldsymbol{\beta}}\right) + \left(\frac{z+1}{2}\right)\boldsymbol{x}^\top\boldsymbol{w} - \log(1+e^{\boldsymbol{x}^\top\boldsymbol{w}}) + \log p(\boldsymbol{x}) \right\}$$
$$= \exp\left\{ \left\langle \begin{bmatrix} \frac{yz\boldsymbol{x}}{2} \\ \frac{z\boldsymbol{x}}{2} \end{bmatrix}, \begin{bmatrix} \boldsymbol{\beta} \\ \boldsymbol{w} \end{bmatrix} \right\rangle + \frac{\boldsymbol{x}^\top(\boldsymbol{w}+\boldsymbol{\beta})}{2} - \log\left[\left(1+e^{\boldsymbol{x}^\top\boldsymbol{\beta}}\right)\left(1+e^{\boldsymbol{x}^\top\boldsymbol{w}}\right)\right] + \log p(\boldsymbol{x}) \right\}.$$

Thus we have recovered the decomposition,

$$p(\boldsymbol{x}, y, z; \boldsymbol{\theta}) = \exp\left\{ \left\langle \begin{bmatrix} \frac{yz\boldsymbol{x}}{2} \\ \frac{z\boldsymbol{x}}{2} \end{bmatrix}, \begin{bmatrix} \boldsymbol{\beta} \\ \boldsymbol{w} \end{bmatrix} \right\rangle + a(\boldsymbol{x}, y, \boldsymbol{\theta}) \right\},, \tag{45}$$

where in $a(\boldsymbol{x}, y, \boldsymbol{\theta})$, $\boldsymbol{x}$ cannot be linearly separated from the parameter $\boldsymbol{\theta}$:

$$a(\boldsymbol{x}, y, \boldsymbol{\theta}) = \frac{\boldsymbol{x}^\top(\boldsymbol{w}+\boldsymbol{\beta})}{2} - \log\left[\left(1+e^{\boldsymbol{x}^\top\boldsymbol{\beta}}\right)\left(1+e^{\boldsymbol{x}^\top\boldsymbol{w}}\right)\right] + \log p(\boldsymbol{x}).$$

Similar to SymMoLinE, we can see that $p(\boldsymbol{x}, y, z; \boldsymbol{\theta})$ does not belong to an exponential family of distribution. Also, note that taking the expectation of $a(\boldsymbol{x}, y, \boldsymbol{\theta})$ over $(\boldsymbol{x}, y) \sim p(\boldsymbol{x}, y; \boldsymbol{\theta}^*)$ yields,

$$\mathbb{E}_{\boldsymbol{X}, Y}[a(\boldsymbol{x}, y, \boldsymbol{\theta})] = -\mathbb{E}_{\boldsymbol{X}}\left[\log\left[\left(1+e^{\boldsymbol{x}^\top\boldsymbol{\beta}}\right)\left(1+e^{\boldsymbol{x}^\top\boldsymbol{w}}\right)\right]\right] - C$$
$$= -A(\boldsymbol{\theta}) - C,$$

where the above follows from $\mathbb{E}_{\boldsymbol{X}}[\frac{\boldsymbol{x}^\top(\boldsymbol{w}+\boldsymbol{\beta})}{2}] = 0$ and $C := -\mathbb{E}_{\boldsymbol{X}, Y|\boldsymbol{\theta}^*}[\log p(\boldsymbol{x})]$ is not a function of the parameter $\boldsymbol{\theta}$. We now continue with the proof.

From here on, the proofs of part a), part b) and part c) follow identically from that of SymMoLinE, so we will refer to Appendix B.2 for those proofs. We will now show part d).

**Part d):** $A(\boldsymbol{\theta})$ and the MD objective is convex with respect to $\boldsymbol{\theta}$.

Here, we will show that the mirror descent objective is strongly convex in $\boldsymbol{\theta}$. It is important for this to hold so that the iterations of MD are well-defined; the minimizer of a strongly convex objective exists and is unique.

Note that the mirror descent objective, $\langle \nabla \mathcal{L}(\boldsymbol{\theta}), \boldsymbol{\theta} - \boldsymbol{\theta}^t \rangle + D_A(\boldsymbol{\theta}, \boldsymbol{\theta}^t)$, is strongly convex in $\boldsymbol{\theta}$ if $A(\boldsymbol{\theta})$ is strongly convex in $\boldsymbol{\theta}$. Therefore, since $A(\boldsymbol{\theta})$ given in (31) is twice continuously differentiable, it is strongly convex with respect to $\boldsymbol{\theta}$ if and only if $\nabla^2 A(\boldsymbol{\theta}) \succeq r\boldsymbol{I}_{2d}$, for some $r > 0$. We begin:

$$\nabla^2 A(\boldsymbol{\theta}) = \frac{\partial^2}{\partial\boldsymbol{\theta}^2}\mathbb{E}_{\boldsymbol{X}}\left[\log\left(\left(1+e^{\boldsymbol{x}^\top\boldsymbol{\beta}}\right)\left(1+e^{\boldsymbol{x}^\top\boldsymbol{w}}\right)\right)\right]$$
$$= \mathbb{E}_{\boldsymbol{X}}\left[\frac{\partial^2}{\partial\boldsymbol{\theta}^2}\left(\log\left(1+e^{\boldsymbol{x}^\top\boldsymbol{\beta}}\right) + \log\left(1+e^{\boldsymbol{x}^\top\boldsymbol{w}}\right)\right)\right]$$
$$= \begin{pmatrix} \mathbb{E}_{\boldsymbol{X}}\left[\boldsymbol{x}\boldsymbol{x}^\top\frac{e^{\boldsymbol{x}^\top\boldsymbol{w}}}{(1+e^{\boldsymbol{x}^\top\boldsymbol{w}})^2}\right] & \mathbf{0} \\ \mathbf{0} & \mathbb{E}_{\boldsymbol{X}}\left[\boldsymbol{x}\boldsymbol{x}^\top\frac{e^{\boldsymbol{x}^\top\boldsymbol{\beta}}}{(1+e^{\boldsymbol{x}^\top\boldsymbol{\beta}})^2}\right] \end{pmatrix}$$
$$= \begin{pmatrix} \mathbb{E}_{\boldsymbol{X}}\left[\boldsymbol{x}\boldsymbol{x}^\top\frac{e^{\boldsymbol{x}^\top\boldsymbol{w}}}{(1+e^{\boldsymbol{x}^\top\boldsymbol{w}})^2}\right] & \mathbf{0} \\ \mathbf{0} & \mathbb{E}_{\boldsymbol{X}}\left[\boldsymbol{x}\boldsymbol{x}^\top\frac{e^{\boldsymbol{x}^\top\boldsymbol{\beta}}}{(1+e^{\boldsymbol{x}^\top\boldsymbol{\beta}})^2}\right] \end{pmatrix}.$$

From the above, we see that $A(\boldsymbol{\theta})$ is strictly convex, and it is strongly convex if $\mathbb{E}_{\boldsymbol{X}}\left[\boldsymbol{x}\boldsymbol{x}^\top \frac{e^{\boldsymbol{x}^\top \boldsymbol{w}}}{\left(1+e^{\boldsymbol{x}^\top \boldsymbol{w}}\right)^2}\right] \succeq r\boldsymbol{I}_d$ and

$\mathbb{E}_{\boldsymbol{X}}\left[\boldsymbol{x}\boldsymbol{x}^\top \frac{e^{\boldsymbol{x}^\top \boldsymbol{\beta}}}{\left(1+e^{\boldsymbol{x}^\top \boldsymbol{\beta}}\right)^2}\right] \succeq r\boldsymbol{I}_d$ for some $r > 0$. This follows from Lemma B.5 where we show their respective eigenvalues

are bounded below by, $\min\left\{\Omega\left(\frac{1}{\|\boldsymbol{w}\|_2}\right), \Omega\left(\frac{1}{\|\boldsymbol{w}\|_2^3}\right)\right\}$ and $\min\left\{\Omega\left(\frac{1}{\|\boldsymbol{\beta}\|_2}\right), \Omega\left(\frac{1}{\|\boldsymbol{\beta}\|_2^3}\right)\right\}$. Thus, it holds that

$$\nabla^2 A(\boldsymbol{\theta}) \succeq \min\left\{\Omega\left(\frac{1}{\|\boldsymbol{w}\|_2}\right), \Omega\left(\frac{1}{\|\boldsymbol{w}\|_2^3}\right), \Omega\left(\frac{1}{\|\boldsymbol{\beta}\|_2}\right), \Omega\left(\frac{1}{\|\boldsymbol{\beta}\|_2^3}\right)\right\} \boldsymbol{I}_{2d}.$$

Restricting $\Omega$ to be all $\boldsymbol{\theta}$ with $\|\boldsymbol{\theta}\|_2 \leq N$ for some $N \in [0, \infty)$, it holds that $A(\boldsymbol{\theta})$ is strongly convex with respect to $\boldsymbol{\theta}$ on $\Omega$.

With part d) proven, this concludes the proof of Theorem 5.1 for SymMoLinE. We now prove the same for SymMoLogE, referring to this section where necessary.

$\square$

### B.4. Convergence Guarantees of EM for SymMoLogE and SymMoLinE

In this section, we provide the proofs of Corollary B.1. Building on prior work (Lu et al., 2018), we contextualize convergence properties of MD for SymMoLinE and SymMoLogE. Before presenting the result, we briefly review key concepts. We say $\boldsymbol{\theta}^1$ is initialized in a locally convex region of $\mathcal{L}(\boldsymbol{\theta})$ if there exists a convex set $\Theta \subseteq \Omega$ containing $\boldsymbol{\theta}^1, \boldsymbol{\theta}^*$ such that for all $\boldsymbol{\phi}, \boldsymbol{\theta} \in \Theta$,

$$\mathcal{L}(\boldsymbol{\phi}) \geq \mathcal{L}(\boldsymbol{\theta}) + \langle \nabla\mathcal{L}(\boldsymbol{\theta}), \boldsymbol{\phi} - \boldsymbol{\theta}\rangle. \tag{46}$$

Furthermore, $\Theta$ is called $\alpha$-strongly convex relative to $h$ if

$$\mathcal{L}(\boldsymbol{\phi}) \geq \mathcal{L}(\boldsymbol{\theta}) + \langle \nabla\mathcal{L}(\boldsymbol{\theta}), \boldsymbol{\phi} - \boldsymbol{\theta}\rangle + \alpha D_h(\boldsymbol{\phi}, \boldsymbol{\theta}). \tag{47}$$

The corollary that follows provides conditions for convergence of EM to (1) a stationary point in the KL divergence, (2) the true parameters at a sub-linear rate, and (3) the true parameters at a linear rate. We further note that the proof is adapted from (Kunstner et al., 2021, Proposition 2, Corollary 1, and Corollary 3) and (Lu et al., 2018, Theorem 3.1), we provide it here for completeness.

**Corollary B.1** (Convergence of EM). *For SymMoLinE, SymMoLogE with mirror map $A(\boldsymbol{\theta})$ given as* (31) (32) *respectively, and denoting $D_A(\boldsymbol{\theta}^t, \boldsymbol{\theta}^{t+1}) := KL[p(\boldsymbol{x}, y, z; \boldsymbol{\theta}^{t+1}) \| p(\boldsymbol{x}, y, z; \boldsymbol{\theta}^t)]$, the EM iterates $\{\boldsymbol{\theta}^t\}_{t \in [T]}$ satisfy:*

1) *Stationarity. For no additional conditions,*

$$\min_{t \in [T]} D_A(\boldsymbol{\theta}^{t+1}, \boldsymbol{\theta}^t) \leq \frac{\mathcal{L}(\boldsymbol{\theta}^1) - \mathcal{L}(\boldsymbol{\theta}^*)}{T}; \tag{48}$$

2) *Sub-linear Rate to $\boldsymbol{\theta}^*$. If $\boldsymbol{\theta}^1$ is initialized in $\Theta$, a locally convex region of $\mathcal{L}(\boldsymbol{\theta})$ containing $\boldsymbol{\theta}^*$, then*

$$\mathcal{L}(\boldsymbol{\theta}^T) - \mathcal{L}(\boldsymbol{\theta}^*) \leq \frac{D_A(\boldsymbol{\theta}^*, \boldsymbol{\theta}^1)}{T} \tag{49}$$

3) *Linear Rate to $\boldsymbol{\theta}^*$. If $\boldsymbol{\theta}^1$ is initialized in $\Theta \subseteq \Omega$, a locally strongly convex region of $\mathcal{L}(\boldsymbol{\theta})$ relative to $A(\boldsymbol{\theta})$ that contains $\boldsymbol{\theta}^*$, then*

$$\mathcal{L}(\boldsymbol{\theta}^T) - \mathcal{L}(\boldsymbol{\theta}^*) \leq (1 - \alpha)^T (\mathcal{L}(\boldsymbol{\theta}^1) - \mathcal{L}(\boldsymbol{\theta}^*)). \tag{50}$$

*Proof.* The proof is divided into three parts that correspond to each of the three sub-results of the corollary.

**Part 1):** Stationarity.
Given Theorem 5.1, this proof follows from identical arguments to that of (Kunstner et al., 2021, Proposition 2). We write it below for completeness.

Recall from Theorem 5.1 that $\boldsymbol{\theta}^{t+1}$ is obtained as the minimizer of the convex objective, (15):

$$\langle \nabla\mathcal{L}(\boldsymbol{\theta}^t), \boldsymbol{\theta} - \boldsymbol{\theta}^t\rangle + D_A(\boldsymbol{\theta}, \boldsymbol{\theta}^t).$$

As such, differentiating and setting equal to $0$, it holds that $\boldsymbol{\theta}^{t+1}$ satisfies

$$\nabla \mathcal{L}(\boldsymbol{\theta}^t) = \nabla A(\boldsymbol{\theta}^t) - \nabla A(\boldsymbol{\theta}^{t+1}) \tag{51}$$

Further, by the above together with relative smoothness, it holds that

$$
\begin{aligned}
\mathcal{L}(\boldsymbol{\theta}^{t+1}) &\leq \mathcal{L}(\boldsymbol{\theta}^t) + \langle \nabla \mathcal{L}(\boldsymbol{\theta}^t), \boldsymbol{\theta}^{t+1} - \boldsymbol{\theta}^t \rangle + D_A(\boldsymbol{\theta}^{t+1}, \boldsymbol{\theta}^t) \\
&= \mathcal{L}(\boldsymbol{\theta}^t) + \langle \nabla A(\boldsymbol{\theta}^t) - \nabla A(\boldsymbol{\theta}^{t+1}), \boldsymbol{\theta}^{t+1} - \boldsymbol{\theta}^t \rangle + D_A(\boldsymbol{\theta}^{t+1}, \boldsymbol{\theta}^t) \\
&= \mathcal{L}(\boldsymbol{\theta}^t) - \langle \nabla A(\boldsymbol{\theta}^{t+1}), \boldsymbol{\theta}^{t+1} - \boldsymbol{\theta}^t \rangle + A(\boldsymbol{\theta}^{t+1}) - A(\boldsymbol{\theta}^t) \\
&= \mathcal{L}(\boldsymbol{\theta}^t) - D_A(\boldsymbol{\theta}^t, \boldsymbol{\theta}^{t+1}).
\end{aligned}
$$

Thus it we have shown that

$$D_A(\boldsymbol{\theta}^t, \boldsymbol{\theta}^{t+1}) \leq \mathcal{L}(\boldsymbol{\theta}^t) - \mathcal{L}(\boldsymbol{\theta}^{t+1}). \tag{52}$$

The claim then follows from taking the mean over $T$ iterations:

$$\min_{t \leq T} D_A(\boldsymbol{\theta}^t, \boldsymbol{\theta}^{t+1}) \leq \frac{1}{T} \sum_{t=1}^{T} D_A(\boldsymbol{\theta}^t, \boldsymbol{\theta}^{t+1}) \leq \frac{1}{T} \sum_{t=1}^{T} \mathcal{L}(\boldsymbol{\theta}^t) - \mathcal{L}(\boldsymbol{\theta}^{t+1}) = \frac{\mathcal{L}(\boldsymbol{\theta}^1) - \mathcal{L}(\boldsymbol{\theta}^T)}{T} \leq \frac{\mathcal{L}(\boldsymbol{\theta}^1) - \mathcal{L}(\boldsymbol{\theta}^*)}{T}.$$

**Part 2):** Sub-linear Rate to $\boldsymbol{\theta}^*$.
Given Theorem 5.1, this proof follows from identical arguments to that of Kunstner et al. (2021, Corollary 1) and Lu et al. (2018, Theorem 3.1). We write it below for completeness.

Here, we assume that $\mathcal{L}(\boldsymbol{\theta})$ is convex on the set $\Theta$. In part 1), we used (51) to show,

$$\langle \nabla \mathcal{L}(\boldsymbol{\theta}^t), \boldsymbol{\theta}^{t+1} - \boldsymbol{\theta}^t \rangle + D_A(\boldsymbol{\theta}^{t+1}, \boldsymbol{\theta}^t) = -D_A(\boldsymbol{\theta}^t, \boldsymbol{\theta}^{t+1}),$$

where the right hand side is non-positive since the Bregman divergence is non-negative if the inducing function $A$ is convex – which it is. Now, starting from relative smoothness, we see that

$$
\begin{aligned}
\mathcal{L}(\boldsymbol{\theta}^{t+1}) &\leq \mathcal{L}(\boldsymbol{\theta}^t) + \langle \nabla \mathcal{L}(\boldsymbol{\theta}^t), \boldsymbol{\theta}^{t+1} - \boldsymbol{\theta}^t \rangle + D_A(\boldsymbol{\theta}^{t+1}, \boldsymbol{\theta}^t) \\
&= \mathcal{L}(\boldsymbol{\theta}^t) + \langle \nabla \mathcal{L}(\boldsymbol{\theta}^t), \boldsymbol{\theta}^{t+1} - \boldsymbol{\theta}^* + \boldsymbol{\theta}^* - \boldsymbol{\theta}^t \rangle + D_A(\boldsymbol{\theta}^{t+1}, \boldsymbol{\theta}^t) \\
&= \mathcal{L}(\boldsymbol{\theta}^t) + \langle \nabla \mathcal{L}(\boldsymbol{\theta}^t), \boldsymbol{\theta}^{t+1} - \boldsymbol{\theta}^* \rangle + \langle \nabla \mathcal{L}(\boldsymbol{\theta}^t), \boldsymbol{\theta}^* - \boldsymbol{\theta}^t \rangle + D_A(\boldsymbol{\theta}^{t+1}, \boldsymbol{\theta}^t) \\
&\stackrel{i)}{\leq} \mathcal{L}(\boldsymbol{\theta}^*) + \langle \nabla \mathcal{L}(\boldsymbol{\theta}^t), \boldsymbol{\theta}^{t+1} - \boldsymbol{\theta}^* \rangle + D_A(\boldsymbol{\theta}^{t+1}, \boldsymbol{\theta}^t) \\
&\stackrel{(51)}{=} \mathcal{L}(\boldsymbol{\theta}^*) + \langle \nabla A(\boldsymbol{\theta}^t) - \nabla A(\boldsymbol{\theta}^{t+1}), \boldsymbol{\theta}^{t+1} - \boldsymbol{\theta}^* \rangle + D_A(\boldsymbol{\theta}^{t+1}, \boldsymbol{\theta}^t)
\end{aligned}
$$

where i) follows from convexity of $\mathcal{L}(\boldsymbol{\theta})$ on the set $\Theta$. Subsequently, we apply the 3-point lemma, $D_A(\boldsymbol{\theta}^*, \boldsymbol{\theta}^t) = D_A(\boldsymbol{\theta}^*, \boldsymbol{\theta}^{t+1}) + \langle \boldsymbol{\theta}^* - \boldsymbol{\theta}^{t+1}, \nabla A(\boldsymbol{\theta}^{t+1}) - A(\boldsymbol{\theta}^t) \rangle + D_A(\boldsymbol{\theta}^{t+1}, \boldsymbol{\theta}^t)$, and obtain,

$$\mathcal{L}(\boldsymbol{\theta}^{t+1}) \leq \mathcal{L}(\boldsymbol{\theta}^*) + D_A(\boldsymbol{\theta}^*, \boldsymbol{\theta}^t) - D_A(\boldsymbol{\theta}^*, \boldsymbol{\theta}^{t+1}). \tag{53}$$

Finally, the result follows from summing the left and right hand side over $T$ iterations:

$$T(\mathcal{L}(\boldsymbol{\theta}^T) - \mathcal{L}(\boldsymbol{\theta}^*)) \leq \sum_{t=1}^{T} \mathcal{L}(\boldsymbol{\theta}^{t+1}) - \mathcal{L}(\boldsymbol{\theta}^*) \leq \sum_{t=1}^{T} D_A(\boldsymbol{\theta}^*, \boldsymbol{\theta}^t) - D_A(\boldsymbol{\theta}^*, \boldsymbol{\theta}^{t+1}) \leq D_A(\boldsymbol{\theta}^*, \boldsymbol{\theta}^1)$$

**Part 3):** Linear Rate to $\boldsymbol{\theta}^*$.
Given Theorem 5.1, this proof follows from identical arguments to that of Kunstner et al. (2021, Corollary 3) and Lu et al. (2018, Theorem 3.1). We write it below for completeness.

In addition to convexity, we now assume that $\mathcal{L}(\boldsymbol{\theta})$ is $\alpha$-strongly convex relative to $A(\boldsymbol{\theta})$ on the set $\Theta$. Specifically, we have that for any $\boldsymbol{\phi}, \boldsymbol{\theta} \in \Theta$,

$$\mathcal{L}(\boldsymbol{\theta}) \geq \mathcal{L}(\boldsymbol{\phi}) + \langle \nabla \mathcal{L}(\boldsymbol{\phi}), \boldsymbol{\theta} - \boldsymbol{\phi} \rangle + \alpha D_A(\boldsymbol{\theta}, \boldsymbol{\phi}). \tag{54}$$

Using the three point lemma again, we have

$$
\begin{aligned}
D_A(\boldsymbol{\theta}^*, \boldsymbol{\theta}^{t+1}) &= D_A(\boldsymbol{\theta}^*, \boldsymbol{\theta}^t) + \langle \boldsymbol{\theta}^* - \boldsymbol{\theta}^t, \nabla A(\boldsymbol{\theta}^t) - \nabla A(\boldsymbol{\theta}^{t+1}) \rangle + D_A(\boldsymbol{\theta}^t, \boldsymbol{\theta}^{t+1}) \\
&\overset{(51)}{=} D_A(\boldsymbol{\theta}^*, \boldsymbol{\theta}^t) + \langle \nabla \mathcal{L}(\boldsymbol{\theta}^t), \boldsymbol{\theta}^* - \boldsymbol{\theta}^t \rangle + D_A(\boldsymbol{\theta}^t, \boldsymbol{\theta}^{t+1}) \\
&\overset{(54)}{\leq} D_A(\boldsymbol{\theta}^*, \boldsymbol{\theta}^t) + \mathcal{L}(\boldsymbol{\theta}^*) - \mathcal{L}(\boldsymbol{\theta}^t) - \alpha D_A(\boldsymbol{\theta}^*, \boldsymbol{\theta}^t) + D_A(\boldsymbol{\theta}^t, \boldsymbol{\theta}^{t+1}) \\
&= (1 - \alpha) D_A(\boldsymbol{\theta}^*, \boldsymbol{\theta}^t) + \mathcal{L}(\boldsymbol{\theta}^*) - \mathcal{L}(\boldsymbol{\theta}^t) + D_A(\boldsymbol{\theta}^t, \boldsymbol{\theta}^{t+1}) \\
&\overset{(52)}{\leq} (1 - \alpha) D_A(\boldsymbol{\theta}^*, \boldsymbol{\theta}^t) + \mathcal{L}(\boldsymbol{\theta}^*) - \mathcal{L}(\boldsymbol{\theta}^t) + \mathcal{L}(\boldsymbol{\theta}^t) - \mathcal{L}(\boldsymbol{\theta}^{t+1}) \\
&\leq (1 - \alpha) D_A(\boldsymbol{\theta}^*, \boldsymbol{\theta}^t) \\
&\leq (1 - \alpha)^T D_A(\boldsymbol{\theta}^*, \boldsymbol{\theta}^1).
\end{aligned}
$$

Finally, from (53), we see that

$$
\begin{aligned}
\mathcal{L}(\boldsymbol{\theta}^{t+1}) - \mathcal{L}(\boldsymbol{\theta}^*) &\leq D_A(\boldsymbol{\theta}^*, \boldsymbol{\theta}^t) - D_A(\boldsymbol{\theta}^*, \boldsymbol{\theta}^{t+1}) \\
&\leq (1 - \alpha)^T D_A(\boldsymbol{\theta}^*, \boldsymbol{\theta}^1) - D_A(\boldsymbol{\theta}^*, \boldsymbol{\theta}^{t+1}) \\
&\leq (1 - \alpha)^T D_A(\boldsymbol{\theta}^*, \boldsymbol{\theta}^1).
\end{aligned}
$$

$\square$

### B.5. Satisfiability of Conditions from Corollary B.1

The above result raises an important question: when does a locally convex or relatively strongly convex region of $\mathcal{L}(\boldsymbol{\theta})$ containing $\boldsymbol{\theta}^*$ exist? Interestingly, this is closely tied to the Signal-to-Noise Ratio (SNR). Before exploring this connection, we first introduce the concept of the Missing Information Matrix (MIM) introduced in (Orchard & Woodbury, 1972).

The MIM relates the level of information the pair $(\boldsymbol{x}, y)$ holds about the latent expert label $z$ given parameters $\boldsymbol{\theta}$, and it is formally defined as

$$
\boldsymbol{M}(\theta) = \boldsymbol{I}_{\boldsymbol{x},z,y|\boldsymbol{\theta}}^{-1} \boldsymbol{I}_{z|\boldsymbol{x},y,\boldsymbol{\theta}}. \tag{55}
$$

Here, $\boldsymbol{I}_{\boldsymbol{x},z,y|\boldsymbol{\theta}}$ is the Fisher information matrix of the complete data distribution and $\boldsymbol{I}_{z|\boldsymbol{x},y,\boldsymbol{\theta}}$ is the Fisher information matrix of the conditional distribution of the latent unobserved variable given the observed ones, denoted by

$$
\begin{aligned}
\boldsymbol{I}_{\boldsymbol{x},z,y|\boldsymbol{\theta}} &:= -\mathbb{E}_{\boldsymbol{X},Y,Z|\boldsymbol{\theta}} \left[ \frac{\partial^2}{\partial \boldsymbol{\theta}^2} \log p(\boldsymbol{x}, z, y; \boldsymbol{\theta}) \right] \\
&= \nabla^2 Q(\boldsymbol{\phi}|\boldsymbol{\theta})|_{\boldsymbol{\phi}=\boldsymbol{\theta}} = \nabla^2 A(\boldsymbol{\theta}) \\
\boldsymbol{I}_{z|\boldsymbol{x},y,\boldsymbol{\theta}} &:= -\mathbb{E}_{\boldsymbol{X},Y} \mathbb{E}_{Z|\boldsymbol{x},y,\boldsymbol{\theta}} \left[ \frac{\partial^2}{\partial \boldsymbol{\theta}^2} \log P(z|\boldsymbol{x}, y; \boldsymbol{\theta}) \right] \\
&= \nabla^2 H(\boldsymbol{\phi}|\boldsymbol{\theta})|_{\boldsymbol{\phi}=\boldsymbol{\theta}}.
\end{aligned}
$$

Thus, it also holds that the MIM is a function of $A(\boldsymbol{\theta})$ and $H(\boldsymbol{\phi}|\boldsymbol{\theta})$, i.e.,

$$
\boldsymbol{M}(\boldsymbol{\theta}) = \nabla^2 A(\boldsymbol{\theta})^{-1} \nabla^2 H(\boldsymbol{\phi}|\boldsymbol{\theta})|_{\boldsymbol{\phi}=\boldsymbol{\theta}}. \tag{56}
$$

Due to the pairwise independence of $\boldsymbol{X}$ and its rotational invariance, there exists an orthonormal matrix $R$ such that $\Delta := R\boldsymbol{I}_{\boldsymbol{x},z,y|\boldsymbol{\theta}}^{-1} R^\top$ is positive semi-definite and diagonal and $J := R\boldsymbol{I}_{z|\boldsymbol{x},y,\theta} R^\top$ is symmetric positive semi-definite (see Lemma B.3). As such, the MIM in (55) is also symmetric and positive semi-definite: $M(\boldsymbol{\theta}) = R^\top \Delta R R^\top J R = R^\top \Delta J R$. Note that the MIM quantifies the difficulty of estimating parameters when only $\boldsymbol{x}, y$ are observed. To understand its significance, consider the following: large eigenvalues of $\boldsymbol{M}$ indicate that $\boldsymbol{x}, y$ contain little information about the true value of the latent variable $z$, making estimation more difficult. Conversely, small eigenvalues suggest that $\boldsymbol{x}, y$ provide enough information to effectively constrain the possible values of $z$. Thus, the MIM can be seen as analogous to the Signal-to-Noise Ratio (SNR).

In the theorem below, we show how the eigenvalues of $\boldsymbol{M}(\boldsymbol{\theta})$ are related to the satisfiability of the conditions for Corollary B.1 regarding the relative strong convexity of $\mathcal{L}(\boldsymbol{\theta})$ with respect to the mirror map $A(\boldsymbol{\theta})$.

**Theorem B.2.** *For SymMoLinE and SymMoLogE and their respective mirror mappings (31) and (32), the objective $\mathcal{L}(\boldsymbol{\theta})$ is $\alpha$-strongly convex relative to the mirror map $A(\boldsymbol{\theta})$ on the convex set $\Theta$ if and only if*

$$\lambda_{\max}(\boldsymbol{M}(\boldsymbol{\theta})) \leq (1 - \alpha) \text{ for all } \boldsymbol{\theta} \in \Theta. \tag{57}$$

*Proof.* Recall that $\mathcal{L}(\boldsymbol{\theta})$ is strongly convex relative to $A(\boldsymbol{\theta})$ on $\Theta$ if for all $\boldsymbol{\theta}, \boldsymbol{\phi} \in \Theta$, it holds that

$$\mathcal{L}(\boldsymbol{\phi}) \geq \mathcal{L}(\boldsymbol{\theta}) + \langle \nabla\mathcal{L}(\boldsymbol{\theta}), \boldsymbol{\phi} - \boldsymbol{\theta} \rangle + \alpha D_A(\boldsymbol{\phi}, \boldsymbol{\theta}).$$

For $\mathcal{L}(\boldsymbol{\theta})$ and $A(\boldsymbol{\theta})$ twice continuously differentiable, it was shown by (Lu et al., 2018) that this is equivalent to the following bound on the Hessian:

$$\nabla^2\mathcal{L}(\boldsymbol{\theta}) \succeq \alpha\nabla^2 A(\boldsymbol{\theta}).$$

Now, using $\mathcal{L}(\boldsymbol{\phi}) = Q(\boldsymbol{\phi}|\boldsymbol{\theta}) - H(\boldsymbol{\phi}|\boldsymbol{\theta})$, we see that

$$\begin{aligned}
\nabla^2\mathcal{L}(\boldsymbol{\theta}) &= \nabla^2(Q(\boldsymbol{\phi}|\boldsymbol{\theta}) - H(\boldsymbol{\phi}|\boldsymbol{\theta}))|_{\boldsymbol{\phi}=\boldsymbol{\theta}} \\
&= \nabla^2 Q(\boldsymbol{\phi}|\boldsymbol{\theta})|_{\boldsymbol{\phi}=\boldsymbol{\theta}} - \nabla^2 H(\boldsymbol{\phi}|\boldsymbol{\theta})|_{\boldsymbol{\phi}=\boldsymbol{\theta}} \\
&= \nabla^2 A(\boldsymbol{\theta}) - \nabla^2 H(\boldsymbol{\phi}|\boldsymbol{\theta})|_{\boldsymbol{\phi}=\boldsymbol{\theta}}
\end{aligned}$$

Therefore, since $\nabla^2 A(\boldsymbol{\theta})$ is symmetric positive definite (proven in Appendix B.1), our condition simplifies to

$$(1 - \alpha)\nabla^2 A(\boldsymbol{\theta}) \succeq \nabla^2 H(\boldsymbol{\phi}|\boldsymbol{\theta})|_{\boldsymbol{\phi}=\boldsymbol{\theta}}$$
$$\iff (1 - \alpha)\boldsymbol{I}_{2d} \succeq \nabla^2 A(\boldsymbol{\theta})^{-1}\nabla^2 H(\boldsymbol{\phi}|\boldsymbol{\theta})|_{\boldsymbol{\phi}=\boldsymbol{\theta}} = \boldsymbol{M}(\boldsymbol{\theta}).$$

Finally, for $\boldsymbol{x}$ from a unit spherical Gaussian distribution, we know that $\boldsymbol{M}(\boldsymbol{\theta})$ is symmetric positive-definite (see Lemma B.3). As a result, the above inequality is equivalent to the following bound on the eigenvalues of the MIM:

$$1 - \alpha \geq \lambda_{\max}(\boldsymbol{M}(\boldsymbol{\theta})).$$

$\square$

We now provide the simple Lemma that the MIM is a symmetric matrix for SymMoLinE and SymMoLogE.

**Lemma B.3** ($\boldsymbol{M}(\boldsymbol{\theta})$ is symmetric). *For SymMoLinE and SymMoLogE, the MIM is a symmetric matrix, i.e. $\boldsymbol{M}(\boldsymbol{\theta}) = \boldsymbol{M}(\boldsymbol{\theta})^{\top}$.*

*Proof.* Recall the assumption that $\boldsymbol{x}$ is sampled from a unit spherical Gaussian distribution: $\boldsymbol{x} \sim \mathcal{N}(\boldsymbol{0}, \boldsymbol{I}_d)$. As such, for any orthonormal $d \times d$ matrix $\boldsymbol{R}$, we know that $\boldsymbol{RX} \sim \mathcal{N}(\boldsymbol{0}, \boldsymbol{I}_d)$; this is called rotational invariance of the Gaussian distribution. Thus, consider orthonormal $\boldsymbol{R_u} \in \mathbb{R}^{d \times d}$, such that $R_u u = e_1\|\boldsymbol{u}\|_2$ where $e_j \in \mathbb{R}^d$ is the $\boldsymbol{0}$ vector with a 1 at index $j$. Now, we can observe that for any $\boldsymbol{w} \in \mathbb{R}^d$, $\mathbb{E}_{\boldsymbol{X}}\left[\boldsymbol{xx}^{\top}\frac{e^{\boldsymbol{x}^{\top}\boldsymbol{u}}}{(1+e^{\boldsymbol{x}^{\top}\boldsymbol{u}})^2}\right]$, is diagonalizable by an orthonormal matrix $\boldsymbol{R}$:

$$\begin{aligned}
\boldsymbol{R_u}\left(\mathbb{E}_{\boldsymbol{X}}\left[\boldsymbol{xx}^{\top}\frac{e^{\boldsymbol{x}^{\top}\boldsymbol{u}}}{(1+e^{\boldsymbol{x}^{\top}\boldsymbol{u}})^2}\right]\right)\boldsymbol{R_u^{\top}} &= \mathbb{E}_{\boldsymbol{X}}\left[\boldsymbol{R_u}\boldsymbol{xx}^{\top}\boldsymbol{R_u^{\top}}\frac{e^{\boldsymbol{x}^{\top}R_u^{\top}R_u u}}{(1+e^{\boldsymbol{x}^{\top}R_u^{\top}R_u u})^2}\right] \\
&= \mathbb{E}_{\boldsymbol{R_u x}}\left[\tilde{x}\tilde{x}^{\top}\frac{e^{\tilde{x}^{\top}e_1\|\boldsymbol{u}\|_2}}{(1+e^{\tilde{x}^{\top}e_1\|\boldsymbol{u}\|_2})^2}\right] \\
&= \begin{pmatrix} \mathbb{E}_{\boldsymbol{R_u x}}\left[\tilde{x}_1^2\frac{e^{\tilde{x}_1\|\boldsymbol{u}\|_2}}{(1+e^{\tilde{x}_1\|\boldsymbol{u}\|_2})^2}\right] & \boldsymbol{0} & \boldsymbol{0} \\ \boldsymbol{0} & \ldots & \boldsymbol{0} \\ \boldsymbol{0} & \boldsymbol{0} & \mathbb{E}_{\boldsymbol{R_u x}}\left[\tilde{x}_d^2\frac{e^{\tilde{x}_1\|\boldsymbol{u}\|_2}}{(1+e^{\tilde{x}_1\|\boldsymbol{u}\|_2})^2}\right] \end{pmatrix}
\end{aligned}$$

Where in the above, 1) $\boldsymbol{R}^{\top}\boldsymbol{R} = I_d$ since $\boldsymbol{R}^{\top} = \boldsymbol{R}^{-1}$ for orthonormal matrices and 2) non diagonal elements evaluate to 0 because for all $i \neq j$, the 0 mean random variables $\tilde{X}_j$ is independent from $\tilde{X}_i$.

Finally, we put the above together to show $M(\boldsymbol{\theta})$ is symmetric. We define the block diagonal orthonormal matrix $R_M$ as

$$R_M := \begin{pmatrix} R_w & 0 \\ 0 & R_\beta \end{pmatrix}.$$

From the above, it follows that both for SymMoLinE and SymMoLogE, $R_M \nabla^2 A(\boldsymbol{\theta})^{-1} R_M^\top$ is a diagonal matrix. We can now use this change of basis matrix to show the MIM is a symmetric matrix:

$$\begin{aligned}
M(\boldsymbol{\theta}) &= \nabla^2 A(\boldsymbol{\theta})^{-1} \nabla^2 H(\boldsymbol{\theta}|\boldsymbol{\theta}) \\
&= R_M^\top \left( R_M \nabla^2 A(\boldsymbol{\theta})^{-1} R_M^\top \right) \left( R_M \nabla^2 H(\boldsymbol{\theta}|\boldsymbol{\theta}) R_M^\top \right) R_M \\
&= R_M^\top \left( R_M \nabla^2 H(\boldsymbol{\theta}|\boldsymbol{\theta}) R_M^\top \right) \left( R_M \nabla^2 A(\boldsymbol{\theta})^{-1} R_M^\top \right) R_M \\
&= \nabla^2 H(\boldsymbol{\theta}|\boldsymbol{\theta})^\top (\nabla^2 A(\boldsymbol{\theta})^{-1})^\top \\
&= M(\boldsymbol{\theta})^\top.
\end{aligned}$$

$\square$

## B.6. Correspondence Between the MIM and SNR for SymMoLinE and SymMoLogE

The above result states if $M(\boldsymbol{\theta}) \prec I_{2d}$ for all $\boldsymbol{\theta} \in \Omega$ and $\boldsymbol{\theta}^* \in \Omega$, then the EM updates will converge linearly to $\boldsymbol{\theta}^*$. This offers a unified framework for analyzing EM for MoE, linking the rate of convergence to a classical statistical metric. To determine whether EM achieves linear or sub-linear convergence, we need to understand the behavior of $M(\boldsymbol{\theta})$ and $M(\boldsymbol{\theta}^*)$, which indicates the existence and size of the local region where EM enjoys such convergence.

**Theorem B.4.** *For SymMoLinE, the eigenvalues of $I_{\boldsymbol{x},z,y|\boldsymbol{\theta}}^{-1}$ belong to the set*

$$\lambda \left( I_{\boldsymbol{x},z,y|\boldsymbol{\theta}}^{-1} \right) = \{\Theta(\|\boldsymbol{w}\|_2^3), \Theta(\|\boldsymbol{w}\|_2), 1\} \tag{58}$$

*and $I_{z|\boldsymbol{x},y,\boldsymbol{\theta}}$, is given as the expectation over $(\boldsymbol{X}, Y)$ of a function that is decreasing as a function of $\|\boldsymbol{\theta}\|_2$:*

$$I_{z|\boldsymbol{x},y,\boldsymbol{\theta}} = \mathbb{E}_{\boldsymbol{X},Y} \left[ \frac{\exp\left\langle \begin{bmatrix} \boldsymbol{x} \\ 2y\boldsymbol{x} \end{bmatrix}, \boldsymbol{\theta} \right\rangle \begin{bmatrix} \boldsymbol{x} \\ 2y\boldsymbol{x} \end{bmatrix} \begin{bmatrix} \boldsymbol{x} \\ 2y\boldsymbol{x} \end{bmatrix}^\top}{\left( 1 + \exp\left\langle \begin{bmatrix} \boldsymbol{x} \\ 2y\boldsymbol{x} \end{bmatrix}, \boldsymbol{\theta} \right\rangle \right)^2} \right]. \tag{59}$$

*Similarly, For SymMoLogE, the eigenvalues of $I_{\boldsymbol{x},z,y|\boldsymbol{\theta}}^{-1}$ belong to the set*

$$\lambda \left( I_{\boldsymbol{x},z,y|\boldsymbol{\theta}}^{-1} \right) = \{\Theta(\|\boldsymbol{w}\|_2^3), \Theta(\|\boldsymbol{w}\|_2), \Theta(\|\boldsymbol{\beta}\|_2^3), \Theta(\|\boldsymbol{\beta}\|_2)\} \tag{60}$$

*and $I_{z|\boldsymbol{x},y,\boldsymbol{\theta}}$ is given as the expectation over $(\boldsymbol{X}, Y)$ of a function that is decreasing as a function of $\|\boldsymbol{\theta}\|_2$:*

$$I_{z|\boldsymbol{x},y,\boldsymbol{\theta}} = \mathbb{E}_{\boldsymbol{X},Y} \left[ \frac{\exp\left\langle \begin{bmatrix} \boldsymbol{x} \\ y\boldsymbol{x} \end{bmatrix}, \boldsymbol{\theta} \right\rangle \begin{bmatrix} \boldsymbol{x} \\ y\boldsymbol{x} \end{bmatrix} \begin{bmatrix} \boldsymbol{x} \\ y\boldsymbol{x} \end{bmatrix}^\top}{\left( 1 + \exp\left\langle \begin{bmatrix} \boldsymbol{x} \\ y\boldsymbol{x} \end{bmatrix}, \boldsymbol{\theta} \right\rangle \right)^2} \right]. \tag{61}$$

*Proof.* We divide the proof into two parts. In the first part, we consider the SymMoLinE setting and, in the second part, we consider the SymMoLogE setting.

**Part a):** SymMoLinE.
For $I_{\boldsymbol{x},z,y|\boldsymbol{\theta}}$, recall that is has the following form:

$$I_{\boldsymbol{x},y,z|\boldsymbol{\theta}} = \begin{pmatrix} \mathbb{E}_{\boldsymbol{X}} \left[ \boldsymbol{x}\boldsymbol{x}^\top \frac{e^{\boldsymbol{x}^\top \boldsymbol{w}}}{(1+e^{\boldsymbol{x}^\top \boldsymbol{w}})^2} \right] & 0 \\ 0 & I_d \end{pmatrix}.$$

By Lemma B.5, we see that $I_{\boldsymbol{x},z,y|\boldsymbol{\theta}}$ can be diagonalized into the following form:

$$I_{\boldsymbol{x},y,z|\boldsymbol{\theta}} = R_M \begin{pmatrix} \lambda_1 & 0 & 0 & 0 \\ 0 & \cdots & 0 & 0 \\ 0 & 0 & \lambda_2 & 0 \\ 0 & 0 & 0 & I_d \end{pmatrix} R_M^\top$$

where $R_M$ is an orthonormal rotation matrix and $\lambda_1 = \Theta\left(\frac{1}{\|\boldsymbol{w}\|_2^3}\right)$ and $\lambda_2 = \Theta\left(\frac{1}{\|\boldsymbol{w}\|_2}\right)$. Therefore, $I_{\boldsymbol{x},z,y|\boldsymbol{\theta}}$ has three eigenvalues given as $\left\{\Theta\left(\frac{1}{\|\boldsymbol{w}\|_2^3}\right), \Theta\left(\frac{1}{\|\boldsymbol{w}\|_2}\right), 1\right\}$. It follows that $I_{\boldsymbol{x},z,y|\boldsymbol{\theta}}^{-1}$ has the form

$$I_{\boldsymbol{x},y,z|\boldsymbol{\theta}}^{-1} = R_M^\top \begin{pmatrix} 1/\lambda_1 & 0 & 0 & 0 \\ 0 & \cdots & 0 & 0 \\ 0 & 0 & 1/\lambda_2 & 0 \\ 0 & 0 & 0 & I_d \end{pmatrix} R_M.$$

Therefore, $I_{\boldsymbol{x},y,z|\boldsymbol{\theta}}^{-1}$ has three eigenvalues given as $\left\{\Theta\left(\|\boldsymbol{w}\|_2^3\right), \Theta\left(\|\boldsymbol{w}\|_2\right), 1\right\}$.

Now, for $I_{z|\boldsymbol{x},y,\boldsymbol{\theta}}$, we first derive a more compact form for the conditional distribution of the latent variable, $p(z|\boldsymbol{x}, y; \boldsymbol{\theta})$. From simple Bayes rule and algebraic manipulation, we see that

$$p(z|\boldsymbol{x}, y; \boldsymbol{\theta}) = \frac{p(y|\boldsymbol{x}, z; \boldsymbol{\theta})p(z|\boldsymbol{x}; \boldsymbol{\theta})p(\boldsymbol{x})}{p(\boldsymbol{x}, y; \boldsymbol{\theta})}$$

$$= \frac{\frac{1}{\sqrt{2\pi}} \exp\left\{-\frac{(y-z\boldsymbol{x}^\top\boldsymbol{\beta})^2}{2}\right\} \frac{\exp\{\frac{z+1}{2}\boldsymbol{x}^\top\boldsymbol{w}\}}{1+e^{\boldsymbol{x}^\top\boldsymbol{w}}}}{\frac{1}{\sqrt{2\pi}} \exp\left\{-\frac{(y-\boldsymbol{x}^\top\boldsymbol{\beta})^2}{2}\right\} \frac{\exp\{\boldsymbol{x}^\top\boldsymbol{w}\}}{1+e^{\boldsymbol{x}^\top\boldsymbol{w}}} + \frac{1}{\sqrt{2\pi}} \exp\left\{-\frac{(y+\boldsymbol{x}^\top\boldsymbol{\beta})^2}{2}\right\} \frac{1}{1+e^{\boldsymbol{x}^\top\boldsymbol{w}}}}$$

$$= \frac{\exp\left\{-\frac{(y-z\boldsymbol{x}^\top\boldsymbol{\beta})^2}{2} + \frac{z+1}{2}\boldsymbol{x}^\top\boldsymbol{w}\right\}}{\exp\left\{-\frac{(y-\boldsymbol{x}^\top\boldsymbol{\beta})^2}{2} + \boldsymbol{x}^\top\boldsymbol{w}\right\} + \exp\left\{-\frac{(y+\boldsymbol{x}^\top\boldsymbol{\beta})^2}{2}\right\}}$$

$$= \frac{\exp\left\{zy\boldsymbol{x}^\top\boldsymbol{\beta} + \frac{z+1}{2}\boldsymbol{x}^\top\boldsymbol{w}\right\}}{\exp\left\{y\boldsymbol{x}^\top\boldsymbol{\beta} + \boldsymbol{x}^\top\boldsymbol{w}\right\} + \exp\left\{-y\boldsymbol{x}^\top\boldsymbol{\beta}\right\}}$$

$$= \frac{\exp\left\{\frac{z+1}{2}(2y\boldsymbol{x}^\top\boldsymbol{\beta} + \boldsymbol{x}^\top\boldsymbol{w})\right\}}{\exp\left\{2y\boldsymbol{x}^\top\boldsymbol{\beta} + \boldsymbol{x}^\top\boldsymbol{w}\right\} + 1}$$

$$= \frac{\exp\left\{\frac{z+1}{2}\left\langle\begin{bmatrix}\boldsymbol{x}\\2y\boldsymbol{x}\end{bmatrix}, \begin{bmatrix}\boldsymbol{w}\\\boldsymbol{\beta}\end{bmatrix}\right\rangle\right\}}{\exp\left\{\left\langle\begin{bmatrix}\boldsymbol{x}\\2y\boldsymbol{x}\end{bmatrix}, \begin{bmatrix}\boldsymbol{w}\\\boldsymbol{\beta}\end{bmatrix}\right\rangle\right\} + 1}$$

Now that we have this simplified form, we are able to derive (59) for $I_{z|\boldsymbol{x},y,\boldsymbol{\theta}}$:

$$I_{z|\boldsymbol{x},y,\boldsymbol{\theta}} = -\mathbb{E}_{\boldsymbol{X},Y}\mathbb{E}_{Z|\boldsymbol{x},y,\boldsymbol{\theta}}\left[\frac{\partial^2}{\partial\boldsymbol{\theta}^2}\log p(z|\boldsymbol{x}, y; \boldsymbol{\theta})\right]$$

$$= -\mathbb{E}_{\boldsymbol{X},Y}\mathbb{E}_{Z|\boldsymbol{x},y,\boldsymbol{\theta}}\left[\frac{\partial^2}{\partial\boldsymbol{\theta}^2}\left(\frac{z+1}{2}\left\langle\begin{pmatrix}\boldsymbol{x}\\2y\boldsymbol{x}\end{pmatrix}, \boldsymbol{\theta}\right\rangle - \log\left(1 + \exp\left\langle\begin{bmatrix}\boldsymbol{x}\\2y\boldsymbol{x}\end{bmatrix}, \boldsymbol{\theta}\right\rangle\right)\right)\right]$$

$$= \mathbb{E}_{\boldsymbol{X},Y}\mathbb{E}_{Z|\boldsymbol{x},y,\boldsymbol{\theta}}\left[\frac{\partial^2}{\partial\boldsymbol{\theta}^2}\log\left(1 + \exp\left\langle\begin{bmatrix}\boldsymbol{x}\\2y\boldsymbol{x}\end{bmatrix}, \boldsymbol{\theta}\right\rangle\right)\right]$$

$$= \mathbb{E}_{\boldsymbol{X},Y}\left[\frac{\partial^2}{\partial\boldsymbol{\theta}^2}\log\left(1 + \exp\left\langle\begin{bmatrix}\boldsymbol{x}\\2y\boldsymbol{x}\end{bmatrix}, \boldsymbol{\theta}\right\rangle\right)\right]$$

$$= \mathbb{E}_{\boldsymbol{X},Y}\left[\frac{\exp\left\langle\begin{bmatrix}\boldsymbol{x}\\2y\boldsymbol{x}\end{bmatrix}, \boldsymbol{\theta}\right\rangle\begin{bmatrix}\boldsymbol{x}\\2y\boldsymbol{x}\end{bmatrix}\begin{bmatrix}\boldsymbol{x}\\2y\boldsymbol{x}\end{bmatrix}^\top}{\left(1 + \exp\left\langle\begin{bmatrix}\boldsymbol{x}\\2y\boldsymbol{x}\end{bmatrix}, \boldsymbol{\theta}\right\rangle\right)^2}\right].$$

This expression depends only on the random variable $(\boldsymbol{X}, Y)$ and the parameter iterate $\boldsymbol{\theta}$.

**Part b):** SymMoLogE.

Recall that for SymMoLogE, $\boldsymbol{I}_{\boldsymbol{x},z,y|\boldsymbol{\theta}}$ has the following form:

$$
\boldsymbol{I}_{\boldsymbol{x},z,y|\boldsymbol{\theta}} = \begin{pmatrix} \mathbb{E}_{\boldsymbol{X}}\left[\boldsymbol{x}\boldsymbol{x}^\top \frac{e^{\boldsymbol{x}^\top \boldsymbol{w}}}{\left(1+e^{\boldsymbol{x}^\top \boldsymbol{w}}\right)^2}\right] & \mathbf{0} \\ \mathbf{0} & \mathbb{E}_{\boldsymbol{X}}\left[\boldsymbol{x}\boldsymbol{x}^\top \frac{e^{\boldsymbol{x}^\top \boldsymbol{\beta}}}{\left(1+e^{\boldsymbol{x}^\top \boldsymbol{\beta}}\right)^2}\right] \end{pmatrix}.
$$

By Lemma B.5, we see that $\boldsymbol{I}_{\boldsymbol{x},z,y|\boldsymbol{\theta}}$ can be diagonalized into the following form:

$$
\boldsymbol{I}_{\boldsymbol{x},z,y|\boldsymbol{\theta}} = R_M \begin{pmatrix} \lambda_1 & \mathbf{0} & \mathbf{0} & \mathbf{0} & \mathbf{0} \\ \mathbf{0} & \lambda_2 & \mathbf{0} & \mathbf{0} & \mathbf{0} \\ \mathbf{0} & \mathbf{0} & \ldots & \mathbf{0} & \mathbf{0} \\ \mathbf{0} & \mathbf{0} & \mathbf{0} & \lambda_3 & \mathbf{0} \\ \mathbf{0} & \mathbf{0} & \mathbf{0} & \mathbf{0} & \lambda_4 \end{pmatrix} R_M^\top
$$

where $R_M$ is an orthonormal rotation matrix and $\lambda_1 = \Theta\left(\frac{1}{\|\boldsymbol{w}\|_2^3}\right)$, $\lambda_2 = \Theta\left(\frac{1}{\|\boldsymbol{w}\|_2}\right)$, $\lambda_3 = \Theta\left(\frac{1}{\|\boldsymbol{\beta}\|_2^3}\right)$, and $\lambda_4 = \Theta\left(\frac{1}{\|\boldsymbol{\beta}\|_2}\right)$.
Therefore, $\boldsymbol{I}_{\boldsymbol{x},z,y|\boldsymbol{\theta}}$ has four eigenvalues given as
$\left\{\Theta\left(\frac{1}{\|\boldsymbol{w}\|_2^3}\right), \Theta\left(\frac{1}{\|\boldsymbol{w}\|_2}\right), \Theta\left(\frac{1}{\|\boldsymbol{\beta}\|_2^3}\right), \Theta\left(\frac{1}{\|\boldsymbol{\beta}\|_2}\right)\right\}$. It follows that $\boldsymbol{I}_{\boldsymbol{x},z,y|\boldsymbol{\theta}}^{-1}$ has the form

$$
\boldsymbol{I}_{\boldsymbol{x},z,y|\boldsymbol{\theta}}^{-1} = R_M \begin{pmatrix} 1/\lambda_1 & \mathbf{0} & \mathbf{0} & \mathbf{0} & \mathbf{0} \\ \mathbf{0} & 1/\lambda_2 & \mathbf{0} & \mathbf{0} & \mathbf{0} \\ \mathbf{0} & \mathbf{0} & \ldots & \mathbf{0} & \mathbf{0} \\ \mathbf{0} & \mathbf{0} & \mathbf{0} & 1/\lambda_3 & \mathbf{0} \\ \mathbf{0} & \mathbf{0} & \mathbf{0} & \mathbf{0} & 1/\lambda_4 \end{pmatrix} R_M^\top.
$$

Therefore, $\boldsymbol{I}_{\boldsymbol{x},z,y|\boldsymbol{\theta}}^{-1}$ has four eigenvalues given as $\left\{\Theta\left(\|\boldsymbol{w}\|_2^3\right), \Theta\left(\|\boldsymbol{w}\|_2\right), \Theta\left(\|\boldsymbol{\beta}\|_2^3\right), \Theta\left(\|\boldsymbol{\beta}\|_2\right)\right\}$.

Now, for $I_{z|\boldsymbol{x},y,\boldsymbol{\theta}}$, we first derive a more compact form for the conditional distribution of the latent variable, $P(z|\boldsymbol{x}, y; \boldsymbol{\theta})$. From simple Bayes rule and algebraic manipulation, we see that

$$
\begin{aligned}
P(z|\boldsymbol{x}, y; \boldsymbol{\theta}) &= \frac{P(y|\boldsymbol{x}, z; \boldsymbol{\theta})P(z|\boldsymbol{x}; \boldsymbol{\theta})p(\boldsymbol{x})}{p(\boldsymbol{x}, y; \boldsymbol{\theta})} \\
&= \frac{\frac{\exp\{\frac{yz+1}{2}\boldsymbol{x}^\top \boldsymbol{\beta}\}}{1+e^{\boldsymbol{x}^\top \boldsymbol{\beta}}} \frac{\exp\{\frac{z+1}{2}\boldsymbol{x}^\top \boldsymbol{w}\}}{1+e^{\boldsymbol{x}^\top \boldsymbol{w}}}}{\frac{\exp\{\frac{y+1}{2}\boldsymbol{x}^\top \boldsymbol{\beta}\}}{1+e^{\boldsymbol{x}^\top \boldsymbol{\beta}}} \frac{\exp\{\boldsymbol{x}^\top \boldsymbol{w}\}}{1+e^{\boldsymbol{x}^\top \boldsymbol{w}}} + \frac{\exp\{\frac{-y+1}{2}\boldsymbol{x}^\top \boldsymbol{\beta}\}}{1+e^{\boldsymbol{x}^\top \boldsymbol{\beta}}} \frac{1}{1+e^{\boldsymbol{x}^\top \boldsymbol{w}}}} \\
&= \frac{\exp\left\{\frac{yz+1}{2}\boldsymbol{x}^\top \boldsymbol{\beta} + \frac{z+1}{2}\boldsymbol{x}^\top \boldsymbol{w}\right\}}{\exp\left\{\frac{y+1}{2}\boldsymbol{x}^\top \boldsymbol{\beta} + \boldsymbol{x}^\top \boldsymbol{w}\right\} + \exp\left\{\frac{-y+1}{2}\boldsymbol{x}^\top \boldsymbol{\beta}\right\}} \\
&= \frac{\exp\left\{\frac{z+1}{2}\left(y\boldsymbol{x}^\top \boldsymbol{\beta} + \boldsymbol{x}^\top \boldsymbol{w}\right)\right\}}{\exp\left\{y\boldsymbol{x}^\top \boldsymbol{\beta} + \boldsymbol{x}^\top \boldsymbol{w}\right\} + 1} \\
&= \frac{\exp\left\{\frac{z+1}{2}\left\langle \begin{bmatrix} \boldsymbol{x} \\ y\boldsymbol{x} \end{bmatrix}, \begin{bmatrix} \boldsymbol{w} \\ \boldsymbol{\beta} \end{bmatrix} \right\rangle\right\}}{\exp\left\{\left\langle \begin{bmatrix} \boldsymbol{x} \\ y\boldsymbol{x} \end{bmatrix}, \begin{bmatrix} \boldsymbol{w} \\ \boldsymbol{\beta} \end{bmatrix} \right\rangle\right\} + 1}
\end{aligned}
$$

Now that we have this simplified form, we are able to derive (61) for $I_{z|\boldsymbol{x},y,\boldsymbol{\theta}}$:

$$
\begin{aligned}
I_{z|\boldsymbol{x},y,\boldsymbol{\theta}} &= -\mathbb{E}_{\boldsymbol{X},Y}\mathbb{E}_{Z|\boldsymbol{x},y,\boldsymbol{\theta}}\left[\frac{\partial^2}{\partial\boldsymbol{\theta}^2}\log P(z|\boldsymbol{x},y;\boldsymbol{\theta})\right] \\
&= -\mathbb{E}_{\boldsymbol{X},Y}\mathbb{E}_{Z|\boldsymbol{x},y,\boldsymbol{\theta}}\left[\frac{\partial^2}{\partial\boldsymbol{\theta}^2}\left(\frac{z+1}{2}\left\langle\begin{pmatrix}\boldsymbol{x}\\y\boldsymbol{x}\end{pmatrix},\boldsymbol{\theta}\right\rangle - \log\left(1+\exp\left\langle\begin{bmatrix}\boldsymbol{x}\\y\boldsymbol{x}\end{bmatrix},\boldsymbol{\theta}\right\rangle\right)\right)\right] \\
&= \mathbb{E}_{\boldsymbol{X},Y}\mathbb{E}_{Z|\boldsymbol{x},y,\boldsymbol{\theta}}\left[\frac{\partial^2}{\partial\boldsymbol{\theta}^2}\log\left(1+\exp\left\langle\begin{bmatrix}\boldsymbol{x}\\y\boldsymbol{x}\end{bmatrix},\boldsymbol{\theta}\right\rangle\right)\right] \\
&= \mathbb{E}_{\boldsymbol{X},Y}\left[\frac{\partial^2}{\partial\boldsymbol{\theta}^2}\log\left(1+\exp\left\langle\begin{bmatrix}\boldsymbol{x}\\y\boldsymbol{x}\end{bmatrix},\boldsymbol{\theta}\right\rangle\right)\right] \\
&= \mathbb{E}_{\boldsymbol{X},Y}\left[\frac{\exp\left\langle\begin{bmatrix}\boldsymbol{x}\\y\boldsymbol{x}\end{bmatrix},\boldsymbol{\theta}\right\rangle\begin{bmatrix}\boldsymbol{x}\\y\boldsymbol{x}\end{bmatrix}\begin{bmatrix}\boldsymbol{x}\\y\boldsymbol{x}\end{bmatrix}^\top}{\left(1+\exp\left\langle\begin{bmatrix}\boldsymbol{x}\\y\boldsymbol{x}\end{bmatrix},\boldsymbol{\theta}\right\rangle\right)^2}\right]
\end{aligned}
$$

This expression depends only on the random variable $(\boldsymbol{X},Y)$ and the parameter iterate $\boldsymbol{\theta}$. $\qquad\square$

**Lemma B.5.** *For $\boldsymbol{x} \sim \mathcal{N}(\boldsymbol{0},\boldsymbol{I}_d)$ and $\boldsymbol{u} \in \mathbb{R}^d$ and $\|\boldsymbol{u}\|_2 \geq \sqrt{2}$, the symmetric positive definite matrix*

$$
\mathbb{E}_{\boldsymbol{X}}\left[\boldsymbol{x}\boldsymbol{x}^\top\frac{e^{\boldsymbol{x}^\top\boldsymbol{u}}}{\left(1+e^{\boldsymbol{x}^\top\boldsymbol{u}}\right)^2}\right] \tag{62}
$$

*is diagonalizable by an orthonormal matrix $\boldsymbol{R}_{\boldsymbol{u}} \in \mathbb{R}^{d\times d}$ and has two eigenvalues, $\lambda_1, \lambda_2 \geq 0$, that satisfy*

$$
\lambda_1 = \Theta\left(\frac{1}{\|\boldsymbol{u}\|_2^3}\right) \tag{63}
$$

$$
\lambda_2 = \Theta\left(\frac{1}{\|\boldsymbol{u}\|_2}\right). \tag{64}
$$

*Proof.* Recall that Gaussian random variables are rotationally invariant. Specifically, for orthonormal matrix $\boldsymbol{R} \in \mathbb{R}^{d\times d}$ and $\boldsymbol{X} \sim \mathcal{N}(\boldsymbol{0},\boldsymbol{I}_d)$, it follows that $\boldsymbol{R}\boldsymbol{X} \sim \mathcal{N}(\boldsymbol{0},\boldsymbol{I}_d)$. Moreover, $\boldsymbol{R}^\top\boldsymbol{R} = \boldsymbol{R}\boldsymbol{R}^\top = \boldsymbol{I}_d$. Using this notion, we will 1) diagonalize (62), then 2) evaluate the eigenvalues of (62) as the diagonal elements.

Consider the orthonormal rotation matrix $\boldsymbol{R}_{\boldsymbol{u}} \in \mathbb{R}^{d\times d}$ that is such that $\boldsymbol{R}_{\boldsymbol{u}}\boldsymbol{u} = \boldsymbol{e}_1\|\boldsymbol{u}\|_2$ where $\boldsymbol{e}_j$ is the $j^{th}$ canonical vector of $\mathbb{R}^d$. Using this change of basis matrix, we can now obtain the diagonal matrix,

$$
\begin{aligned}
\boldsymbol{R}_{\boldsymbol{u}}\left(\mathbb{E}_{\boldsymbol{X}}\left[\boldsymbol{x}\boldsymbol{x}^\top\frac{e^{\boldsymbol{x}^\top\boldsymbol{u}}}{\left(1+e^{\boldsymbol{x}^\top\boldsymbol{u}}\right)^2}\right]\right)\boldsymbol{R}_{\boldsymbol{u}}^\top &= \mathbb{E}_{\boldsymbol{X}}\left[\boldsymbol{R}_{\boldsymbol{u}}\boldsymbol{x}\boldsymbol{x}^\top\boldsymbol{R}_{\boldsymbol{u}}^\top\frac{e^{\boldsymbol{x}^\top\boldsymbol{R}_{\boldsymbol{u}}^\top\boldsymbol{R}_{\boldsymbol{u}}\boldsymbol{u}}}{\left(1+e^{\boldsymbol{x}^\top\boldsymbol{R}_{\boldsymbol{u}}^\top\boldsymbol{R}_{\boldsymbol{u}}\boldsymbol{u}}\right)^2}\right] \\
&= \mathbb{E}_{\boldsymbol{R}_{\boldsymbol{u}}\boldsymbol{x}}\left[\tilde{x}\tilde{x}^\top\frac{e^{\tilde{x}^\top\boldsymbol{e}_1\|\boldsymbol{u}\|_2}}{\left(1+e^{\tilde{x}^\top\boldsymbol{e}_1\|\boldsymbol{u}\|_2}\right)^2}\right] \\
&= \begin{pmatrix}\mathbb{E}_{\boldsymbol{R}_{\boldsymbol{u}}\boldsymbol{x}}\left[\frac{\tilde{x}_1^2 e^{\tilde{x}_1\|\boldsymbol{u}\|_2}}{\left(1+e^{\tilde{x}_1\|\boldsymbol{u}\|_2}\right)^2}\right] & 0 & 0 \\ 0 & \cdots & 0 \\ 0 & 0 & \mathbb{E}_{\boldsymbol{R}_{\boldsymbol{u}}\boldsymbol{x}}\left[\frac{e^{\tilde{x}_1\|\boldsymbol{u}\|_2}}{\left(1+e^{\tilde{x}_1\|\boldsymbol{u}\|_2}\right)^2}\right]\end{pmatrix}.
\end{aligned}
$$

It has only two eigenvalues given in closed form as

$$
\lambda_1 = \mathbb{E}_{\tilde{X}_1}\left[\frac{\tilde{x}_1^2 e^{\tilde{x}_1\|\boldsymbol{u}\|_2}}{\left(1+e^{\tilde{x}_1\|\boldsymbol{u}\|_2}\right)^2}\right] = \int_{-\infty}^{\infty}\frac{\tilde{x}_1^2 e^{\tilde{x}_1\|\boldsymbol{u}\|_2}}{\left(1+e^{\tilde{x}_1\|\boldsymbol{u}\|_2}\right)^2}p(\tilde{\boldsymbol{x}}_1)d\tilde{\boldsymbol{x}}_1
$$

$$
\lambda_2 = \mathbb{E}_{\tilde{X}_1}\left[\frac{e^{\tilde{x}_1\|\boldsymbol{u}\|_2}}{\left(1+e^{\tilde{x}_1\|\boldsymbol{u}\|_2}\right)^2}\right] = \int_{-\infty}^{\infty}\frac{e^{\tilde{x}_1\|\boldsymbol{u}\|_2}}{\left(1+e^{\tilde{x}_1\|\boldsymbol{u}\|_2}\right)^2}p(\tilde{\boldsymbol{x}}_1)d\tilde{\boldsymbol{x}}_1.
$$

The rest of the proof is spent evaluating tight lower and upper bounds on $\lambda_1, \lambda_2$ in terms of $\|\boldsymbol{u}\|_2$.

**Part a):** Bounds for $\lambda_1$.

For $\tilde{x}_1 \sim \mathcal{N}(0, 1)$ the probability density function is bounded above by 1: $p(\tilde{x}_1) \leq 1$. Then we can upper bound $\lambda_1$ as follows:

$$
\begin{aligned}
\lambda_1 &= \int_{-\infty}^{\infty} \frac{\tilde{x}_1^2 e^{\tilde{x}_1 \|\boldsymbol{u}\|_2}}{\left(1 + e^{\tilde{x}_1 \|\boldsymbol{u}\|_2}\right)^2} p(\tilde{x}_1) d\tilde{x}_1 \\
&\leq \int_{-\infty}^{\infty} \frac{\tilde{x}_1^2 e^{\tilde{x}_1 \|\boldsymbol{u}\|_2}}{\left(1 + e^{\tilde{x}_1 \|\boldsymbol{u}\|_2}\right)^2} d\tilde{x}_1 \\
&= 2 \int_0^{\infty} \tilde{x}_1^2 e^{-\tilde{x}_1 \|\boldsymbol{u}\|_2} d\tilde{x}_1 \\
&= \frac{4}{\|\boldsymbol{u}\|_2^3} \\
&= \mathcal{O}\left(\frac{1}{\|\boldsymbol{u}\|_2^3}\right)
\end{aligned}
$$

For the lower bounds, we will use the fact that $e^{-(\|\boldsymbol{u}\|_2 \tilde{x} + \tilde{x}_1^2/2)} \geq e^{-\|\boldsymbol{u}\|_2^2 \tilde{x}^2/2}$ for $x \in \left[\frac{4}{\|\boldsymbol{u}\|_2}, \infty\right]$ and $\|\boldsymbol{u}\|_2 \geq \sqrt{2}$. Then, we can lower bound $\lambda_1$ as follows:

$$
\begin{aligned}
\lambda_1 &= \int_{-\infty}^{\infty} \frac{\tilde{x}_1^2 e^{\tilde{x}_1 \|\boldsymbol{u}\|_2}}{\left(1 + e^{\tilde{x}_1 \|\boldsymbol{u}\|_2}\right)^2} p(\tilde{x}_1) d\tilde{x}_1 \\
&\geq 2 \int_0^{\infty} \frac{\tilde{x}_1^2 e^{\tilde{x}_1 \|\boldsymbol{u}\|_2}}{\left(2 e^{\tilde{x}_1 \|\boldsymbol{u}\|_2}\right)^2} \left(\frac{e^{-\frac{\tilde{x}_1^2}{2}}}{\sqrt{2\pi}}\right) d\tilde{x}_1 \\
&= \frac{1}{2\sqrt{2\pi}} \int_0^{\infty} \tilde{x}_1^2 e^{-\tilde{x}_1 \|\boldsymbol{u}\|_2 - \frac{\tilde{x}_1^2}{2}} d\tilde{x}_1 \\
&\geq \frac{1}{2\sqrt{2\pi}} \int_{\frac{4}{\|\boldsymbol{u}\|_2}}^{\infty} \tilde{x}_1^2 e^{-\|\boldsymbol{u}\|_2^2 \tilde{x}^2/2} d\tilde{x}_1 \\
&\geq \frac{1}{4\sqrt{2\pi} \|\boldsymbol{u}\|_2^3} \left(\sqrt{\pi} \mathrm{erf}(\tilde{x}_1 \|\boldsymbol{u}\|_2) - 2\|\boldsymbol{u}\|_2 \tilde{x}_1 e^{-\|\boldsymbol{u}\|_2^2 \tilde{x}_1^2}\right)_{\frac{4}{\|\boldsymbol{u}\|_2}}^{\infty} \\
&\geq \Omega\left(\frac{1}{\|\boldsymbol{u}\|_2^3}\right).
\end{aligned}
$$

Therefore, it holds that $\lambda_1 = \Theta\left(\frac{1}{\|\boldsymbol{u}\|_2^3}\right)$.

**Part b):** Bounds for $\lambda_2$.

For $\tilde{x}_1 \sim \mathcal{N}(0, 1)$ the probability density function is bounded above by 1: $p(\tilde{x}_1) \leq 1$. Then we can upper bound $\lambda_2$ as follows:

$$
\begin{aligned}
\lambda_2 &= \int_{-\infty}^{\infty} \frac{e^{\tilde{x}_1 \|\boldsymbol{u}\|_2}}{\left(1 + e^{\tilde{x}_1 \|\boldsymbol{u}\|_2}\right)^2} p(\tilde{x}_1) d\tilde{x}_1 \\
&\leq \int_{-\infty}^{\infty} \frac{e^{\tilde{x}_1 \|\boldsymbol{u}\|_2}}{\left(1 + e^{\tilde{x}_1 \|\boldsymbol{u}\|_2}\right)^2} d\tilde{x}_1 \\
&= 2 \int_0^{\infty} e^{-\tilde{x}_1 \|\boldsymbol{u}\|_2} d\tilde{x}_1 \\
&= \frac{1}{\|\boldsymbol{u}\|_2} \\
&= \mathcal{O}\left(\frac{1}{\|\boldsymbol{u}\|_2}\right)
\end{aligned}
$$

For the lower bounds, we will use the fact that $e^{-(\|\boldsymbol{u}\|_2 \tilde{\boldsymbol{x}}_1 + \tilde{\boldsymbol{x}}_1^2/2)} \geq e^{-\|\boldsymbol{u}\|_2^2 \tilde{\boldsymbol{x}}_1^2/2}$ for $x \in \left[\frac{4}{\|\boldsymbol{u}\|_2}, \infty\right]$ and $\|\boldsymbol{u}\|_2 \geq \sqrt{2}$. Then, we can lower bound $\lambda_2$ as follows:

$$
\begin{aligned}
\lambda_2 &= \int_{-\infty}^{\infty} \frac{e^{\tilde{x}_1 \|\boldsymbol{u}\|_2}}{\left(1 + e^{\tilde{x}_1 \|\boldsymbol{u}\|_2}\right)^2} p(\tilde{\boldsymbol{x}}_1) d\tilde{\boldsymbol{x}}_1 \\
&\geq 2 \int_0^{\infty} \frac{e^{\tilde{x}_1 \|\boldsymbol{u}\|_2}}{\left(2 e^{\tilde{x}_1 \|\boldsymbol{u}\|_2}\right)^2} \left(\frac{e^{-\frac{\tilde{x}_1^2}{2}}}{\sqrt{2\pi}}\right) d\tilde{\boldsymbol{x}}_1 \\
&= \frac{1}{2\sqrt{2\pi}} \int_0^{\infty} e^{-\tilde{x}_1 \|\boldsymbol{u}\|_2 - \frac{\tilde{x}_1^2}{2}} d\tilde{\boldsymbol{x}}_1 \\
&\geq \frac{1}{2\sqrt{2\pi}} \int_{\frac{4}{\|\boldsymbol{u}\|_2}}^{\infty} e^{-\|\boldsymbol{u}\|_2^2 \tilde{\boldsymbol{x}}^2/2} d\tilde{\boldsymbol{x}}_1 \\
&\geq \frac{1}{4\sqrt{2\pi}\|\boldsymbol{u}\|_2} \left(\sqrt{\pi} \mathrm{erf}(\tilde{\boldsymbol{x}}_1 \|\boldsymbol{u}\|_2)\right)_{\frac{4}{\|\boldsymbol{u}\|_2}}^{\infty} \\
&\geq \Omega\left(\frac{1}{\|\boldsymbol{u}\|_2}\right).
\end{aligned}
$$

Therefore, it holds that $\lambda_2 = \Theta\left(\frac{1}{\|\boldsymbol{u}\|_2}\right)$. $\qquad\square$

## B.7. Existence of Locally Convex Region

In this section, we further discuss the consequences of Corollary B.1, Theorem B.2, and Theorem B.4.

To begin, we will discuss the sufficient condition for Gradient Descent to converge linearly to the true parameters $\boldsymbol{\theta}^*$ and compare it to that of EM. For Gradient Descent, it is well understood that if $\boldsymbol{\theta}^1$ is initialized in a convex set $\Theta$ that contains $\boldsymbol{\theta}^*$ and where $\mathcal{L}(\boldsymbol{\theta})$ is strongly convex, i.e.

$$
\nabla^2 \mathcal{L}(\boldsymbol{\theta}) \succeq \alpha \boldsymbol{I}_{2d} \qquad \text{for all } \boldsymbol{\theta} \in \Theta, \tag{65}
$$

then the parameter iterates converge linearly to $\boldsymbol{\theta}^*$. However, as we have shown for SymMoLinE and SymMoLogE, the sufficient condition for EM to converge linearly to $\boldsymbol{\theta}^*$ is slightly different. Instead, we require $\Theta$ to satisfy that $\mathcal{L}(\boldsymbol{\theta})$ is strongly convex relative to $\boldsymbol{A}(\boldsymbol{\theta})$, i.e.,

$$
\nabla^2 \mathcal{L}(\boldsymbol{\theta}) \succeq \alpha \nabla^2 A(\boldsymbol{\theta}) \qquad \text{for all } \boldsymbol{\theta} \in \Theta. \tag{66}
$$

Interestingly, if it holds that $A(\boldsymbol{\theta})$ is 1-smooth, we see that (66) is weaker than (65), i.e.,

$$
\nabla^2 A(\boldsymbol{\theta}) \preceq I_{2d} \text{ and (65) holds} \implies (66). \tag{67}
$$

But more interestingly, as long as $A(\boldsymbol{\theta})$ is $\mu$-smooth for some $\mu > 0$, it will hold that any set $\Theta$ satisfying (65) for some $\alpha > 0$ will also satisfy (66) with $\tilde{\alpha} = \frac{\alpha}{\mu} > 0$. We note that the converse holds for $A(\boldsymbol{\theta})$ strongly convex. In summary, it then holds that, for SymMoLinE and SymMoLogE, EM's sufficient conditions for a linear rate is strictly weaker than that of Gradient Descent when the mirror map $A(\boldsymbol{\theta})$ is convex, but not strongly convex.

Next, we will further discuss the implications of Theorem B.4. In the theorem, we obtain clear lower and upper bounds for the eigenvalues of $\boldsymbol{I}_{\boldsymbol{x},y,z|\boldsymbol{\theta}}$. However, it is not clear how to do the same for $\boldsymbol{I}_{z|\boldsymbol{x},y,\boldsymbol{\theta}}$ as the trick to rotate the axis with an orthonormal matrix $R$ to simplify the expression will not work here because the distribution of the vector $(\boldsymbol{x}, y\boldsymbol{x})^\top$ is not invariant to rotation. Still, there are some things that we can say for special cases. For this discussion, we will constrain ourselves to SymMoLogE. However, the same lines of reasoning also apply to SymMoLinE. First we recall from Theorem B.4 that

$$
\boldsymbol{I}_{z|\boldsymbol{x},y,\boldsymbol{\theta}} = \mathbb{E}_{\boldsymbol{X},Y} \left[ \frac{\exp\left\langle \begin{bmatrix} \boldsymbol{x} \\ y\boldsymbol{x} \end{bmatrix}, \boldsymbol{\theta} \right\rangle \begin{bmatrix} \boldsymbol{x} \\ y\boldsymbol{x} \end{bmatrix} \begin{bmatrix} \boldsymbol{x} \\ y\boldsymbol{x} \end{bmatrix}^\top}{\left(1 + \exp\left\langle \begin{bmatrix} \boldsymbol{x} \\ y\boldsymbol{x} \end{bmatrix}, \boldsymbol{\theta} \right\rangle\right)^2} \right].
$$

From here, one easy way to approach bounding the above is to 1) recall that any outer product of the form $\boldsymbol{u}\boldsymbol{u}^\top$ has a single eigenvalue given as $\|\boldsymbol{u}\|_2^2$, and 2) the inner product between two vectors is equal to the product of their norms and the cosine of the angle between them, i.e. $\boldsymbol{s}^\top\boldsymbol{u} = \|\boldsymbol{s}\|_2\|\boldsymbol{u}\|_2\cos(\phi_{\boldsymbol{s},\boldsymbol{u}})$. Thus, denoting $\phi$ to be the angle between $\boldsymbol{\theta}$ and the vector $(\boldsymbol{x}, y\boldsymbol{x})^\top$, we obtain

$$\boldsymbol{I}_{z|\boldsymbol{x},y,\boldsymbol{\theta}} \preceq \mathbb{E}_{\boldsymbol{X},Y,\Phi}\left[\frac{e^{\sqrt{(1+y^2)\|\boldsymbol{x}\|_2^2}\|\boldsymbol{\theta}\|_2\cos(\phi)}(1+y^2)\|\boldsymbol{x}\|_2^2}{\left(1+e^{\sqrt{(1+y^2)\|\boldsymbol{x}\|_2^2}\|\boldsymbol{\theta}\|_2\cos(\phi)}\right)^2}\right]\boldsymbol{I}_{2d}$$

$$\preceq \mathbb{E}_{\boldsymbol{X},Y,\Phi}\left[e^{-\left|\sqrt{(1+y^2)\|\boldsymbol{x}\|_2^2}\|\boldsymbol{\theta}\|_2\cos(\phi)\right|}(1+y^2)\|\boldsymbol{x}\|_2^2\right]\boldsymbol{I}_{2d}.$$

Denoting $s := \left|\sqrt{(1+y^2)\|\boldsymbol{x}\|_2^2}\right|$, we can write the above expectation as

$$8\int_0^{\pi/2}\int_0^\infty s^2 e^{-s\|\boldsymbol{\theta}\|_2\cos(\phi)}p(s,\phi)dsd\phi.$$

Subsequently, the idea is bound $p(s,\phi) = p(s)p(\phi|s)$ in a way that makes integration easy.

The case where $x, w, \beta \in \mathbb{R}$ is fairly easy. Under this scenario, $x \sim \mathcal{N}(0,1)$ and $\boldsymbol{I}_{x,y,z|\boldsymbol{\theta}}$ can be upper-bounded as

$$\boldsymbol{I}_{z|\boldsymbol{x},y,\boldsymbol{\theta}} = \mathbb{E}_{X,Y}\left[\begin{pmatrix} x^2 & yx^2 \\ yx^2 & y^2x^2 \end{pmatrix}\frac{e^{x(w+y\beta)}}{\left(1+e^{x(w+y\beta)}\right)^2}\right]$$

$$\leq \mathbb{E}_{X,Y}\left[x^2(1+y^2)e^{-|x(w+y\beta)|}\right]\boldsymbol{I}_2.$$

Now, recall that $y \in \{-1,1\}$, $P(y|x) \leq 1$, $p(x) = \frac{e^{-x^2/2}}{\sqrt{2\pi}} \leq \frac{2e^{-|x|}}{\sqrt{2\pi}}$, and see that

$$\boldsymbol{I}_{z|\boldsymbol{x},y,\boldsymbol{\theta}} = \mathbb{E}_X\left[2x^2\left(e^{-|x(w-\beta)|}P(y=1|x) + e^{-|x(w+\beta)|}P(y=-1|x)\right)\right]$$

$$\leq \mathbb{E}_X\left[2x^2\left(e^{-|x(w-\beta)|} + e^{-|x(w+\beta)|}\right)\right]$$

$$= 4\int_0^\infty x^2\left(e^{-|x(w-\beta)|} + e^{-|x(w+\beta)|}\right)p(x)dx$$

$$\leq 4\int_0^\infty x^2\left(e^{-x(|w-\beta|+1)} + e^{-x(|w+\beta|+1)}\right)dx$$

$$\leq 8\left(\frac{1}{(1+|w-\beta|)^3} + \frac{1}{(1+|w+\beta|)^3}\right)$$

$$\leq \mathcal{O}\left(\frac{1}{(1+|w-\beta|)^3} + \frac{1}{(1+|w+\beta|)^3}\right).$$

Subsequently, together with the fact that $\boldsymbol{I}_{x,y,z|\boldsymbol{\theta}}^{-1} \leq \max\left\{\mathcal{O}\left(\|\boldsymbol{w}\|_2^3\right), \left(\|\boldsymbol{\beta}\|_2^3\right)\right\}$, it holds that the eigenvalues of the MIM are upper-bounded by

$$\max\left\{\mathcal{O}\left(\left(\frac{w}{1+|w-\beta|}\right)^3 + \left(\frac{w}{1+|w+\beta|}\right)^3\right), \mathcal{O}\left(\left(\frac{\beta}{1+|w-\beta|}\right)^3 + \left(\frac{\beta}{1+|w+\beta|}\right)^3\right)\right\}.$$

This special case is closely related to the case where $\boldsymbol{\beta}$ is parallel to $\boldsymbol{w}$; a similar approach will work.

# C. Additional Experiments

## C.1. Experiment on Grayscale CIFAR-10

In this section, we consider an additional mini-batch experiment on grayscale CIFAR-10 with a 5-component MoE. The MoE to be trained consists of individual experts that each consist of a single hidden layer MLP with hidden dimension 100 and ReLU activation. The gating function also consists of a single hidden layer MLP with hidden dimension 100 and ReLU activation. We randomly initialize each linear layer to have rows that are unit-norm and execute the algorithms on the same datasets and with the same initializations. For Gradient EM, the only additional code needed over GD is to define the EM Loss function appropriately, and then perform a Gradient Step on the Gating parameters and the Expert parameters separately as describe in Algorithm 2. For EM, for each iteration, we perform several gradient steps in an inner loop to approximately recover the solutions to the sub-problems described in (9). We report our findings for the mini-batch iteration of the respective algorithms in Figure 4.

We report the respective final test accuracy and cross-entropy loss values after 50 epochs of EM, Gradient EM and GD for fitting a 5-component MoE on the CIFAR-10. We see that EM boasts a much improved final test accuracy of $41.6\%$. Meanwhile, Gradient EM also registers an improvement over GD at $37.4\%$. Finally, GD obtained a final accuracy of $34.5\%$. While these accuracies are themselves not good, the challenge was to find an optimal partitioning of the data and utilize very weak experts. In Figure 4, we report the progress made on the accuracy for the test set over the 50 epochs, averaged over 25 instances. As was observed in our synthetic experiment and Fashion MNIST experiments, EM takes considerably less iterations to fit the mixture than both Gradient EM and GD, where the former also takes considerably less time to fit the mixture than GD. To validate our results further, we perform a paired t-test. For EM and Gradient EM compared to GD, we obtain a T-statistic $\geq 22$ indicating that the difference in final accuracy is statistically significant (p-value $\sim 0.000$).

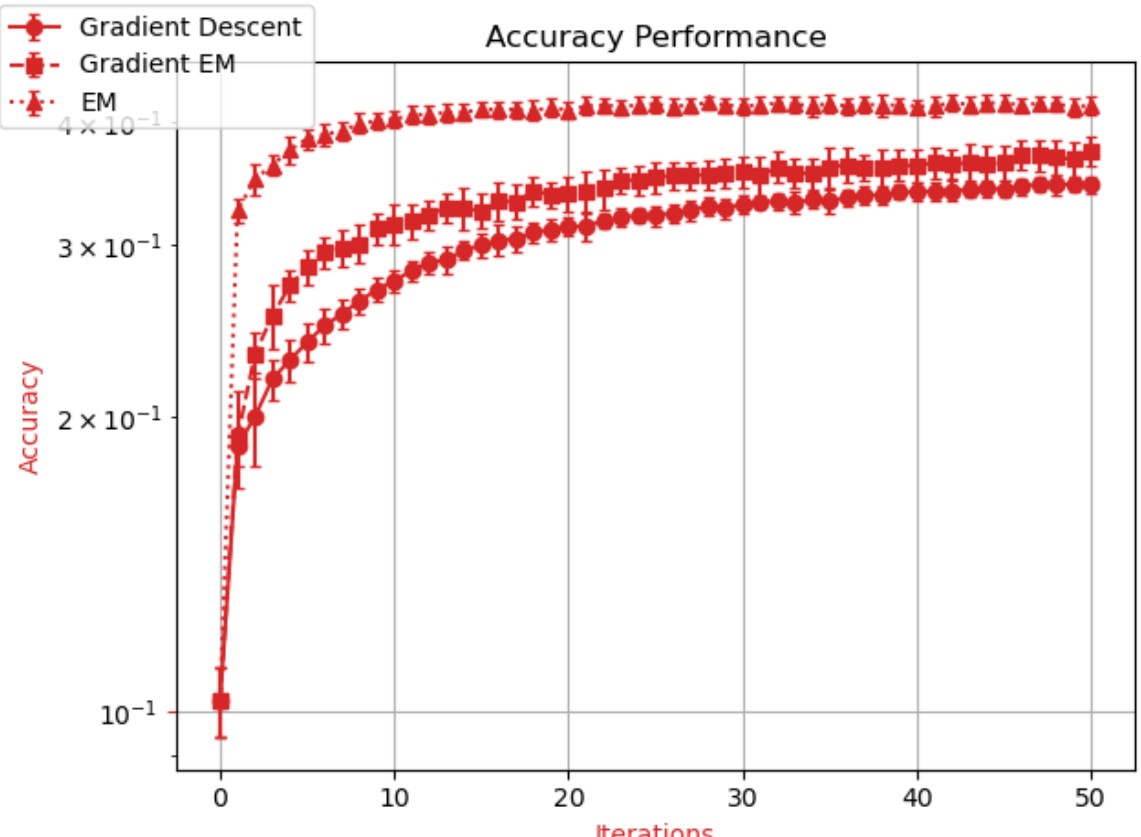

*Figure 4.* Convergence in predicted label Accuracy for different optimization methods.

# D. Algorithms

In this section, we provide explicit formulations of EM, Gradient EM and Gradient Descent for the context of MoE optimization and EM for deep and sparse MoE.

## D.1. EM for MoE

**Expectation-Maximization (EM):** EM takes a structured approach to minimizing the objective $\mathcal{L}(\boldsymbol{\theta})$ in (7). Each iteration of EM is decomposed into two steps as follows. The first step is called "expectation": For current parameter estimate $\boldsymbol{\theta}^t$, we compute the expectation of the complete-data log-likelihood with respect to the latent variables, using the current parameter estimates $\boldsymbol{\theta}^t$ and denote it by $Q(\boldsymbol{\theta}|\boldsymbol{\theta}^t)$, i.e.,

$$Q(\boldsymbol{\theta}|\boldsymbol{\theta}^t) = -\mathbb{E}_{X,Y}\left[\mathbb{E}_{Z|\boldsymbol{x},y;\boldsymbol{\theta}^t}[\log p(\boldsymbol{x},y,z;\boldsymbol{\theta})]\right]. \tag{68}$$

Then, in the second step called "maximization", we simply minimize the objective $Q(\boldsymbol{\theta}|\boldsymbol{\theta}^t)$ (or maximize $-Q(\boldsymbol{\theta}|\boldsymbol{\theta}^t)$) with respect to $\boldsymbol{\theta} \in \Omega$ and obtain our new parameter as

$$\boldsymbol{\theta}^{t+1} := \underset{\boldsymbol{\theta}\in\Omega}{\arg\min}\, Q(\boldsymbol{\theta}|\boldsymbol{\theta}^t). \tag{69}$$

For MoE described in Section 2, $\log p(y,z|\boldsymbol{x};\boldsymbol{\theta}) = \log p(y|z,\boldsymbol{x};\boldsymbol{\beta}) + \log p(z|\boldsymbol{x};\boldsymbol{w})$. It follows that the EM objective (8) is linearly separable in the parameters $\boldsymbol{\beta}$ and $\boldsymbol{w}$. Thus, we can rewrite $Q(\boldsymbol{\theta}|\boldsymbol{\phi})$ as the sum of two functions that depend only on $\boldsymbol{\beta}$ and $\boldsymbol{w}$, respectively. Subsequently, the EM update (9) is obtained as the concatenation $\boldsymbol{\theta}^{t+1} = (\boldsymbol{w}^{t+1}, \boldsymbol{\beta}^{t+1})^\top$, where

$$\boldsymbol{w}^{t+1} = \underset{\boldsymbol{w}\in\mathbb{R}^d}{\arg\min}\, -\mathbb{E}_{\boldsymbol{X},Y}\left[\mathbb{E}_{Z|\boldsymbol{x},y;\boldsymbol{\theta}^t}\left[\log p(z|\boldsymbol{x};\boldsymbol{w})\right]\right],$$

$$\boldsymbol{\beta}^{t+1} = \underset{\boldsymbol{\beta}\in\mathbb{R}^d}{\arg\min}\, -\mathbb{E}_{\boldsymbol{X},Y}\left[\mathbb{E}_{Z|\boldsymbol{x},y;\boldsymbol{\theta}^t}\left[\log p(y|z,\boldsymbol{x};\boldsymbol{\beta})\right]\right].$$

---

**Algorithm 1** EM for MoE

---

**Input:** Initial $\boldsymbol{\theta}^1 \in \Omega$, data: $(\boldsymbol{X},Y) \sim p(\boldsymbol{x},y;\boldsymbol{\theta}^*)$
**for** $t = 1$ to $T$ **do**
    $\boldsymbol{\theta}$-**Update:** Obtain $\boldsymbol{\theta}^{t+1}$ as
        $\boldsymbol{\theta}^{t+1} := \arg\min_{\boldsymbol{\theta}\in\Omega} Q(\boldsymbol{\theta} \mid \boldsymbol{\theta}^t)$
**end for**
**Output:** $\boldsymbol{\theta}^T = (\boldsymbol{w}^T, \boldsymbol{\beta}^T)$

---

## D.2. Gradient EM for MoE

**Gradient EM.** Whereas EM performs the global minimization of the EM objective given in (68), Gradient EM obtains its next parameter iterate as the concatenation of two gradient updates on the sub objectives,

$$-\,\mathbb{E}_{\boldsymbol{X},Y}\left[\mathbb{E}_{Z|\boldsymbol{x},y;\boldsymbol{\theta}^t}\left[\log P(z|\boldsymbol{x};\boldsymbol{w})\right]\right] \tag{70}$$

$$-\,\mathbb{E}_{\boldsymbol{X},Y}\left[\mathbb{E}_{Z|\boldsymbol{x},y;\boldsymbol{\theta}^t}\left[\log p(y|z,\boldsymbol{x};\boldsymbol{\beta})\right]\right] \tag{71}$$

where the EM objective is given as the summation of (70) and (71).

---

**Algorithm 2** Gradient EM for MOE

---

**Input**: Initial $\boldsymbol{\theta}^1 \in \Omega$, data: $(\boldsymbol{X}, Y) \sim p(\boldsymbol{x}, y; \boldsymbol{\theta}^*))$, step-size: $\gamma_1, \gamma_2 \in (0, \infty)$.
**for** $t = 1, \ldots, T$: **do**
   $\boldsymbol{\beta}$-**Update**: Obtain $\boldsymbol{\beta}^{t+1}$ as

$$\boldsymbol{\beta}^{t+1} = \boldsymbol{\beta}^t + \gamma_1 \mathbb{E}_{\boldsymbol{X}, Y} \mathbb{E}_{Z|\boldsymbol{x}, y; \boldsymbol{\theta}^t} \left[ \frac{\partial}{\partial \boldsymbol{\beta}} \log p(y|z, \boldsymbol{x}; \boldsymbol{\beta}) \right].$$

   $\boldsymbol{w}$-**Update**: Obtain $\boldsymbol{w}^{t+1}$ as

$$\boldsymbol{w}^{t+1} = \boldsymbol{w}^t + \gamma_2 \mathbb{E}_{\boldsymbol{X}, Y} \mathbb{E}_{Z|\boldsymbol{x}, y; \boldsymbol{\theta}^t} \left[ \frac{\partial}{\partial \boldsymbol{w}} \log P(z|\boldsymbol{x}; \boldsymbol{w}) \right].$$

**end for**
**Output**: $\boldsymbol{\theta}^T = (\boldsymbol{w}^T, \boldsymbol{\beta}^T)$

---

## D.3. Gradient Descent for MoE

**Gradient Descent.** Gradient descent is given as the global minimizer of the first order approximation of $\mathcal{L}(\boldsymbol{\theta})$ at $\boldsymbol{\theta}$ plus a quadratic regularizer, i.e.,

$$\mathcal{L}(\boldsymbol{\theta}^t) + \langle \nabla \mathcal{L}(\boldsymbol{\theta}^t), \boldsymbol{\theta} - \boldsymbol{\theta}^t \rangle + \frac{1}{2\eta} \|\boldsymbol{\theta} - \boldsymbol{\theta}^t\|_2^2.$$

Differentiating, and solving for equality at 0 yields the well known gradient update.

---

**Algorithm 3** Gradient Descent for MoE

---

**Input:** Initial $\boldsymbol{\theta}^1 \in \Omega$, data: $(\boldsymbol{X}, Y) \sim p(\boldsymbol{x}, y; \boldsymbol{\theta}^*)$, step-size: $\gamma \in \mathbb{R}^+$
**for** $t = 1$ to $T$ **do**
   $\boldsymbol{\theta}$-**Update:**
     $\boldsymbol{\theta}^{t+1} := \boldsymbol{\theta}^t - \gamma \nabla \mathcal{L}(\boldsymbol{\theta}^t)$
**end for**
**Output:** $\boldsymbol{\theta}^T = (\boldsymbol{w}^T, \boldsymbol{\beta}^T)$

---

## D.4. EM for Deep and Sparse MoE

We begin by formalizing the concepts of Deep and Sparse Mixtures of Experts (MoE), extending the classical MoE framework.

**Deep MoE:** A *deep MoE* is a composition of $l \geq 2$ MoE blocks, denoted as $\text{MoE}_1, \text{MoE}_2, \ldots, \text{MoE}_l$, stacked sequentially. Each block $\text{MoE}_i$ consists of a gating function $g_i(\boldsymbol{x}; \boldsymbol{w}_i)$ and a set of $k$ experts $\{f_{i,j}(\boldsymbol{x}; \boldsymbol{\beta}_{i,j})\}_{j=1}^k$. The input $\boldsymbol{x}$ is processed through each MoE block in sequence, producing intermediate representations $\boldsymbol{h}_1, \boldsymbol{h}_2, \ldots, \boldsymbol{h}_l$, where:

$$\boldsymbol{h}_1 = \sum_{j=1}^k g_1(\boldsymbol{x}; \boldsymbol{w}_1)_j f_{1,j}(\boldsymbol{x}; \boldsymbol{\beta}_{1,j}),$$

$$\boldsymbol{h}_i = \sum_{j=1}^k g_i(\boldsymbol{h}_{i-1}; \boldsymbol{w}_i)_j f_{i,j}(\boldsymbol{h}_{i-1}; \boldsymbol{\beta}_{i,j}) \quad \text{for } i = 2, \ldots, l.$$

The final output is $\boldsymbol{y} = \boldsymbol{h}_l$.

**Sparse MoE:** The *Sparse MoE* is a variant of the MoE that is popular in deep MoE applications where, at each MoE block, only a small subset of experts (typically one or a few) are activated per input both during training and inference. This is achieved via a deterministic or stochastic selection mechanism (e.g., top-$k$ gating), resulting in a sparse latent variable $z = (z_1, z_2, \ldots, z_l) \in [k]^l$ that encodes the sequence of selected experts across the $l$ blocks.

**EM for Deep and Sparse MoE:** We now describe an EM-like algorithm for training deep and sparse MoE models. The key idea is to treat the expert selection sequence $z = (z_1, \ldots, z_l)$ as a latent variable and optimize the expected complete-data log-likelihood. We let $\boldsymbol{\theta} = (\boldsymbol{w}_1, \ldots, \boldsymbol{w}_l, \boldsymbol{\beta}_{1,1}, \ldots, \boldsymbol{\beta}_{l,k})$ denote all model parameters.

In the classical *non-sparse MoE setting* where we do not choose a subset of experts to go through at each layer, the latent variables can only be resolved at the last layer. The EM surrogate is then given as follows:

$$Q(\boldsymbol{\theta}|\boldsymbol{\theta}^t) = -\mathbb{E}_{X,Y}\left[\mathbb{E}_{Z_l|\boldsymbol{x},y;\boldsymbol{\theta}^t}\left[\log p(y|\boldsymbol{x}, z_l; \boldsymbol{\theta})P(z_l|\boldsymbol{x};\boldsymbol{\theta})\right]\right]. \tag{72}$$

where the given probability functions is given at the last layer as

$$p(y|\boldsymbol{x}, z_1, ..., z_l; \boldsymbol{\theta}) = p\left(y|\boldsymbol{h}_{l-1}\right)$$
$$P(z_1, ..., z_l|\boldsymbol{x}; \boldsymbol{\theta}) = p(z_l|\boldsymbol{h}_{l-1}; \theta).$$

In the *sparse MoE setting* where $p$ experts are chosen at each layer, we re-define the latent variables $Z_i$ to be defined over the set of all possible ordered combinations of $p$ experts that could have been chosen out of the $k$ available experts. The latent space embeddings is now given to be $\hat{h}_i := \sum_{j \in z_i} g_i(\boldsymbol{h}_{i-1}; \boldsymbol{w}_i)_j f_{i,j}(\boldsymbol{h}_{i-1}; \boldsymbol{\beta}_{i,j})$. The EM surrogate is given by

$$Q(\boldsymbol{\theta}|\boldsymbol{\theta}^t) = -\mathbb{E}_{X,Y}\left[\mathbb{E}_{Z_1,...,Z_l|\boldsymbol{x},y;\boldsymbol{\theta}^t}\left[\log p(y|\boldsymbol{x}, z_1, ..., z_l; \boldsymbol{\theta})P(z_1, ..., z_l|\boldsymbol{x};\boldsymbol{\theta})\right]\right]. \tag{73}$$

where the given probability functions are decomposed per layer as

$$p(y|\boldsymbol{x}, z_1, ..., z_l; \boldsymbol{\theta}) = p\left(y|\hat{\boldsymbol{h}}_{l-1}\right)$$
$$P(z_1, ..., z_l|\boldsymbol{x}; \boldsymbol{\theta}) = P(z_1|x; \theta) \cdots p(z_l|z_1, ..., z_{l-1}; \theta).$$

In practice, constructing the above EM surrogate this would require to make a combinatorial number of forward passes through the model to evaluate the surrogate loss function. To remediate this, we will use the strategy to only evaluate the loss on the greedily chosen sequence through the model as is the case for GD-type solutions for Sparse MoE.

