# OpenReview forum: "Learning Mixtures of Experts with EM: A Mirror Descent Perspective"
_ICML.cc/2025/Conference — ICML 2025 poster_

### Official Review · Reviewer_kaEY · 2025-03-08

**Overall Recommendation:** 3

**Summary:**

In this paper, the authors discussed the relationship between the EM algorithm and mirror descent in the context of the mixture of experts (MOE) learning. In the beginning, the authors proposed an overview of the EM-based parameter learning procedure in the MOE. On this basis, the authors then introduced the projected mirror descent (PMD), bridged the relationship between EM-based MOE learning and PMD-based MOE learning, and derived the convergence property of the EM-based MOE with the help of PMD. After that, they extend their proposed theorem to the Symmetric Mixture of Linear Experts (SymMoLinE) as a special case. Finally, various experiments are conducted to demonstrate the efficacy of the proposed approach.

**Claims And Evidence:**

Nearly all the claims made in the submission are supported by clear and convincing evidence. Nevertheless, the symbols seem quite strange in this manuscript. For example, in Eq. (1), the authors introduced $\phi$ but did not state what $\phi$ means. The parameter of some MOE models? In addition, on page 6, right column, `Additionally, while GD regularizes progress based on the Euclidean distance between iterates, EM adjusts progress based on the divergence between probability distributions across iterations. This is often more suitable for latent variable models, where small Euclidean changes may cause large shifts in the mixture distribution, and vice versa.`, although the reviewer buys in this statement, the authors should add extra citations or add related figures to back up this phenomenon.

**Essential References Not Discussed:**

The reviewer thought that the GD-based approach is mainly focused on the first-order parameter learning procedure, and the GD utilized in the manuscript is a non-mini-batch. Thus two related works may be considered:
1. Optimizer with Second Order Information like reference [1].
2. Minibatch sampling effect for parameter learning [2].
---
References:
[1]. Optimizing Neural Networks with Kronecker-factored Approximate Curvature (ICML 2015)
[2]. Stochastic Gradient Hamiltonian Monte Carlo (ICML 2014)

**Experimental Designs Or Analyses:**

1. In the context of deep learning, our training is always conducted by a mini-batch paradigm. Thus, during the evolution of mirror descent, the parameter may be perturbed by some noise. Specifically, according to the reviewer's understanding, the results should be:
$\theta_{t+1} = \theta_t - lr\nabla_\theta\mathcal{L}(\theta) + \mathcal{N}(0,\sigma^2)$. The noise may improve the model performance to some extent, and the authors have not considered this issue.
2. To the reviewer's understanding, the change of initial value may affect the model performance greatly during model training. However, this part of the experiments has not been investigated. It would be better to add some asymptotic convergence analysis to sidestep conducting this experiment?

**Methods And Evaluation Criteria:**

Yes, the methods and evaluation criteria make sense for demonstrating the derivation of convergence analysis, the authors also provide experiments on the Fashion MINST dataset to demonstrate the efficacy of the proposed approach.

**Other Comments Or Suggestions:**

See the abovementioned chat window.

**Other Strengths And Weaknesses:**

### Strengths
1. The topic is related to the ICML conferences.
2. The proposed approach is interesting.
3. The derivation is rigorous.

### Weaknesses
1. The organization of the main content should be revised. For example, the parameter learning of MOE should be more compact, this part can be moved to supplementary material.
2. The illustration of the convergence analysis can be reformulated. For example, an introduction of SymMoLinE can be given then the generalization to other MOE structures can be given in the latter content.
3. Figures 2 and 3 have not proposed the error bars.
4. The results in Tables 1 and 2 have not included the statistically significant test.

**Questions For Authors:**

See the abovementioned chat window.

**Relation To Broader Scientific Literature:**

The MOE is of great importance in the era of LLM, and the authors proposed a novel understanding of why the GD-based MOE learning procedure does not take effect and how to improve it from the perspective of mirror descent.

**Theoretical Claims:**

The reviewer attempts to check the derivation of the theorem. But there remain some issues to be addressed.
1. At the beginning, the authors are suggested to add a nomenclature to better facilitate the derivation.
2. On this basis, some symbols should be rectified. For example, $\mathbb{E}_{Z|\boldsymbol{x},y;\theta^t}$ should it be $q(Z|\boldsymbol{x},y;\theta^t)$? where the expectation operator is applied for some probability density function with the input $\boldsymbol{x}$ and $y$?
3. In appendix B.6, to the reviewer's understanding, the introduction of MIM is for proving the locally convex or relatively strongly convex region of $\mathcal{L}(\theta)$. Can we extend this condition to the generalized MOE model?
4. Can we directly use the concept of $\lambda$-convex during the proof to simplify and generalize the convergence condition?

---
References:
[1]. { Euclidean, Metric, and Wasserstein } Gradient Flows: an overview

---

> ### Author Rebuttal · Authors · 2025-04-01
>
> We sincerely thank the reviewer for their constructive feedback in highlighting key strengths and limitations of our work. Below, we address the main points raised:
>
> - **Theory:**
>   **The reviewer raised concerns about the nomenclature, use of KL vs L2 regularizer, the relationship between the MIM and the generalized result, and whether $\lambda$-convexity can be used to simplify and generalize the convergence condition.**
>   We agree that adding a dedicated notation section and a table of contents in the supplementary materials would improve clarity. Regarding the convergence conditions in Theorem 4.2, we acknowledge that assuming $L$ is $\lambda$-convex relative to the mirror map $A$ would suffice to establish the claim. Since $L$ is $\lambda$-convex almost surely for all $x$, the same holds in expectation, satisfying our convergence conditions. However, this imposes a stronger condition on $L$, which may not hold in general for MoE. Instead, we provide weaker conditions that relax this requirement to hold (1) in expectation with respect to $x$ and (2) only within a neighborhood around the true parameters. We verified these assumptions in Appendix B for special cases of MoE (SymMoLogE and SymMoLinE). A similar argument applies to the MIM conditions. Lastly, KL divergence regularizer makes sense in the context of MoE since it is invariant to the choice of parametrization while the L2 norm is not (see Kunstner et al., 2021 for an in depth discussion).
>
>   **Planned Revision:**
>   We will include a notation section and table of contents in the Appendix and allocate space after Theorem 4.2 to discuss how our conditions relate to standard $\lambda$-relative convexity.
>
> - **Mini-Batch:**
>   **In deep learning, training is typically done via mini-batches, which introduce noise that may impact model performance. This issue has not been considered.**
>   We acknowledge the limitation of focusing on full-batch EM and GD. However, our study maintains consistency by considering full-batch versions in both theoretical and empirical analyses.
>   Theoretically, since EM in our settings is equivalent to projected Mirror Descent, any theoretical result for mini-batch projected Mirror Descent would apply to EM.
>   Empirically, following your recommendation and that of other reviewers, we conducted an additional experiment on the mini-batch training of a mixture of 5 lightweight CNN experts on CIFAR-10. The results align with our FMNIST experiments (see our reply to reviewer z9yn for details).
>
>   **Planned Revision:**
>   We will discuss challenges in extending our theoretical results to mini-batch EM and include our mini-batch empirical results on CIFAR-10 in Appendix C.
>
> - **Initialization:**
>   **The reviewer suggests that initialization may significantly impact model performance and asks whether asymptotic convergence analysis could substitute for these experiments.**
>   Theorem 5.2 and Corollary B.1 guarantee that EM will converge sub-linearly to a stationary point of $L$ regardless of initialization.
>   Empirically, we tested the effect of initialization in the synthetic setting. We observed that for "close-enough" initialization, EM converged super-linearly to the true parameters, while for poor initialization, EM and GD converged sub-linearly, sometimes to suboptimal solutions. These findings align with previously known empirical results on EM (Kunstner et al., 2021; Xu and Jordan, 1996).
>
>   **Planned Revision:**
>   We will provide our empirical results on initialization for the synthetic setting in Appendix C.
>
> - **Three key areas for improvement:**
>   **1) Reorganizing the main text for clarity while moving relevant parts to the supplementary material for completeness.**
>   **2) Reordering key results for better comprehension.**
>   **3) Strengthening empirical results with additional statistical metrics.**
>
>   **Planned Revision:**
>   1) We will move lines 185-220 (left column) to the appendix and condense Section 5.1. For completeness, we will add a dedicated Appendix section covering all relevant introductory details.
>   2) We will provide a proof sketch for Theorem 4.2 (see our reply to reviewer yx5r).
>   3) Following your recommendation, we re-ran the FMNIST experiment (50 iterations, 25 instances, full batch) to include error bars and a paired t-test (implemented using scipy's ``ttest_rel"). We are also conducting this analysis for synthetic and CIFAR-10 experiments. Our statistical results are:
>      - **EM vs GD:** T-statistic = 24.32, p-value = 0.000.
>      - **Gradient EM vs GD:** T-statistic = 17.81, p-value = 0.000.
>      We conclude that our results are statistically significant. See the updated Figure 3:
>      [https://prnt.sc/p9rfbnIfycX2](https://prnt.sc/p9rfbnIfycX2).
>
> **Lastly, we thank the reviewer for any additional comments and encourage the reviewer to read "Summary of Planned Revisions Based on All Reviewers' Feedback" in rebuttal to reviewer z9yn. Have a nice day!**

---

> > ### Comment · Reviewer_kaEY · 2025-04-05
> >
> > Thank you for your detailed response. I intend to improve the score for this work. However, please ensure that the revised manuscript includes the theoretical discussion on mini-batch training as well as the supplementary experimental results.

---

### Official Review · Reviewer_z9yn · 2025-03-11

**Overall Recommendation:** 3

**Summary:**

This paper focuses on integrating MoE optimization and EM algorithm. The authors first proved the theoretical guarantees of EM algorithm for training MoE models. Then, the authors focus on the special case of mixture of 2 linear or logistic experts and analyze the guarantees for the linear convergence. Next, the authors conduct experiments on small-scale data to demonstrate that EM algorithm can outperforms the GD algorithm.

**Claims And Evidence:**

This paper provides detailed theoretical proof and experimental evaluation on special cases.

**Essential References Not Discussed:**

The reference is sufficient

**Experimental Designs Or Analyses:**

As mentioned in “Methods And Evaluation Criteria”, the authors just select two special case to conduct experiments. Moreover, the used datasets are synthetic and small scale. Furthermore, the compared method is only GD algorithm. All of these settings limit the contribution and convincing of this paper.

**Methods And Evaluation Criteria:**

The provided results are synthetic and small-scale datasets. It is hard to distinguish whether the proposed method is still effective on large-scale datasets. Moreover, this paper use two special cases to conduct the experiments, which is not so convincing. Moreover, the authors just compare EM algorithms with GD algorithm, which also limits the contribution of this paper.

**Other Comments Or Suggestions:**

1.	An additional notation table is helpful to follow the proof of this paper.

2.	Experiments on large-scale datasets or diverse datasets can enhance the convincing of this paper.

**Other Strengths And Weaknesses:**

Please refer to the “Methods And Evaluation Criteria” and “Experimental Designs Or Analyses” parts

**Questions For Authors:**

Please refer to the “Methods And Evaluation Criteria” and “Experimental Designs Or Analyses” parts

**Relation To Broader Scientific Literature:**

The key contribution can provide some insights for training better MoE methods or designing MoE-based methods.

**Theoretical Claims:**

The theoretical claims in this paper are convincing. I have checked all the claims in the main pages.

---

> ### Author Rebuttal · Authors · 2025-04-01
>
> We sincerely thank the reviewer for their constructive feedback in highlighting key strengths and limitations of our work. Below, we address the main points raised:
>
> - **The provided results are synthetic and small-scale datasets. It is hard to distinguish whether the proposed method is still effective on large-scale datasets. Moreover, this paper uses two special cases to conduct the experiments, which is not so convincing. Moreover, the authors just compare EM algorithms with GD algorithm, which also limits the contribution of this paper.**
>
>   We agree with the reviewer that the paper does not feature large-scale experiments which could help motivate practical use of EM for large-scale MoE training. As this paper is a purely theoretical work within the broad EM and MoE literature that provides correspondence between EM and MD and subsequent SNR conditions for linear convergence, we chose to restrict the empirical experiments to supporting our theoretical claims on the two sub-cases considered in Appendix B.
>
>   Nonetheless, based on your review, we scaled our experiments from training a mixture of 2 logistic experts model on FMNIST to training a mixture of 5 lightweight CNN models on CIFAR-10 (620k parameters, batch_size=2500, 50 epochs, and 25 instances). We report that compared to GD, EM had an average final test accuracy of **41.6%**, which is **7.1%** higher than GD. Similarly, we report that compared to GD, Gradient EM had an average final test accuracy of **37.4%**, which is **2.9%** higher than GD, while GD had a final test accuracy of **34.5%**. A screenshot of the resulting plot can be found at: [https://prnt.sc/ZXBLKD2e8bN2](https://prnt.sc/ZXBLKD2e8bN2).
>
>   Moreover, as per reviewer kaEY, we are also providing a paired t-test to check for statistical significance.
>   - For **EM vs GD**, we report a **T-statistic of 25.15** (p-value of **0.000**).
>   - For **Gradient EM vs GD**, we report a **T-statistic of 22.88** (p-value of **0.000**).
>
>   We conclude that our results are **statistically significant**.
>
>   **Planned Revision:**
>   *We will include our CIFAR experiments in Appendix C to highlight preliminary results on scaling EM to 'large'-scale MoE.*
>
> - **An additional notation table is helpful to follow the proof of this paper.**
>   **Planned Revision:**
>   *We will include a notation section and table of contents at the beginning of the Appendix.*
>
> # Summary of Planned Revisions Based on All Reviewers' Feedback
>
> As part of our commitment to addressing the reviewers' valuable suggestions and concerns, we have outlined a set of planned revisions. We sincerely appreciate the constructive feedback and believe that our responses, along with these well-defined and feasible revisions, will demonstrate the paper’s readiness for acceptance into the conference.
>
> 1) **Readability:**
>    To enhance clarity and accessibility, we will:
>    A) streamline the main text by moving portions of the background on EM theory and MoE parameter estimation to the supplementary material,
>    B) introduce a dedicated **notation section and table of contents** in the Appendix to improve readability, and
>    C) use the additional space in the main text to provide a **proof sketch of Theorem 4.1** (see our reply to Reviewer yx5r to see our suggested proof sketch), highlighting key proof techniques that address previous gaps in the literature.
>
> 2) **Discussions:**
>    We will:
>    A) provide a thorough discussion regarding the **scalability of EM to large-scale and sparse MoE** as well as challenges in extending our **theoretical framework to mini-batch EM**, and
>    B) expand our discussion on the class of models for which our results hold (general MoE), clarifying how our convergence conditions in Theorem 4.2 and Corollary B.1 compare to standard convexity assumptions for GD. Specifically, we will elaborate on their relationship to $\lambda$-(relative) convexity in the discussion following Theorem 4.2.
>
> 3) **Experiments:**
>    To strengthen our empirical analysis, we will include **error bars in the plots and perform a statistical significance test** (e.g., a paired t-test) to support the claim that EM provides a meaningful improvement over GD (we have obtained these results, please see our reply to reviewer kaEY and z9yn).
>
>    In Appendix C, we will:
>    A) add a discussion connecting theoretical results to the experiments such as verifiability of the linear rate and satisfiability of assumptions $A_1$-$A_3$,
>    B) include **initialization experiments** (see our reply to reviewer kaEY),
>    C) provide our **experiments on mini-batch training of a mixture of 5 lightweight CNNs on CIFAR-10** (see our reply to reviewer z9yn), and
>    D) provide all necessary implementation details in Appendix C (such as learning rate, precise model architecture, number of instances, etc.).
>
> We are looking forward to hearing your valuable feedback on our responses and proposed revisions.

---

> > ### Comment · Reviewer_z9yn · 2025-04-06
> >
> > The authors have answered all my questions and I have no further questions.

---

### Official Review · Reviewer_yx5r · 2025-03-13

**Overall Recommendation:** 3

**Summary:**

This paper studies the relationship between EM for general MoE with projected mirror descent algorithms. Based on the equivalence between EM and mirror descent, this work provides non-asymptotic convergence rates for training MoE with EM.

**Claims And Evidence:**

Yes, this work provides solid proofs to the theorems and claims.

**Essential References Not Discussed:**

I am not aware of any missing references.

**Experimental Designs Or Analyses:**

My expertise in MoE experiments is limited, making it difficult to assess their validity. However, I believe the experimental results can support the theoretical claims to some extents as a theoretical paper.

**Methods And Evaluation Criteria:**

This paper mainly focus on theoretically understanding the relationship between the EM method and the mirror descent method. Their theory demonstrates that the EM method is theoretically effective for training MoE.

**Other Comments Or Suggestions:**

I suggest that the author provide a more technical highlight of the challenges or analytical innovations involved in establishing equivalence in MoE, compared to [Kunstner et al., 2021]. This could enhance the paper’s impact and make it more accessible for future research to build upon.

**Other Strengths And Weaknesses:**

Strengths:

- This paper is well-written and easy-to-follow. It clearly presents its contributions, and the relevant theoretical assumptions are well-stated.
- This work provides a new approach for analyzing the convergence rate of MoE training when using EM method.
- When the $\beta$-update in the EM method has a closed-form solution (as is the case for linear experts, mentioned in the paper), the EM approach does not require tuning learning rates, potentially offering a more robust learning algorithm.

Weakness:

- The assumptions on the distribution are restricted to the exponential family, which may be overly idealized compared to realistic applications.
- The EM method might potentially introduce additional computational costs and is more difficult to implement compared to the gradient descent method, limiting the applicability of the proposed method in the paper.

**Questions For Authors:**

- In line 147, could an additional parameter be introduced to consider the distribution $p(z \mid x; w) \propto \exp(\gamma x^\top w)$?
- The analysis in the paper relies on the convexity of the function L. How can the obtained results be further interpreted in the context of non-convex optimization in neural networks?

**Relation To Broader Scientific Literature:**

The key contributions in this paper primarily follow [Kunstner et al., 2021]. This work extends the results for MoE model and considers the mixtures of regression setting.

**Theoretical Claims:**

I briefly checked the proof of Theorem 4.2, the main theorem.

---

> ### Author Rebuttal · Authors · 2025-04-01
>
> We begin by extending our thanks to the reviewer for the very constructive feedback in highlighting important strengths and limitations of our work. We address the main points raised below:
> - **Concerns:**
>   **1) The assumptions on the distribution are restricted to the exponential family. 2) The EM method might potentially introduce additional computational costs and is more difficult to implement compared to GD.**
>   1) Whilst we agree with the reviewer that our results only hold for realizable settings where the conditional distribution belongs to an exponential family, we maintain that the theoretical results obtained in this paper constitute a significant improvement over the previous literature, extending the connection between EM and MD from simpler model classes like Mixture of Gaussians to a more practically relevant class today like that of General MoE described in Section 2.
>   2) To address this concern, the empirical results in Figures 1, 2, and 3 also include the Gradient EM algorithm ("GradEM") which is a variant of the EM algorithm with per-iteration cost identical to that of GD. Even then, we still observe an improvement over GD, though less significant.
>
>   **Planned Revision:**
>   *We will include a thorough discussion of the limitations, discussing the scalability of the EM algorithm to large-scale and sparse MoE.*
>
> - **Suggestion:**
>   **Author provide a more technical highlight of the challenges or analytical innovations involved in establishing equivalence in MoE, compared to [Kunstner et al., 2021].**
>  This is a very good point. We clarify that in Kunstner et al. Theorem 1, the assumption that $p(y,z,x|\theta)$ is an exponential family of distributions allows to decompose $Q(\theta|\theta^t)$ to show direct equivalence between EM and MD. In our results, we relax this assumption, only requiring the conditional distribution $p(y,z|x, \theta)$ to belong to an exponential family of distributions (this includes more complicated mixtures like MoE). With this relaxation, we can only decompose $Q(\theta|\theta^t)$ inside the expectation with respect to x. Then, we utilize a  decomposition on $\nabla L$ to relate the expression inside the expectation to the point-wise (in x) iterations of MD. Finally, this allows we show direct equivalence between the iterations of EM and a projection over the iterations of MD.
>
>   **Planned Revision:**
>   *We will make space in the main text to provide a proof sketch of Theorem 4.1 that will highlight the proof techniques that allow us to bridge this key gap in the literature. See our proposed proof sketch in the screenshot: https://prnt.sc/wMDRIY7neary.*
>
> - **Questions:**
>   **1) In line 147, could an additional parameter be introduced to consider the distribution $p(z|x;w) \propto \exp\{\gamma x^\top w\}$?**
>   We clarify that our results hold for any probability functions $p(z|x;w)$ and $p(y|z,x;\beta)$ so long as $p(y,z|x,\theta)$ belongs to an exponential family. Since the norm of w is unbounded in our setting, introducing the new parameter $\gamma$ does not provide a generalization of the problem.
>
>   **2) The analysis in the paper relies on the convexity of $L$. How can the obtained results be further interpreted in the context of non-convex optimization?**
>
>   We clarify that none of our results explicitly require that the objective function $L$ is convex, as such an assumption does not generally hold for MoE. Instead, we require weaker conditions; Theorem 4.2 requires that the algorithm is initialized inside some neighborhood around the true parameter $\theta^*$ that satisfies inequality (20) (or (21)), wherein a similar notion to convexity is maintained in expectation over feature variable x (similarly for Corollary B.1). We emphasize that the form of convergence results we obtain is consistent with previous works on EM such as [Kunstner et al. 2021, Balakrishnan et al. 2017].
>
>   If it is assumed that $p(y|x,z;\beta)$ is Gaussian, our result in Theorem 4.1 can be readily applied to the setting where the experts are neural networks. In appendix B, we uncovered an interpretable result for SymMoLogE and SymMoLinE that, when training MoE with EM, one should initialize the gating parameter with a norm much smaller than that of the expert parameters, a finding that was observed empirically on page 10 of [Switch Transformers: Scaling to Trillion Parameter Models with Simple and Efficient Sparsity, 2022].
>
>   **Planned Revision:**
>   *We will enhance the discussion of the class of models for which our results hold and how our conditions for convergence differ from standard assumptions of convexity. We will also add a discussion on how to scale our theory to more complex experts such as neural networks.*
>
> **Lastly, we thank the reviewer for any additional comments and encourage the reviewer to read "Summary of Planned Revisions Based on All Reviewers' Feedback" in rebuttal to reviewer z9yn. Have a nice day!**

---

### Official Review · Reviewer_KP5h · 2025-03-13

**Overall Recommendation:** 4

**Summary:**

This paper discusses the relationship between the EM algorithm and MoE models. In particular, a relationship is shown between the EM update of the experts and router parameters and Mirror Descent (MD) with a specific expression for the Bregman divergence regulariser.

**Claims And Evidence:**

The paper claims to extend the result of Kunstner et al. (2021) beyond exponential family mixtures, which is that an EM update of MoE can be interpreted as a MD with KL divergence regularizer. This includes Mixture of Linear and Logistic Experts.

**Essential References Not Discussed:**

N/A.

**Experimental Designs Or Analyses:**

The experiments are conducted on both synthetic data with SymMoLinE and SymMoLogE, and Fashion MNIST with random color image inversion to model the need for 2 experts. It shows the effectiveness of the EM algorithm in convergence speed and performance in those settings.

**Methods And Evaluation Criteria:**

The paper demonstrates the relevance of the result by conducting experiments on both synthetic data and Fashion MNIST, where the goal is to compare the convergence of vanilla GD, gradient-based EM and EM. It is shown that (i) EM converges faster than gradient-based EM, and (ii) EM outperforms the vanilla GD algorithm in terms of final performance (e.g. accuracy or cross-entropy), which makes sense.

**Other Comments Or Suggestions:**

Typo: It should be $p(y | x, z ; \theta)$ instead of $p(y | x ; \theta)$ in the second equality when deriving the decomposition (10) in section 3 (L 203).

It would be great to improve the connection between the experimental and theoretical results. Question 2: To what extent are the assumptions of $A_1-A_3$ verified in the experiments? Question 3: What are the practical implications for the future of MoE of this connection with MD?

**Other Strengths And Weaknesses:**

Strengths: The paper is well-written and is easy to read. The scope of the theoretical contribution compared to Kunstner et al., (2021) is clear, and the results are relevant in the context of MoE, as it helps to improve their interpretation. The experimental results also highlight the practical advantages of EM-based optimization.

Weaknesses: Limitations of the work are not mentioned. Question 1: Are there still settings where standard GD (and momentum-based variants, such as Adam) has advantages over EM-based optimization in the context of MoE? Furthermore, more details about the experimental setup (needed to reproduce the results: e.g., batch size, and learning rate used) are required in Appendix C.

**Questions For Authors:**

Question 4: Did the asymptotic convergence error of the GD in Figures 1 and 2 reach the one of the EM and Gradient EM?
Question 5: How do you explain that the Single Expert in Table 1 trained without Random Inversion reaches a better accuracy ($83.2  \\%$) compared to the three 2-Component MoLogE methods in Table 2?

**Relation To Broader Scientific Literature:**

The paper is mainly built upon Kunstner et al. (2021), which derived a connection between EM and MD, and Makkuva et al. (2019), Becker et al. (2020) which relates EM and MoE. It extends Kunstner et al. (2021) to general MoE that includes non-exponential family mixtures.

**Theoretical Claims:**

The main theoretical claim is that EM is a projected Mirror Descent algorithm for general MoE, extending the result of Kunstner et al. (2021), which focuses on exponential family mixtures. This conducted in Convergence Guarantees is adapted for General MoE in Section 4.1. The result is then applied to SymMoLinE and SymMoLogE, providing a closed-form expression of the Mirror Map in this special case.

---

> ### Author Rebuttal · Authors · 2025-04-01
>
> We begin by extending our thanks to the reviewer for the very constructive feedback in highlighting important strengths and limitations of our work. We address the main points raised below:
> - **The limitations of the work are not mentioned**
>
>  Thanks for your feedback. We would like to mention that the main limitations of our theoretical results are that they hold for the realizable setting and extension to agnostic setting require further investigation. Also, currently we only investigated the full-batch EM, while we believe extension to mini-batch case is possible, as illustrated by our new experiments. We will include these points in the revised paper, and in particular, will mention them in the revised conclusion.
>
> - **More details about the experimental setup (needed to reproduce the results: e.g., batch size, learning rate used) are required in Appendix C.**
>
> That is a valid point. In the revised paper, we will add all details necessary to reproduce our empirical results to Appendix C. Thanks for your comment.
>
> - **Question 1:**
>   **Are there still settings where standard GD (and momentum-based variants, such as Adam) has advantages over EM-based optimization in the context of MoE?**
>
> This is a very good point. We have theoretically shown that EM requires relative convexity to hold in expectation and within a neighborhood of the true parameters to ensure linear convergence. These sufficient conditions differ from those of GD, which requires strong convexity within a neighborhood around the true parameters for linear convergence. The key insight is that we characterize the conditions under which EM enters and remains in the region where linear convergence holds, whereas it remains unclear under what conditions GD iterates satisfy the local neighborhood assumptions required for linear convergence. That said, this does not imply that EM is necessarily superior to GD or vice versa. We believe there are settings where EM may converge linearly with higher probability than GD and its variants, and vice versa.
>
> - **Question 2:**
>   **To what extent are the assumptions A₁ - A₃ verified in the experiments?**
>
>   For the synthetic experiments, assumptions A₁-A₃ are verifiably true; the conditions of Theorem 5.1 hold exactly. As for the experiment on FMNIST, we are not in the realizable setting, so it is safe to assume the assumptions do not hold. Still, the empirical results suggest that EM is robust to non-realizability. We will add this point to the revised paper. Thanks for raising this point.
>
> - **Question 3:**
>   **What are the practical implications for the future of MoE of this connection with MD?**
>
>   An example of a practical implication we rigorously showed in Appendix B.6 and B.7 is that it is preferable to initialize the gating parameters of the MoE with a much smaller norm relative to that of the expert parameters. This finding corroborates similar **empirical** findings reported on page 10 of *Switch Transformers: Scaling to Trillion Parameter Models with Simple and Efficient Sparsity, 2022*. These corroborating results suggest that initializing the gating parameters with a small norm is a universally good heuristic when training MoE (We will add this to the discussion).
>
> - **Question 4:**
>   **Did the asymptotic convergence error of GD in Figures 1 and 2 reach that of EM and Gradient EM?**
>
>   This is a very good question. Interestingly, we did not observe that this was the case, but we cannot deterministically conclude that it wouldn't have if we kept training for many more iterations.
>
> - **Question 5:**
>   **How do you explain that the Single Expert in Table 1 trained without Random Inversion reaches a better accuracy (83.2%) compared to the three 2-Component MoLogE methods in Table 2?**
>
>   Firstly, we clarify that results in Table 2 are obtained on the mixed dataset with random inversions whereas the single expert with accuracy 83.2% was trained on the homogeneous dataset. There are three reasons for this:
>
>   1. **The mixed dataset is more difficult to learn than the homogeneous one.** This is observed as we report the single expert having a test accuracy of 10.2% after training on the mixed dataset, whereas training on the homogeneous dataset yielded a test accuracy of 83.2%.
>   2. **Information sharing:** Each expert now trains on half as many images as in the base case.
>   3. **Task difficulty:** Learning both expert and gating parameters to correctly separate the inverted and non-inverted images—and then correctly label each—is a more difficult task than labeling all inverted or non-inverted images. In fact, we observed in our experiments that the best-performing MoE model was learning to route samples based on whether they were inverted or non-inverted.
>
> **Lastly, we thank the reviewer for any additional comments and encourage the reviewer to read "Summary of Planned Revisions Based on All Reviewers' Feedback" in rebuttal to reviewer z9yn. Have a nice day!**

---

> > ### Comment · Reviewer_KP5h · 2025-04-02
> >
> > Thank you for your response, I appreciate it! For the next revision of your paper, could you provide an updated version of Figures 1 and 2 (perhaps in the Appendix) with additional iterations? This would help us better understand the asymptotic behavior of GD. Additionally, you could run the Fashion MNIST experiments with multiple training seeds. I will raise my score accordingly, I think the paper deserves acceptance.

---

> > > ### Author Response · Authors · 2025-04-03
> > >
> > > Thank you for your reply and comments!
> > >
> > > As per your suggestion, we have updated our Figures 1-3 plots to include more iterations as can be found at the following screenshot link: https://prnt.sc/40azK725P6SF. We note that, for Figure 1 b), we could not plot the y-axis on the log-scale like before because the error bars would not play nice when the log-likelihood error was of the order of $10^{-10}$ and below. We've also renewed our test of significance and confirm that the results have not changed (p-value of 0.000 in each instance).
> > >
> > > We also clarify that in each of the 25 FMNIST instances, we sample a new initialization of the MoE (similarly in the synthetic experiments). As a result, we do not believe that changing the seed will impact the results. Still, we will make this change for our mini-batch experiment on CIFAR-10.

---

### Decision · Program_Chairs · 2025-05-01

**Decision:**

Accept (poster)

**Comment:**

This paper revisits the idea of learning mixtures of experts (MoEs) using the classical expectation-maximization (EM) algorithm. The main contribution of the paper is making the connection between the EM for MoEs and projected mirror descent with unit step size and a KL regularizer; this connection enables the development of convergence rates leveraging theory from mirror decent. The paper concludes with empirical studies comparing to MoEs trained using gradient descent, with the goal of highlighting that EM is more suitable than gradient descent for fitting specific models.

Overall reviewers were positive about the paper and unanimously have recommended acceptance. The reviewers agreed taht the paper is well-written, the scope of the theoretical contribution is clear and its relationship to related work is well-discussed. The reviewers also appreciated the experimental results as a means to highlight practical advantages of EM-based optimization. Thus, I recommend accepting this paper.

Some points that the reviewers would like to see discussed in the revision:
* Limitations of the approach
* More discussion comparing to the setting of gradient descent
* More discussion regarding the empirical setup for reproducibility purposes
* Some revisions to streamline the theoretical results discussed

Overall, most of these points were addressed in the author rebuttal, and the authors provided a very thorough plan for revising the paper according to the reviewers' comments.